# A Finite-Time Analysis of Distributed Q-Learning

## Abstract

Multi-agent reinforcement learning (MARL) has witnessed a remarkable surge in interest, fueled by the empirical success achieved in applications of single-agent reinforcement learning (RL). In this study, we consider a distributed Q-learning scenario, wherein a number of agents cooperatively solve a sequential decision making problem without access to the central reward function which is an average of the local rewards. In particular, we study finite-time analysis of a distributed Q-learning algorithm, and provide a new sample complexity result of $\tilde{\mathcal{O}}\left(\max\left\{\frac{1}{\epsilon^2}\frac{t_{\mathrm{mix}}}{(1-\gamma)^6 d_{\mathrm{min}}^4}, \frac{1}{\epsilon}\frac{\sqrt{|\mathcal{S}||\mathcal{A}|}}{(1-\sigma_2(\boldsymbol{W}))(1-\gamma)^4 d_{\mathrm{min}}^3}\right\}\right)$ under tabular lookup setting for Markovian observation model.

## 1 Introduction

Multi-agent reinforcement learning (MARL) aims to solve a sequential decision making problem, where a number of agents sharing an environment collaborates. Accompanied by advancements in algorithms (Sunehag et al., 2017; Rashid et al., 2020), MARL has shown impressive success in various fields such as robotics (de Witt et al., 2020) and autonomous driving (Shalev-Shwartz et al., 2016). Beyond its empirical success, there has also been notable interest in theoretical investigations (Zhang et al., 2018b; Dou et al., 2022).

MARL has been studied under various scenarios including an access to central reward function (Tan, 1993; Claus and Boutilier, 1998; Littman, 2001). In particular, our interest lies in the the distributed learning paradigm where agents collaborate to solve a shared problem, constrained to communicate solely with their neighboring agents and does not have access to central reward function. Such setting has came of interest due to its wide applications (Blumenkamp et al., 2022; Prabuchandran et al., 2014; Zhao et al., 2021). Compared to scenarios where a centralized coordinate exists, the distributed paradigm has advantage in terms of privacy-preservation and scalability. One notable example is the distributed adaptation of temporal-difference (TD) learning, as demonstrated in studies by Doan et al. (2019); Wang et al. (2020); Lim and Lee (2023), to name a few.

Meanwhile, in the literature of single-agent RL, Q-learning (Watkins and Dayan, 1992) is one of the most important algorithms in RL. The non-linear max-operator in Q-learning algorithm imposes difficulty in the analysis, and its non-asymptotic analysis has been an active research area recently (Even-Dar et al., 2003; Chen et al., 2021; Lee et al., 2023; Li et al., 2024). However, distributed learning framework for Q-learning has not been studied in detail. In particular, distributed Q-learning has been studied in an asymptotic sense (Kar et al., 2013), i.e., the algorithm converges over time as it approaches infinity, or in a non-asymptotic sense under additional assumptions on the problem (Heredia et al., 2020; Zeng et al., 2022b). Wang et al. (2022) studied a version of distributed Q-learning in tabular setting but differs from the one in Kar et al. (2013). This motivates our study to understand its non-asymptotic behavior under tabular setup, i.e., all the state-action values are stored in a table. Our contribution can be summarized as follows:

1. For Markovian observation model, we provide the sample complexity $\tilde{\mathcal{O}}\left(\max\left\{\frac{t_{\mathrm{mix}}}{\epsilon^2}\frac{1}{(1-\gamma)^6 d_{\mathrm{min}}^4}, \frac{1}{\epsilon}\frac{\sqrt{|\mathcal{S}||\mathcal{A}|}}{(1-\sigma_2(\boldsymbol{W}))(1-\gamma)^4 d_{\mathrm{min}}^3}\right\}\right)$ in terms of the infinity norm under

tabular setting. We derive, for the first time, the finite-time analysis of QD-learning (Kar et al., 2013) in its original form, which is one of the most fundamental and widely used distributed Q-learning methods. While several works have addressed other types of distributed Q-learning, the analysis of QD-learning has remained unexplored until now. Furthermore, we also provide a sample complexity result for the independent and identically distributed (i.i.d.) observation model.

2. Our analysis relies on switched system modeling of Q-learning, providing new insights for interpretation of distributed Q-learning algorithms. We show that the distributed Q-learning also allows switched system interpretation as in the single-agent case.

**Related Works:**

The non-asymptotic behavior of distributed TD-learning was studied in Doan et al. (2019); Sun et al. (2020); Wang et al. (2020); Lim and Lee (2023), which were motivated from the distributed optimization and control literature (Nedic and Ozdaglar, 2009; Wang and Elia, 2010; Pu and Nedić, 2021). Distributed versions of various TD-learning algorithms were investigated in Macua et al. (2014); Lee et al. (2018). As for actor-critic algorithm (Konda and Tsitsiklis, 1999), its extension to distributed setting was studied in Zhang et al. (2018a;b); Zhang and Zavlanos (2019); Zeng et al. (2022a). Meanwhile, Yang et al. (2023) considered a distributed policy gradient approach. Moreover, Zhang et al. (2021) investigated distributed algorithm for fitted Q-iteration, which is similar to solving a least squares problem. Furthermore, a line of research has focused on dealing with exponential scaling in the action space Lin et al. (2021); Qu et al. (2022); Zhang et al. (2023); Gu et al. (2024).

The distributed Q-learning algorithm under the setting when only the local reward is observable, was first studied by Kar et al. (2013). They proposed the so-called QD-learning proving asymptotic convergence using two-time scale stochastic approximation approaches. Zeng et al. (2022b); Heredia et al. (2020) proved finite-time bounds of distributed Q-learning with linear function approximation. However, the works require additional strong assumptions, which may not hold even in the tabular setup. In particular, Zeng et al. (2022b) considered a strongly monotone condition to hold, and Heredia et al. (2020) posed a particular assumption on the state-action distribution. Wang et al. (2022) studied a distributed Q-learning model motivated from the adapt-then-combine scheme (Chen and Sayed, 2012) in the distributed optimization literature and provided a sample complexity bound in terms of high-probability.

Considering a single-agent case, the non-asymptotic analysis of Q-learning has made great success. An incomplete list is provided in the following: An early result by Even-Dar et al. (2003) studied the sample complexity under i.i.d. observation model. Lee et al. (2023) developed a switched system method to analyze the behavior of Q-learning. Qu and Wierman (2020) considered a shifted Martingale approach to deal with the Markovian observation model. Li et al. (2024) proved the sample complexity using refined analysis under the Markovian observation model.

Meanwhile, a separate line of research focusing on multi-agent problems is the federated reinforcement learning literature (Khodadadian et al., 2022; Woo et al., 2023; Zheng et al., 2023). This approach differs from the distributed learning scenario in two key aspects: it employs a centralized controller, and all agents share a common reward function.

The paper is organized as follows: Section 2 provides background for the MARL setting. Section 3 provides result under i.i.d. observation model and sketch of the proof. The result for Markovian observation model is provided in Section 4.

## 2 Preliminaries

### 2.1 Multi Agent MDP

A multi-agent Markov decision process (MAMDP) consists of the tuple $(\mathcal{S}, \{\mathcal{A}_i\}_{i=1}^N, \mathcal{P}, \{r^i\}_{i=1}^N, \gamma)$, where $\mathcal{S} := \{1, 2, \ldots, |\mathcal{S}|\}$ is the finite set of states, $\mathcal{A}_i := \{1, 2, \ldots, |\mathcal{A}_i|\}$ is the finite set of actions for each agent $i \in \mathcal{V}$, $\mathcal{P} : \mathcal{S} \times \prod_{i=1}^N \mathcal{A}_i \times \mathcal{S} \to [0, 1]$ is the transition probability, and $r^i : \mathcal{S} \times \prod_{i=1}^N \mathcal{A}_i \times \mathcal{S} \to \mathbb{R}$ is the reward function of agent

$i \in \mathcal{V}$. We will use the notation $\mathcal{A} := \prod_{i=1}^N \mathcal{A}_i = \{1, 2, \ldots, |\mathcal{A}|\}$ where tuple of actions are mapped to unique integer. $\gamma \in (0, 1)$ is the discount factor.

At time $k \in \mathbb{N}$, the agents share the state $s \in \mathcal{S}$, and each agent $i \in \mathcal{V}$ selects an action $a_i \in \mathcal{A}_i$ following its own policy $\pi^i : \mathcal{S} \to \Delta^{|\mathcal{A}_i|}$. The collection of the actions selected by each agents are denoted as $\boldsymbol{a} = (a_1, a_2, \ldots, a_N)$, and transition occurs to $s' \sim \mathcal{P}(s, \boldsymbol{a}, \cdot)$. Each agents receives local reward $r^i(s, \boldsymbol{a}, s')$, which is not shared with other agents.

The main goal of MAMDP is to find a deterministic optimal policy, $\pi^* := (\pi^1, \pi^2, \ldots, \pi^N)$ : $\mathcal{S} \to \mathcal{A}$ such that the average of cumulative discounted rewards of each agents is maximized: $\pi^* := \arg\max_{\pi \in \Omega} \mathbb{E}\left[\sum_{k=0}^{\infty} \sum_{i=1}^N \frac{\gamma^k}{N} r^i(s_k, \boldsymbol{a}_k, s_{k+1}) \middle| \pi\right]$, where $\Omega$ is the set of possible deterministic policies, and $\{(s_k, \boldsymbol{a}_k)\}_{k \geq 0}$ is a state-action trajectory generated by Markov chain under policy $\pi$. The Q-function for a policy $\pi : \mathcal{S} \to \mathcal{A}$, denotes the average of cumulative discounted rewards of each agents following the policy $\pi$, i.e., $Q^\pi(s, \boldsymbol{a}) := \mathbb{E}\left[\sum_{k=0}^{\infty} \sum_{i=1}^N \frac{\gamma^k}{N} r_{k+1}^i \middle| \pi, (s_0, a_0) = (s, a)\right]$ for $s \in \mathcal{S}, \boldsymbol{a} \in \mathcal{A}$, where $r_{k+1}^i := r^i(s_k, \boldsymbol{a}_k, s_k')$. The optimal Q-function, $Q^{\pi^*}$, which is the Q-function induced by the optimal policy $\pi^*$, is denoted as $Q^*$. The optimal policy can be recovered via a greedy policy over $Q^*$, i.e., $\pi^*(s) = \arg\max_{\boldsymbol{a} \in \mathcal{A}} Q^*(s, \boldsymbol{a})$ for $s \in \mathcal{S}$. The optimal Q-function, $Q^*$ satisfies the following so-called optimal Bellman equation (Bellman, 1966):

$$Q^*(s, \boldsymbol{a}) = \mathbb{E}\left[\frac{1}{N}\sum_{i=1}^N r^i(s, \boldsymbol{a}, s') + \gamma \max_{\boldsymbol{u} \in \mathcal{A}} Q^*(s', \boldsymbol{u})\right], \quad \forall s \in \mathcal{S}, \boldsymbol{a} \in \mathcal{A}. \tag{1}$$

Since each agent only has an access to its local reward $r^i$, it is impossible to learn the central optimal Q-function without sharing additional information among the agents. However, we assume that there is no central coordinator that can communicate with all the agents. Instead, we will consider a more restricted communication scenario where each agent can share its learning parameter only with a subset of the agents. This communication constraint can be caused by several reasons such as infrastructures, privacy, and spacial topology. The communication structure among the agents can be described by an undirected simple connected graph $\mathcal{G} := (\mathcal{V}, \mathcal{E})$, where $\mathcal{V}$ denotes the set of vertices and $\mathcal{E} \subset \mathcal{V} \times \mathcal{V}$ is the set of edges. Each agent will be described by a vertex $v \in \mathcal{V} := \{1, 2, \ldots, N\}$, where $N$ is the number of agents. Moreover, each agent $i \in \mathcal{V}$ only communicates with its neighbours, denoted as $\mathcal{N}_i := \{j \in \mathcal{V} \mid (i, j) \in \mathcal{E}\}$.

To further proceed, we will use the following matrix and vector notations: $\boldsymbol{P} := \begin{bmatrix} \boldsymbol{P}_{1,1} & \boldsymbol{P}_{1,2} & \cdots & \boldsymbol{P}_{|\mathcal{S}|,|\mathcal{A}|} \end{bmatrix}^\top$, $\boldsymbol{R}^i := \begin{bmatrix} \boldsymbol{R}_1^{i\top} & \cdots & \boldsymbol{R}_{|\mathcal{S}|}^{i\top} \end{bmatrix}^\top$ where $\boldsymbol{P}_{s,\boldsymbol{a}} \in \mathbb{R}^{|\mathcal{S}|}$ and $\boldsymbol{R}_s^i \in \mathbb{R}^{|\mathcal{A}|}$ are column vectors such that $[\boldsymbol{P}_{s,\boldsymbol{a}}]_{s'} = \mathcal{P}(s, \boldsymbol{a}, s')$ for $s' \in \mathcal{S}$, and $[\boldsymbol{R}_s^i]_{\boldsymbol{a}} = \mathbb{E}\left[r^i(s, \boldsymbol{a}, s') \mid s, \boldsymbol{a}\right]$, respectively. We assume that $||R^i||_\infty \leq R_{\max}$ for some positive real number $R_{\max}$. Throughout the paper, we will represent a policy in a matrix form. A greedy policy over $\boldsymbol{Q} \in \mathbb{R}^{|\mathcal{S}||\mathcal{A}|}$, which is denoted as $\pi_{\boldsymbol{Q}} : \mathcal{S} \to \mathcal{A}$, i.e., $\pi_{\boldsymbol{Q}}(s) = \arg\max_{\boldsymbol{a} \in \mathcal{A}}(\boldsymbol{e}_s \otimes \boldsymbol{e}_{\boldsymbol{a}})^\top \boldsymbol{Q}$, can be represented as a matrix as follows:

$$\boldsymbol{\Pi}^{\boldsymbol{Q}} := \begin{bmatrix} \boldsymbol{e}_1 \otimes \boldsymbol{e}_{\pi(1)} & \boldsymbol{e}_2 \otimes \boldsymbol{e}_{\pi(2)} & \cdots & \boldsymbol{e}_{|\mathcal{S}|} \otimes \boldsymbol{e}_{\pi(|\mathcal{S}|)} \end{bmatrix}^\top \in \mathbb{R}^{|\mathcal{S}| \times |\mathcal{S}||\mathcal{A}|},$$

where $\boldsymbol{e}_s$ and $\boldsymbol{e}_{\boldsymbol{a}}$ represent the canonical basis vector whose $s$-th and $\boldsymbol{a}$-th element is only one and others are all zero in $\mathbb{R}^{|\mathcal{S}|}$ and $\mathbb{R}^{|\mathcal{A}|}$, respectively, and $\otimes$ denotes the Kronecker product. We can prove that $\boldsymbol{P}\boldsymbol{\Pi}^{\boldsymbol{Q}}$ for $\boldsymbol{Q} \in \mathbb{R}^{|\mathcal{S}||\mathcal{A}|}$ represents a transition probability of state-action pairs under policy $\pi$, i.e., $(\boldsymbol{e}_{s'} \otimes \boldsymbol{e}_{\boldsymbol{a}'})^\top(\boldsymbol{P}\boldsymbol{\Pi}^{\boldsymbol{Q}})(\boldsymbol{e}_s \otimes \boldsymbol{e}_{\boldsymbol{a}}) = \mathbb{P}\left[(s_{k+1}, \boldsymbol{a}_{k+1}) = (s', \boldsymbol{a}') \mid (s_k, \boldsymbol{a}_k) = (s, \boldsymbol{a}), \pi_{\boldsymbol{Q}}\right]$ for $s, s' \in \mathcal{S}$ and $\boldsymbol{a}, \boldsymbol{a}' \in \mathcal{A}$. Now, we can rewrite the Bellman equation in (1) using the matrix notations as follows: $\boldsymbol{R}^{\text{avg}} + \gamma \boldsymbol{P}\boldsymbol{\Pi}^{\boldsymbol{Q}^*}\boldsymbol{Q}^* = \boldsymbol{Q}^*$, where $\boldsymbol{R}^{\text{avg}} = \frac{1}{N}\sum_{i=1}^N \boldsymbol{R}^i \in \mathbb{R}^{|\mathcal{S}||\mathcal{A}|}$ and $\boldsymbol{Q}^* \in \mathbb{R}^{|\mathcal{S}||\mathcal{A}|}$ represents optimal Q-function, $\boldsymbol{Q}^*$, i.e., $(\boldsymbol{e}_s \otimes \boldsymbol{e}_{\boldsymbol{a}})^\top \boldsymbol{Q}^* = Q^*(s, \boldsymbol{a})$ for $s, \boldsymbol{a} \in \mathcal{S} \times \mathcal{A}$.

## 2.2 DISTRIBUTED Q-LEARNING

In this section, we discuss a distributed Q-learning algorithm motivated from Nedic and Ozdaglar (2009). The non-asymptotic behavior of the algorithm was first investigated

in Heredia et al. (2020); Zeng et al. (2022b) under linear function approximation scheme. Instead, we consider the tabular setup with mild assumptions, and detailed comparisons are given in Section 5. Each agent $i \in \mathcal{V}$ at time $k \in \mathbb{N}$ updates its estimate $\boldsymbol{Q}_k^i \in \mathbb{R}^{|\mathcal{S}||\mathcal{A}|}$ upon observing $s_k, \boldsymbol{a}_k, s_k' \in \mathcal{S} \times \mathcal{A} \times \mathcal{S}$ as follows:

$$
\begin{aligned}
\boldsymbol{Q}_{k+1}^i(s_k, \boldsymbol{a}_k) &= \sum_{j \in \mathcal{N}_i} [\boldsymbol{W}]_{ij} \boldsymbol{Q}_k^j(s_k, \boldsymbol{a}_k) + \alpha \left( r_{k+1}^i + \gamma \max_{\boldsymbol{a} \in \mathcal{A}} \boldsymbol{Q}_k^i(s_k', \boldsymbol{a}) - \boldsymbol{Q}_k^i(s_k, \boldsymbol{a}_k) \right) \\
\boldsymbol{Q}_{k+1}^i(s, \boldsymbol{a}) &= \sum_{j \in \mathcal{N}_i} [\boldsymbol{W}]_{ij} \boldsymbol{Q}_k^j(s, \boldsymbol{a}), \quad s, \boldsymbol{a} \in \mathcal{S} \times \mathcal{A} \setminus \{(s_k, \boldsymbol{a}_k)\},
\end{aligned}
\tag{2}
$$

where $\boldsymbol{Q}_k^i(s, \boldsymbol{a}) := (\boldsymbol{e}_s \otimes \boldsymbol{e}_{\boldsymbol{a}})^\top \boldsymbol{Q}_k^i$ for $s, \boldsymbol{a} \in \mathcal{S} \times \mathcal{A}$, $\alpha \in (0, 1)$ is the steps-size, and $\boldsymbol{W} \in \mathbb{R}^{N \times N}$ is a non-negative matrix such that agent $i$ assigns a weight $[\boldsymbol{W}]_{ij}$ to its neighbour $j \in \mathcal{N}_i$. The agent $i \in \mathcal{V}$ sends its estimate $\boldsymbol{Q}_k^i$ to its neighbour $j \in \mathcal{N}_i$, and receives $\boldsymbol{Q}_k^j$, which is weighted by $[\boldsymbol{W}]_{ij}$. The update is different from that of distributed optimization over an objective function in sense that (2) does not use any gradient of a function. Furthermore, note that the memory space of each agent can be expensive due to exponential scaling in the action space, but one can choose linear or neural network approximation (Zhang et al., 2018b; Sunehag et al., 2017) to overcome such issue.

To ensure the consensus among the agents, i.e., $\boldsymbol{Q}_k^i \to \boldsymbol{Q}^*$ for all $i \in [N]$, where $[N] := \{1, 2, \ldots, N\}$, a commonly adopted condition on $\boldsymbol{W}$ is the so-called doubly stochastic matrix:

**Assumption 2.1.** *For all $i \in [N]$, $[\boldsymbol{W}]_{ii} > 0$ and $[\boldsymbol{W}]_{ij} > 0$ if $(i, j) \in \mathcal{E}$, otherwise $[\boldsymbol{W}]_{ij} = 0$. Furthermore, $\sum_{j=1}^N [\boldsymbol{W}]_{ij} = \sum_{i=1}^N [\boldsymbol{W}]_{ji} = 1$, and $\boldsymbol{W}^\top = \boldsymbol{W}$.*

The assumption is widely adopted in the literature of distributed learning scheme (Heredia et al., 2020; Zeng et al., 2022b). In Appendix B, we provided a simple strategy to construct the doubly stochastic matrix by communicating only with its neighbour.

### 2.3 Switched system

In this paper, we consider a system, called the *switched affine system* (Liberzon, 2005),

$$
\boldsymbol{x}_{k+1} = \boldsymbol{A}_{\sigma_k} \boldsymbol{x}_k + \boldsymbol{b}_{\sigma_k}, \quad \boldsymbol{x}_0 \in \mathbb{R}^n, \quad k \in \mathbb{N},
\tag{3}
$$

where $\boldsymbol{x}_k \in \mathbb{R}^n$ is the state, $\mathcal{M} := \{1, 2, \ldots, M\}$ is called the set of modes, $\sigma_k \in \mathcal{M}$ is called the switching signal, $\{\boldsymbol{A}_\sigma \in \mathbb{R}^{n \times n} \mid \sigma \in \mathcal{M}\}$ and $\{\boldsymbol{b}_\sigma \in \mathbb{R}^n \mid \sigma \in \mathcal{M}\}$ are called the subsystem matrices, and the set of affine terms, respectively. The switching signal can be either arbitrary or controlled by the user under a certain switching policy. If the system in (3) evolves without the affine term, i.e., $\boldsymbol{b}_{\sigma_k} = \boldsymbol{0}$ for $k \in \mathbb{N}$, then it is called the switched linear system. The distributed Q-learning algorithm in (2) will be modeled as a switched affine system motivated from the recent connection of switched system and Q-learning (Lee and He, 2020), which will become clearer in Section 3.4

## 3 Error Analysis : i.i.d. observation model

In this section, we first consider i.i.d. observation model, which provides simple and clear intuitive results. In the subsequent section, we will extend the result to the Markovian observation model. By an i.i.d. observation model, we refer to a sequence of trajectory $\{(s_k, \boldsymbol{a}_k, s_k')\}_{k \geq 0}$ where each $(s_k, \boldsymbol{a}_k, s_k')$ are an i.i.d. random variables. Suppose that each state-action pair is sampled from a distribution $d \in \Delta^{|\mathcal{S} \times \mathcal{A}|}$, i.e., $\mathbb{P}[(s_k, \boldsymbol{a}_k) = (s, \boldsymbol{a})] = d(s, \boldsymbol{a})$ and $s_k' \sim \mathcal{P}(s_k, \boldsymbol{a}_k, \cdot)$. The pseudo-code of the algorithm is given in Algorithm 1 in the Appendix J. We will adopt the following standard assumption in the literature:

**Assumption 3.1.** *For all $s, \boldsymbol{a} \in \mathcal{S} \times \mathcal{A}$, we have $d(s, \boldsymbol{a}) > 0$.*

### 3.1 Matrix notations

Let us introduce the following vector and matrix notations used throughout the paper to re-write (2) in matrix notations: $\boldsymbol{D}_s := \mathrm{diag}(d(s, 1), \cdots, d(s, |\mathcal{A}|)) \in \mathbb{R}^{|\mathcal{A}| \times |\mathcal{A}|}, \quad \boldsymbol{D} =$

$\operatorname{diag}(\boldsymbol{D}_1, \boldsymbol{D}_2, \ldots, \boldsymbol{D}_{|\mathcal{S}|}) \in \mathbb{R}^{|\mathcal{S}||\mathcal{A}| \times |\mathcal{S}||\mathcal{A}|}$, where $\operatorname{diag}(\cdot)$ is a diagonal matrix whose diagonal elements correspond to the input vector or matrix, and we will denote $d_{\max} = \max_{s,\boldsymbol{a} \in \mathcal{S} \times \mathcal{A}} d(s, \boldsymbol{a})$ and $d_{\min} := \min_{s,\boldsymbol{a} \in \mathcal{S} \times \mathcal{A}} d(s, \boldsymbol{a})$. Furthermore, for $i \in [N]$, $o = (s, \boldsymbol{a}, s') \in \mathcal{S} \times \mathcal{A} \times \mathcal{S}$ and $\boldsymbol{Q} \in \mathbb{R}^{|\mathcal{S}||\mathcal{A}|}$, we define

$$\boldsymbol{\delta}^i(o, \boldsymbol{Q}) := (\boldsymbol{e}_s \otimes \boldsymbol{e}_{\boldsymbol{a}})(r^i(s, \boldsymbol{a}, s') + \boldsymbol{e}_{s'}^\top \gamma \boldsymbol{\Pi}^{\boldsymbol{Q}} \boldsymbol{Q} - (\boldsymbol{e}_s \otimes \boldsymbol{e}_{\boldsymbol{a}})^\top \boldsymbol{Q}),$$
$$\boldsymbol{\Delta}^i(\boldsymbol{Q}) := \boldsymbol{D}(\boldsymbol{R}^i + \gamma \boldsymbol{P} \boldsymbol{\Pi}^{\boldsymbol{Q}} \boldsymbol{Q} - \boldsymbol{Q}),$$

which denotes the TD-error and expected TD-error in vector representation. For simplicity of the notation, we denote $\boldsymbol{\delta}_k^i := \boldsymbol{\delta}^i(o_k, \boldsymbol{Q}_k^i)$, $\boldsymbol{\Delta}_k^i := \boldsymbol{\Delta}^i(\boldsymbol{Q}_k^i)$, and

$$\bar{\boldsymbol{Q}}_k := \begin{bmatrix} \boldsymbol{Q}_k^1 \\ \boldsymbol{Q}_k^2 \\ \vdots \\ \boldsymbol{Q}_k^N \end{bmatrix}, \quad \bar{\boldsymbol{\Pi}}^{\bar{\boldsymbol{Q}}_k} := \begin{bmatrix} \boldsymbol{\Pi}^{\boldsymbol{Q}_k^1} & & \\ & \ddots & \\ & & \boldsymbol{\Pi}^{\boldsymbol{Q}_k^N} \end{bmatrix}, \quad \bar{\boldsymbol{\epsilon}}_k(o_k, \bar{\boldsymbol{Q}}_k) := \begin{bmatrix} \boldsymbol{\delta}^1(o_k, \boldsymbol{Q}_k^1) - \boldsymbol{\Delta}^1(\boldsymbol{Q}_k^1) \\ \boldsymbol{\delta}^2(o_k, \boldsymbol{Q}_k^2) - \boldsymbol{\Delta}^2(\boldsymbol{Q}_k^2) \\ \vdots \\ \boldsymbol{\delta}^N(o_k, \boldsymbol{Q}_k^N) - \boldsymbol{\Delta}^N(\boldsymbol{Q}_k^N) \end{bmatrix},$$

$$\bar{\boldsymbol{P}} := \boldsymbol{I}_N \otimes \boldsymbol{P}, \quad \bar{\boldsymbol{D}} := \boldsymbol{I}_N \otimes \boldsymbol{D}, \quad \bar{\boldsymbol{W}} := \boldsymbol{W} \otimes \boldsymbol{I}_{|\mathcal{S}||\mathcal{A}|}, \quad \bar{\boldsymbol{R}} := \begin{bmatrix} \boldsymbol{R}^1 & \boldsymbol{R}^2 & \cdots & \boldsymbol{R}^N \end{bmatrix}^\top,$$
(4)

where $\boldsymbol{I}_N$ is a $N \times N$ identity matrix, $\boldsymbol{Q}_k^i$ is defined in (2). Moreover, we denote $\bar{\boldsymbol{\epsilon}}_k := \bar{\boldsymbol{\epsilon}}_k(o_k, \bar{\boldsymbol{Q}}_k)$. With the above set of notations, we can re-write the update in (2) as follows:

$$\bar{\boldsymbol{Q}}_{k+1} = \bar{\boldsymbol{W}} \bar{\boldsymbol{Q}}_k + \alpha \bar{\boldsymbol{D}} \left( \bar{\boldsymbol{R}} + \gamma \bar{\boldsymbol{P}} \bar{\boldsymbol{\Pi}}^{\bar{\boldsymbol{Q}}_k} \bar{\boldsymbol{Q}}_k - \bar{\boldsymbol{Q}}_k \right) + \alpha \bar{\boldsymbol{\epsilon}}_k. \tag{5}$$

## 3.2 Distributed Q-learning : Error analysis

In this section, we provide a sketch of the proof to bound the error of distributed Q-learning. Let us first decompose the error $\bar{\boldsymbol{Q}}_k - \boldsymbol{1}_N \otimes \boldsymbol{Q}^*$ into consensus error and optimality error:

$$\bar{\boldsymbol{Q}}_k - \boldsymbol{1}_N \otimes \boldsymbol{Q}^* = \underbrace{\bar{\boldsymbol{Q}}_k - \boldsymbol{1}_N \otimes \left( \frac{1}{N} \sum_{i=1}^N \boldsymbol{Q}_k^i \right)}_{\text{Consensus Error}} + \underbrace{\boldsymbol{1}_N \otimes \left( \frac{1}{N} \sum_{i=1}^N \boldsymbol{Q}_k^i - \boldsymbol{Q}^* \right)}_{\text{Optimality Error}}, \tag{6}$$

where $\boldsymbol{1}_N$ is a $N$-dimensional vector whose elements are all one. The consensus error measures the difference of $\boldsymbol{Q}_k^i$ and the overall average, $\frac{1}{N} \sum_{i=1}^N \boldsymbol{Q}_k^i$. As the consensus error vanishes, we will have $\boldsymbol{Q}_k^1 = \boldsymbol{Q}_k^2 = \cdots = \boldsymbol{Q}_k^N$. Meanwhile, the optimality error denotes the difference between the true solution $\boldsymbol{Q}^*$ and the average, $\frac{1}{N} \sum_{k=1}^N \boldsymbol{Q}_k^i$. Together with the consensus error, as optimality error vanishes, we should have $\boldsymbol{Q}_k^i - \boldsymbol{Q}^* \to 0$ for all $i \in [N]$.

## 3.3 Analysis of Consensus Error

Now, we provide an error bound on the consensus error in (6). We will represent the consensus error as $\boldsymbol{\Theta} \bar{\boldsymbol{Q}}_k = \bar{\boldsymbol{Q}}_k - \boldsymbol{1}_N \otimes \boldsymbol{Q}_k^{\text{avg}}$ where $\boldsymbol{Q}_k^{\text{avg}} := \frac{1}{N} \sum_{i=1}^N \boldsymbol{Q}_k^i$ and $\boldsymbol{\Theta} := \boldsymbol{I}_{N|\mathcal{S}||\mathcal{A}|} - \frac{1}{N}(\boldsymbol{1}_N \boldsymbol{1}_N^\top) \otimes \boldsymbol{I}_{|\mathcal{S}||\mathcal{A}|}$. Let us first provide an important lemma that characterizes the convergence of the consensus error:

**Lemma 3.2.** *For $k \in \mathbb{N}$, we have $\left\| \bar{\boldsymbol{W}}^k \boldsymbol{\Theta} \right\|_2 \leq \sigma_2(\boldsymbol{W})^k$, where $\sigma_2(\boldsymbol{W})$ is the second largest singular value of $\boldsymbol{W}$, and it holds that $\sigma_2(\boldsymbol{W}) < 1$.*

The proof is given in Appendix D.1. Moving on, we show that $\bar{\boldsymbol{Q}}_k$ will be remain bounded, which will be useful throughout the paper:

**Lemma 3.3.** *For $k \in \mathbb{N}$, and $\alpha \leq \min_{i \in [N]}[\boldsymbol{W}]_{ii}$, we have : $\left\| \bar{\boldsymbol{Q}}_k \right\|_\infty \leq \frac{R_{\max}}{1-\gamma}$.*

The proof is given in Appendix D.2. The step-size depends on $\min_{i \in [N]}[\boldsymbol{W}]_{ii}$, which can be considered as a global information. However, considering the method in Example B.1 in Appendix, which requires only local information to construct $\boldsymbol{W}$, we have $\min_{i \in [N]}[\boldsymbol{W}]_{ii} \geq \frac{1}{2}$. Therefore, it should be enough to choose $\alpha \leq \frac{1}{2}$. Furthermore, the step-size in many

distributed RL algorithms (Zeng et al., 2022b; Wang et al., 2020; Doan et al., 2021; Sun et al., 2020) depend on $\sigma_2(\boldsymbol{W})$, which also can be viewed as a global information. Moreover, we can use an agent-specific step-size, i.e., each agent keeps its own step-size, $\alpha_i$. Then, we only require $\alpha_i < [\boldsymbol{W}]_{ii}$, which only uses local information.

Now, we are ready to analyze the behavior of $\boldsymbol{\Theta}\bar{\boldsymbol{Q}}_k$. Multiplying $\boldsymbol{\Theta}$ to (5), we get

$$\boldsymbol{\Theta}\bar{\boldsymbol{Q}}_{k+1} = \prod_{i=0}^{k} \bar{\boldsymbol{W}}^i \boldsymbol{\Theta}\bar{\boldsymbol{Q}}_0 + \alpha \sum_{j=0}^{k} \bar{\boldsymbol{W}}^{k-j}\boldsymbol{\Theta}\left(\bar{\boldsymbol{D}}\left(\bar{\boldsymbol{R}} + \gamma\bar{\boldsymbol{P}}\bar{\boldsymbol{\Pi}}^{\bar{\boldsymbol{Q}}_j}\bar{\boldsymbol{Q}}_j - \bar{\boldsymbol{Q}}_j\right) + \bar{\boldsymbol{\epsilon}}_j\right). \qquad (7)$$

The equality results from recursively expanding the terms. Now, we are ready to bound $\boldsymbol{\Theta}\bar{\boldsymbol{Q}}_{k+1}$ using the fact that $\left\|\bar{\boldsymbol{W}}^i\boldsymbol{\Theta}\right\|_2$ for $i \in \mathbb{N}$ will decay at a rate of $\sigma_2(\boldsymbol{W})$ from Lemma 3.2, and the boundedness of $\bar{\boldsymbol{Q}}_k$ in Lemma 3.3.

**Theorem 3.4.** *For $k \in \mathbb{N}$, and $\alpha \leq \min_{i \in [N]}[\boldsymbol{W}]_{ii}$, we have the following:*

$$\left\|\boldsymbol{\Theta}\bar{\boldsymbol{Q}}_{k+1}\right\|_\infty \leq \sigma_2(\boldsymbol{W})^{k+1}\left\|\boldsymbol{\Theta}\bar{\boldsymbol{Q}}_0\right\|_2 + \alpha\frac{8R_{\max}}{1-\gamma}\frac{\sqrt{N|\mathcal{S}||\mathcal{A}|}}{1-\sigma_2(\boldsymbol{W})}.$$

The proof is given in Appendix D.3. As we can expect, the convergence rate of the consensus error depends on the $\sigma_2(\boldsymbol{W})$ with a constant error bound proportional to $\alpha$. Furthermore, we note that the above result also holds for the Markovian observation model in Section 4.

### 3.4 ANALYSIS OF OPTIMALITY ERROR

Throughout this section, we analyze the error bound on the optimality error, $\boldsymbol{Q}_k^{\text{avg}} - \boldsymbol{Q}^*$. Multiplying $\frac{1}{N}(\mathbf{1}_N\mathbf{1}_N^\top)\otimes\boldsymbol{I}_{|\mathcal{S}||\mathcal{A}|}$ on (5), we can see that $\boldsymbol{Q}_k^{\text{avg}}$ evolves via the following update:

$$\boldsymbol{Q}_{k+1}^{\text{avg}} = \boldsymbol{Q}_k^{\text{avg}} + \alpha\boldsymbol{D}\left(\boldsymbol{R}^{\text{avg}} + \frac{\gamma}{N}\sum_{i=1}^{N}\boldsymbol{P}\boldsymbol{\Pi}^{\boldsymbol{Q}_k^i}\boldsymbol{Q}_k^i - \boldsymbol{Q}_k^{\text{avg}}\right) + \alpha\boldsymbol{\epsilon}^{\text{avg}}(o_k, \bar{\boldsymbol{Q}}_k), \qquad (8)$$

where $\boldsymbol{\epsilon}^{\text{avg}}(o, \bar{\boldsymbol{Q}}) := \frac{1}{N}(\mathbf{1}_N\mathbf{1}_N^\top)\otimes\boldsymbol{I}_{|\mathcal{S}||\mathcal{A}|}\bar{\boldsymbol{\epsilon}}(o, \bar{\boldsymbol{Q}})$ for $o \in \mathcal{S}\times\mathcal{A}\times\mathcal{S}$, $\bar{\boldsymbol{Q}} \in \mathbb{R}^{N|\mathcal{S}||\mathcal{A}|}$, and $\bar{\boldsymbol{\epsilon}}(\cdot)$ is defined in (4). We will denote $\boldsymbol{\epsilon}_k^{\text{avg}} := \boldsymbol{\epsilon}^{\text{avg}}(o_k, \bar{\boldsymbol{Q}}_k)$. The update of (8) resembles that of Q-learning update in the single agent case, i.e., $N = 1$, whose Q-function is $\boldsymbol{Q}_k^{\text{avg}}$. However, the difference with the update of single-agent case lies in the fact that we take average of the maximum of Q-function of each agent, i.e., the term $\frac{1}{N}\sum_{i=1}^{N}\boldsymbol{\Pi}^{\boldsymbol{Q}_k^i}\boldsymbol{Q}_k^i$ in (8), rather than the maximum of average of Q-function of each agents, .i.e., $\boldsymbol{\Pi}^{\boldsymbol{Q}_k^{\text{avg}}}\boldsymbol{Q}_k^{\text{avg}}$. This poses difficulty in the analysis since $\frac{1}{N}\sum_{i=1}^{N}\boldsymbol{\Pi}^{\boldsymbol{Q}_k^i}\boldsymbol{Q}_k^i$ cannot be represented in terms of $\boldsymbol{Q}_k^{\text{avg}}$. Consequently, it makes difficult to interpret it as switched affine system whose state-variable is $\boldsymbol{Q}_k^{\text{avg}}$, which is introduced in Section 2.3. To handle this issue, motivated from the approach in Kar et al. (2013), we introduce an additional error term $\frac{1}{N}\sum_{i=1}^{N}\boldsymbol{\Pi}^{\boldsymbol{Q}_k^i}\boldsymbol{Q}_k^i - \boldsymbol{\Pi}^{\boldsymbol{Q}_k^{\text{avg}}}\boldsymbol{Q}_k^{\text{avg}}$, which can be bounded by the consensus error discussed in Section 3.3. Therefore, we re-write (8) as:

$$\boldsymbol{Q}_{k+1}^{\text{avg}} = \boldsymbol{Q}_k^{\text{avg}} + \alpha\boldsymbol{D}\left(\boldsymbol{R}^{\text{avg}} + \gamma\boldsymbol{P}\boldsymbol{\Pi}^{\boldsymbol{Q}_k^{\text{avg}}}\boldsymbol{Q}_k^{\text{avg}} - \boldsymbol{Q}_k^{\text{avg}}\right) + \alpha\boldsymbol{\epsilon}_k^{\text{avg}}$$

$$+ \alpha\underbrace{\left(\frac{\gamma}{N}\sum_{i=1}^{N}\boldsymbol{D}\left(\boldsymbol{P}\boldsymbol{\Pi}^{\boldsymbol{Q}_k^i}\boldsymbol{Q}_k^i - \gamma\boldsymbol{P}\boldsymbol{\Pi}^{\boldsymbol{Q}_k^{\text{avg}}}\boldsymbol{Q}_k^{\text{avg}}\right)\right)}_{:=\boldsymbol{E}_k}. \qquad (9)$$

Now, we can see that $\boldsymbol{Q}_k^{\text{avg}}$ evolves via a single-agent Q-learning update whose estimator is $\boldsymbol{Q}_k^{\text{avg}}$, including an additional stochastic noise term, $\boldsymbol{\epsilon}_k^{\text{avg}}$, and an error term, $\boldsymbol{E}_k$ that can be bounded by the consensus error. In the following lemma, we use the contraction property of the max-operator to bound $\boldsymbol{E}_k$ by the consensus error:

**Lemma 3.5.** *For $k \in \mathbb{N}$, we have $\|\boldsymbol{E}_k\|_\infty \leq \gamma d_{\max}\left\|\boldsymbol{\Theta}\bar{\boldsymbol{Q}}_k\right\|_\infty$.*

The proof is given in Appendix D.4. We note that similar argument in Lemma 3.5 has been also considered in Kar et al. (2013). However, Kar et al. (2013) considered a different

distributed algorithm using two-time scale approach and focused on asymptotic convergence whereas we consider a single step-size and finite-time bounds.

Now, we follow the switched system approach (Lee and He, 2020) to bound the optimality error. In contrast to Lee and He (2020), we have an additional error term caused by $\boldsymbol{E}_k$, which will be bounded using Theorem 3.4. Using a coordinate transformation, $\tilde{\boldsymbol{Q}}_k^{\mathrm{avg}} = \boldsymbol{Q}_k^{\mathrm{avg}} - \boldsymbol{Q}^*$, we can re-write (9) as

$$\tilde{\boldsymbol{Q}}_{k+1}^{\mathrm{avg}} = \boldsymbol{A}_{\boldsymbol{Q}_k^{\mathrm{avg}}}\tilde{\boldsymbol{Q}}_k^{\mathrm{avg}} + \alpha\boldsymbol{b}_{\boldsymbol{Q}_k^{\mathrm{avg}}} + \alpha\boldsymbol{\epsilon}_k^{\mathrm{avg}} + \alpha\boldsymbol{E}_k,$$

where, for $\boldsymbol{Q} \in \mathbb{R}^{|\mathcal{S}||\mathcal{A}|}$, we let

$$\boldsymbol{A}_{\boldsymbol{Q}} := \boldsymbol{I} + \alpha\boldsymbol{D}(\gamma\boldsymbol{P}\boldsymbol{\Pi}^{\boldsymbol{Q}} - \boldsymbol{I}) \in \mathbb{R}^{|\mathcal{S}||\mathcal{A}|\times|\mathcal{S}||\mathcal{A}|}, \quad \boldsymbol{b}_{\boldsymbol{Q}} := \gamma\boldsymbol{D}\boldsymbol{P}(\boldsymbol{\Pi}^{\boldsymbol{Q}} - \boldsymbol{\Pi}^{\boldsymbol{Q}^*})\boldsymbol{Q}^*. \quad (10)$$

We can see that $\boldsymbol{\epsilon}_k^{\mathrm{avg}}$ is a stochastic term, and we will bound the error caused by this term using concentration inequalities. The consensus error, $\boldsymbol{E}_k$, can be bounded from Theorem 3.4. However, the affine term, $\boldsymbol{b}_{\boldsymbol{Q}_k^{\mathrm{avg}}}$, does not admit simple bounds. The approach in Lee and He (2020) provides a method to construct a system without an affine term, making the analysis simpler. In details, we introduce a lower and upper comparison system, denoted as $\boldsymbol{Q}_k^{\mathrm{avg},l}$ and $\boldsymbol{Q}_k^{\mathrm{avg},u}$, respectively such that the following element-wise inequaltiy holds:

$$\boldsymbol{Q}_k^{\mathrm{avg},l} \leq \boldsymbol{Q}_k^{\mathrm{avg}} \leq \boldsymbol{Q}_k^{\mathrm{avg},u}, \quad \forall k \in \mathbb{N}, \quad (11)$$

Letting $\tilde{\boldsymbol{Q}}_k^{\mathrm{avg},l} := \boldsymbol{Q}_k^{\mathrm{avg},l} - \boldsymbol{Q}^*$ and $\tilde{\boldsymbol{Q}}_k^{\mathrm{avg}.u} := \boldsymbol{Q}_k^{\mathrm{avg},u} - \boldsymbol{Q}^*$, a candidate of update that satisfies (11), which is without the affine term $\boldsymbol{b}_{\boldsymbol{Q}_k}$, is:

$$\tilde{\boldsymbol{Q}}_{k+1}^{\mathrm{avg},l} = \boldsymbol{A}_{\boldsymbol{Q}^*}\tilde{\boldsymbol{Q}}_k^{\mathrm{avg},l} + \alpha\boldsymbol{\epsilon}_k^{\mathrm{avg}} + \alpha\boldsymbol{E}_k, \quad \tilde{\boldsymbol{Q}}_{k+1}^{\mathrm{avg},u} = \boldsymbol{A}_{\boldsymbol{Q}_k^{\mathrm{avg},u}}\tilde{\boldsymbol{Q}}_k^{\mathrm{avg},u} + \alpha\boldsymbol{\epsilon}_k^{\mathrm{avg}} + \alpha\boldsymbol{E}_k, \quad (12)$$

where $\boldsymbol{Q}_0^{\mathrm{avg},l} \leq \boldsymbol{Q}_0^{\mathrm{avg}} \leq \boldsymbol{Q}_0^{\mathrm{avg},u}$. The detailed construction of each systems are given in Appendix E. Note that the lower comparison system, $\tilde{\boldsymbol{Q}}_k^{\mathrm{avg},l}$ follows a linear system governed by the matrix $\boldsymbol{A}_{\boldsymbol{Q}^*}$ where as the upper comparison system ,$\tilde{\boldsymbol{Q}}_k^{\mathrm{avg},u}$, can be viewed as a switched linear system without an affine term. To prove the finite-time bound of $\tilde{\boldsymbol{Q}}_k^{\mathrm{avg}}$, we will instead derive the finite-time bound of $\tilde{\boldsymbol{Q}}_k^{\mathrm{avg},l}$ and $\tilde{\boldsymbol{Q}}_k^{\mathrm{avg},u}$, and using the relation in (11), we can obtain the desired result. Nonetheless, still the switching in the upper comparison system imposes difficulty in the analysis. Therefore, we consider the difference of upper and lower comparison system $\tilde{\boldsymbol{Q}}_k^{\mathrm{avg},l} - \tilde{\boldsymbol{Q}}_k^{\mathrm{avg},u}$, which gives the following bound: $\left\|\tilde{\boldsymbol{Q}}_k^{\mathrm{avg}}\right\|_\infty \leq \left\|\tilde{\boldsymbol{Q}}_k^{\mathrm{avg},l}\right\|_\infty + \left\|\boldsymbol{Q}_{k+1}^{\mathrm{avg},u} - \boldsymbol{Q}_{k+1}^{\mathrm{avg},l}\right\|_\infty$. The sketch of the proof for deriving the finite-time bound of each systems are as follows:

1. Bounding $\tilde{\boldsymbol{Q}}_k^{\mathrm{avg},l}$ (Proposition F.1 in the Appendix): We recursively expand the equation in (12). We have $\|\boldsymbol{A}_{\boldsymbol{Q}}\|_\infty \leq 1 - (1-\gamma)\alpha d_{\min}$ for any $\boldsymbol{Q} \in \mathbb{R}^{|\mathcal{S}||\mathcal{A}|}$, which is in Lemma C.1 in the Appendix, and the error induced by $\boldsymbol{\epsilon}_k^{\mathrm{avg}}$ can be bounded using Azuma-Hoeffding inequality in Lemma C.4 in the Appendix. Meanwhile, the error term $\boldsymbol{E}_k$ can be bounded by the consensus error from Lemma 3.5, which is again bounded by using Theorem 3.4.

2. Bounding $\tilde{\boldsymbol{Q}}_k^{\mathrm{avg},u} - \tilde{\boldsymbol{Q}}_k^{\mathrm{avg},l}$ (Proposition F.3 in the Appendix): Thanks to the fact that both the upper an lower comparison systems share $\boldsymbol{\epsilon}_k^{\mathrm{avg}}$ and $\boldsymbol{E}_k$, if we subtract $\tilde{\boldsymbol{Q}}_k^{\mathrm{avg},l}$ from $\tilde{\boldsymbol{Q}}_k^{\mathrm{avg},u}$ in (12), both terms are eliminated. Therefore, the iterate can be bounded with an additional error by $\tilde{\boldsymbol{Q}}_k^{\mathrm{avg},l}$.

Now, we are ready to present the optimality error bound, $\|\boldsymbol{Q}_k^{\mathrm{avg}} - \boldsymbol{Q}^*\|_\infty$, as follows:

**Theorem 3.6.** *For $k \in \mathbb{N}$, and $\alpha \leq \min_{i\in[N]}[\boldsymbol{W}]_{ii}$, we have the following result :*

$$\mathbb{E}\left[\|\boldsymbol{Q}_k^{\mathrm{avg}} - \boldsymbol{Q}^*\|_\infty\right] = \tilde{\mathcal{O}}\left((1 - \alpha(1-\gamma)d_{\min})^{\frac{k}{2}} + \sigma_2(\boldsymbol{W})^{\frac{k}{4}}\right)$$

$$+ \tilde{\mathcal{O}}\left(\alpha^{\frac{1}{2}}\frac{d_{\max}R_{\max}}{(1-\gamma)^{\frac{5}{2}}d_{\min}^{\frac{3}{2}}} + \alpha\frac{d_{\max}^2\sqrt{|\mathcal{S}||\mathcal{A}|}R_{\max}}{(1-\gamma)^3d_{\min}^2(1-\sigma_2(\boldsymbol{W}))}\right),$$

*where the notation $\tilde{\mathcal{O}}(\cdot)$ is used to hide the logarithmic factors.*

The proof is given in Appendix F.1. Note that even the logarithmic terms are hidden, due to exponential scaling of the action space, $\ln(|\mathcal{S}||\mathcal{A}|)$ could contribute $\mathcal{O}(N)$ factor to the error bound. However, noting that $d_{\min} \leq \frac{1}{|\mathcal{S}||\mathcal{A}|}$, $\mathcal{O}\left(\frac{1}{d_{\min}}\right)$ already dominates the $\mathcal{O}(N)$ if $|\mathcal{A}_i| \geq 2$ for all $i \in [N]$, hence we omit the logarithmic terms. Likewise $\mathcal{O}(|\mathcal{A}|)$ dominates $\mathcal{O}(N)$, which is hided when both terms are multiplied.

## 3.5 Final error

In this section, we present the error bound of the total error term $\bar{\boldsymbol{Q}}_k - \boldsymbol{1}_N \otimes \boldsymbol{Q}^*$. From (6), the bound follows from the decomposition into the consensus error and optimality error. In particular, collecting the results in Theorem 3.4 and Theorem 3.6 yields the following:

**Theorem 3.7.** *For $k \in \mathbb{N}$, and $\alpha \leq \min_{i \in [N]}[\boldsymbol{W}]_{ii}$, we have*

$$\mathbb{E}\left[\left\|\bar{\boldsymbol{Q}}_k - \boldsymbol{1}_N \otimes \boldsymbol{Q}^*\right\|_\infty\right] = \tilde{\mathcal{O}}\left((1 - \alpha(1-\gamma)d_{\min})^{\frac{k}{2}} + \sigma_2(\boldsymbol{W})^{\frac{k}{4}}\right)$$
$$+ \tilde{\mathcal{O}}\left(\alpha^{\frac{1}{2}}d_{\max}\frac{R_{\max}}{(1-\gamma)^{\frac{5}{2}}d_{\min}^{\frac{3}{2}}} + \alpha\frac{d_{\max}^2\sqrt{|\mathcal{S}||\mathcal{A}|}R_{\max}}{(1-\gamma)^3 d_{\min}^2(1-\sigma_2(\boldsymbol{W}))}\right).$$

The proof is given in Appendix F.2. One can see that the convergence rate has exponentially decaying terms, $(1 - (1-\gamma)d_{\min}\alpha)^{\frac{k}{2}}$ and $\sigma_2(\boldsymbol{W})^{\frac{k}{4}}$, with a bias term caused by using a constant step-size. Furthermore, we note that the bias term depends on $\frac{1}{1-\sigma_2(\boldsymbol{W})}$. If we construct $\boldsymbol{W}$ as in Example B.1 in the Appendix, then it will contribute $O(N^2)$ factor in the error bound (Olshevsky, 2014).

**Corollary 3.8.** *Suppose $\alpha = \tilde{\mathcal{O}}\left(\min\left\{\frac{(1-\gamma)^5 d_{\min}^3}{R_{\max}^2 d_{\max}^2}\epsilon^2, \frac{(1-\gamma)^3 d_{\min}^2(1-\sigma_2(\boldsymbol{W}))}{R_{\max}d_{\max}^2\sqrt{|\mathcal{S}||\mathcal{A}|}}\epsilon\right\}\right)$. Then, the following number of samples are required for $\mathbb{E}\left[\left\|\bar{\boldsymbol{Q}}_k - \boldsymbol{1}_N \otimes \boldsymbol{Q}^*\right\|_\infty\right] \leq \epsilon$:*

$$\tilde{\mathcal{O}}\left(\max\left\{\frac{1}{\epsilon^2}\frac{d_{\max}^2}{(1-\gamma)^6 d_{\min}^4}, \frac{1}{\epsilon}\frac{d_{\max}^2\sqrt{|\mathcal{S}||\mathcal{A}|}}{(1-\gamma)^4 d_{\min}^3(1-\sigma_2(\boldsymbol{W}))}\right\}\right).$$

The proof is given in Appendix Section F.3. As the known sample complexity of (single-agent) Q-learning, our bound depends on the factors, $d_{\min}$ and $\frac{1}{1-\gamma}$. The result is improvabale in sense that the known tight dependency for single-agent case is $\frac{1}{(1-\gamma)^4 d_{\min}}$ by Li et al. (2020). Furthermore, we note that the dependency on the spectral property of the graph, $\frac{1}{\epsilon}\frac{1}{1-\sigma_2(\boldsymbol{W})}$ is common in the literature of distributed learning as can be seen in Table 1.

## 4 Error Analysis : Markovian observation model

Now, we consider a Markovian observation model instead of the i.i.d. model. Starting from an initial distribution $\boldsymbol{\mu}_0 \in \Delta^{|\mathcal{S}||\mathcal{A}|}$, the samples are observed from a behavior policy $\beta : \mathcal{S} \to \Delta^{|\mathcal{A}|}$, i.e., from $(s_k, \boldsymbol{a}_k)$, transition occurs to $s_{k+1} \sim \mathcal{P}(s_k, \boldsymbol{a}_k, \cdot)$ and the action is selected by $\boldsymbol{a}_{k+1} \sim \beta(\cdot \mid s_{k+1})$. This setting is closer to practical scenarios, but poses significant challenges in the analysis due to the dependence between the past observations and current estimates. To overcome this difficulty, we consider the so-called uniformly ergodic Markov chain (Paulin, 2015), which ensures that the Markov chain converges to its unique stationary distribution, $\boldsymbol{\mu}_\infty \in \Delta^{|\mathcal{S}||\mathcal{A}|}$, exponentially fast in sense of total variation distance, which is defined as $d_{\text{TV}}(\boldsymbol{p}, \boldsymbol{q}) := \frac{1}{2}\sum_{x \in \mathcal{S} \times \mathcal{A}}|[\boldsymbol{p}]_x - [\boldsymbol{q}]_x|$ where $\boldsymbol{p}, \boldsymbol{q} \in \Delta^{|\mathcal{S}||\mathcal{A}|}$. That is, there exist positive real numbers $m \in \mathbb{R}$ and $\rho \in (0, 1)$ such that we have $\max_{s, \boldsymbol{a} \in \mathcal{S} \times \mathcal{A}} d_{\text{TV}}(\boldsymbol{\mu}_k^{s, \boldsymbol{a}}, \boldsymbol{\mu}_\infty) \leq m\rho^k$, where $\boldsymbol{\mu}_k^{s, \boldsymbol{a}} := ((\boldsymbol{e}_s \otimes \boldsymbol{e}_{\boldsymbol{a}})^\top \boldsymbol{P}_\beta^k)^\top$ is the probability distribution of state-action pair after $k$ number of transition occurs starting from $s, \boldsymbol{a} \in \mathcal{S} \times \mathcal{A}$, and $\boldsymbol{P}_\beta \in \mathbb{R}^{|\mathcal{S}||\mathcal{A}| \times |\mathcal{S}||\mathcal{A}|}$ is the transition matrix induced by behavior policy $\beta$, i.e., $(\boldsymbol{e}_s \otimes \boldsymbol{e}_{\boldsymbol{a}})^\top \boldsymbol{P}_\beta(\boldsymbol{e}_{s'} \otimes \boldsymbol{e}_{\boldsymbol{a}'})^\top = (\boldsymbol{e}_s \otimes \boldsymbol{e}_{\boldsymbol{a}})^\top \boldsymbol{P}\boldsymbol{e}_{s'} \cdot \beta(\boldsymbol{a}' \mid s')$. Moreover, we will denote

$$\tau^{\text{mix}}(\epsilon) := \min\{t \in \mathbb{N} : m\rho^t \leq \epsilon\}, \quad \tau := \tau^{\text{mix}}(\alpha), \quad t_{\text{mix}} := \tau^{\text{mix}}(1/4), \quad (13)$$

for $\epsilon > 0$, and $\tau$ is the so-called mixing time. The concept of mixing time is widely used in the literature (Zeng et al., 2022b; Bhandari et al., 2018). Note that $\tau$ is approximately proportional to $\log\left(\frac{1}{\alpha}\right)$, which is provided in Lemma C.7 in the Appendix. This contributes only logarithmic factor to the final error bound. Furthermore, we will denote

$$\boldsymbol{D}_\infty = \text{diag}(\boldsymbol{\mu}_\infty), \quad \boldsymbol{D}_k^{s,\boldsymbol{a}} = \text{diag}(\boldsymbol{\mu}_k^{s,\boldsymbol{a}}), \tag{14}$$

where $\boldsymbol{D}_k^{s,\boldsymbol{a}}$ denotes the probability distribution of the state-action pair after $k$ number of transitions from $s, \boldsymbol{a} \in \mathcal{S} \times \mathcal{A}$. $\bar{\epsilon}_k$ in (5) will be defined in terms of $\boldsymbol{D}_\infty$ instead of $\boldsymbol{D}$, and the overall details are provided in Appendix G. To proceed, with slight abuse of notation, we will denote $d_{\max} = \max_{s,\boldsymbol{a} \in \mathcal{S} \times \mathcal{A}}[\boldsymbol{\mu}_\infty]_{s,\boldsymbol{a}}$ and $d_{\min} = \min_{s,\boldsymbol{a} \in \mathcal{S} \times \mathcal{A}}[\boldsymbol{\mu}_\infty]_{s,\boldsymbol{a}}$.

Now, we provide the technical difference with the proof of i.i.d. case in Section 3. The challenge in the analysis lies in the fact that $\mathbb{E}\left[\boldsymbol{\epsilon}_k^{\text{avg}} \middle| \{(s_t, \boldsymbol{a}_t)\}_{t=0}^k, \bar{\boldsymbol{Q}}_0\right] \neq \boldsymbol{0}$ due to Markovian observation scheme. Therefore, we cannot use Azuma-Hoeffding inequality as in the proof of i.i.d. case in the Appendix F.1. Instead, we consider the shifted sequence as in Qu and Wierman (2020). By shifted sequence, it means to consider the error by the stochastic observation at $k$ with $\bar{\boldsymbol{Q}}_{k-\tau}$ instead of $\bar{\boldsymbol{Q}}_k$, i.e., $\boldsymbol{w}_{k,1} := \boldsymbol{\delta}^{\text{avg}}(o_k, \bar{\boldsymbol{Q}}_{k-\tau}) - \boldsymbol{\Delta}_{k-\tau,k}^{\text{avg}}(\bar{\boldsymbol{Q}}_{k-\tau})$ where $\boldsymbol{\Delta}_{\tau,k}^{\text{avg}}(\bar{\boldsymbol{Q}}_k) := \boldsymbol{D}_\tau^{s_{k-\tau},\boldsymbol{a}_{k-\tau}} \frac{1}{N} \sum_{i=1}^N \left(\boldsymbol{R}^i + \gamma \boldsymbol{P} \boldsymbol{\Pi}^{\boldsymbol{Q}_k^i} \boldsymbol{Q}_k^i - \boldsymbol{Q}_k^i\right)$. Then, we have $\mathbb{E}\left[\boldsymbol{w}_{k,1} \middle| \{(s_t, \boldsymbol{a}_t)\}_{t=0}^{k-\tau}, \bar{\boldsymbol{Q}}_0\right] = \boldsymbol{0}$. Now, we separately calculate the errors induced by $\{\boldsymbol{w}_{\tau j+l,1}\}_{j \in \{t \in \mathbb{N} | \tau t+l \leq k\}}$ for each $0 \leq l \leq \tau - 1$, and invoke the Azuma-Hoeffding inequality. Overall details are given in Appendix G, and we have the following result:

**Theorem 4.1.** *For $k \geq \tau$, and $\alpha \leq \min\left\{\min_{i \in [N]}[\boldsymbol{W}]_{ii}, \frac{1}{2\tau}\right\}$, we have*

$$\mathbb{E}\left[\|\boldsymbol{Q}_{k+1} - \boldsymbol{Q}^*\|_\infty\right] = \tilde{\mathcal{O}}\left((1 - \alpha(1-\gamma)d_{\min})^{\frac{k-\tau}{2}} + \sigma_2(\boldsymbol{W})^{\frac{k-\tau}{4}}\right)$$

$$+ \tilde{\mathcal{O}}\left(\alpha^{\frac{1}{2}} \frac{d_{\max}\sqrt{\tau}R_{\max}}{(1-\gamma)^{\frac{5}{2}}d_{\min}^{\frac{3}{2}}} + \alpha \frac{R_{\max}d_{\max}\sqrt{|\mathcal{S}||\mathcal{A}|}}{(1-\gamma)^3 d_{\min}^2(1-\sigma_2(\boldsymbol{W}))}\right).$$

The proof is given in Appendix Section G.2.

**Corollary 4.2.** *Suppose $\alpha = \tilde{\mathcal{O}}\left(\frac{\epsilon^2}{\ln\left(\frac{1}{\epsilon^2}\right)} \frac{(1-\gamma)^5 d_{\min}^3}{t_{\text{mix}} d_{\max}^2}\right)$. Then, the following number of samples are required for $\mathbb{E}\left[\|\bar{\boldsymbol{Q}}_k - \boldsymbol{1}_N \otimes \boldsymbol{Q}^*\|_\infty\right] \leq \epsilon$:*

$$\tilde{\mathcal{O}}\left(\max\left\{\frac{\ln^2\left(\frac{1}{\epsilon^2}\right)}{\epsilon^2} \frac{t_{\text{mix}} d_{\max}^2}{(1-\gamma)^6 d_{\min}^4}, \frac{\ln\left(\frac{1}{\epsilon}\right)}{\epsilon} \frac{d_{\max}\sqrt{|\mathcal{S}||\mathcal{A}|}}{(1-\gamma)^4 d_{\min}^3(1-\sigma_2(\boldsymbol{W}))}\right\}\right).$$

The proof is given in Appendix Section G.3. As in the result of i.i.d. case in Corollary 3.8, we have the dependency on $\frac{1}{1-\gamma}, \frac{1}{d_{\min}}$, and $\frac{1}{1-\sigma_2(\boldsymbol{W})}$ with additional factor on mixing time. The known tight sample complexity result in the single-agent case is $\tilde{\mathcal{O}}\left(\frac{1}{(1-\gamma)^4 d_{\min}\epsilon^2} + \frac{t_{\text{mix}}}{(1-\gamma)d_{\min}}\right)$ by Li et al. (2024), and our result leaves room for improvement. Assuming a uniform sampling scheme, i.e., $d_{\min} = d_{\max} = \frac{1}{|\mathcal{S}||\mathcal{A}|}$, and $|\mathcal{A}_i| = A$ for all $i \in [N]$ and $A \geq 2$, the sample complexity becomes $\tilde{\mathcal{O}}\left(\max\left\{\frac{t_{\text{mix}}}{\epsilon^2} \frac{|\mathcal{S}|^2 A^{2N}}{(1-\gamma)^6}, \frac{1}{\epsilon} \frac{|\mathcal{S}|^{\frac{5}{2}} A^{\frac{5N}{2}}}{(1-\gamma)^4(1-\sigma_2(\boldsymbol{W}))}\right\}\right)$. We note that the exponential scaling in the action space is inevitable in the tabular setting unless we consider a near-optimal solution (Qu et al., 2022). Lastly, to verify the convergence of our algorithm, experiments are provided in Appendix Section I.

## 5 DISCUSSION

| | Q-function | Assumption | Sample complexity | Bound type | Remarks |
|---|---|---|---|---|---|
| Ours | Tabular | ✗ | $\max\left\{\frac{t_{\text{mix}}}{\epsilon^2} \frac{1}{(1-\gamma)^6 d_{\min}^3}, \frac{1}{\epsilon} \frac{\sqrt{|\mathcal{S}||\mathcal{A}|}}{(1-\sigma_2(\boldsymbol{W}))(1-\gamma)^4 d_{\min}^3}\right\}$ | Expectation | - |
| Wang et al. (2022) | Tabular | ✗ | $\frac{1}{(1-\gamma)^5 d_{\min}\epsilon^2} + \frac{t_{\text{mix}}}{1-\gamma}$ | High probability | $\epsilon \in \left[0, \frac{1}{1-\gamma}\right)$ |
| Heredia et al. (2020) | LFA | (15) | $\frac{R^2}{(d_{\min} - \gamma^2 d_{\max}^2)^2(1-\sigma_2(\boldsymbol{W}))}$ | Expectation Averaged squared error | Continuous state space $R$ is projection radius |
| Zeng et al. (2022b) | LFA | (16) | $\frac{1}{\kappa^2(1-\gamma)^2(1-\sigma_2(\boldsymbol{W}))}$ | Expectation | - |

Table 1: LFA stands for linear function approximation.

In this section, we provide comparison with recent works analyzing non-asymptotic behavior of distributed Q-learning algorithm. Our analysis relies on the minimal assumption in sense that we do not require any assumption further than standard assumptions in the literature, e.g., the state-action distribution induced by the behavior policy, is positive for all state-action pairs in Assumption 3.1.

Heredia et al. (2020) considered linear function approximation scheme to represent the Q-function with continuous state-space and finite-action space scenario. However, to prove the convergence, it requires the following condition:

$$d_{\min} > \gamma^2 d_{\max}^* := \max_s d(s, \pi^*(s)), \tag{15}$$

which is difficult to be met even in the tabular case, and an example is given in Appendix H.

Furthermore, Zeng et al. (2022b) considered a Q-learning model under linear function approximation with continuous-state space and finite action space. The work also covered the case when the features for linear function approximation is differently selected for each agents. However, it requires the following condition to hold for all $\boldsymbol{Q} \in \mathbb{R}^{|\mathcal{S}||\mathcal{A}|}$:

$$(\gamma \boldsymbol{DP}(\boldsymbol{\Pi^Q Q} - \boldsymbol{\Pi^{Q^*} Q^*}) - \boldsymbol{D}(\boldsymbol{Q} - \boldsymbol{Q^*}))^\top (\boldsymbol{Q} - \boldsymbol{Q^*}) \leq -\kappa \|\boldsymbol{Q} - \boldsymbol{Q^*}\|_2^2, \tag{16}$$

for some $\kappa > 0$. We have provided examples where the above conditions in (15) and (16) are not met even in the tabular case in Appendix Section H.

Overall, the assumptions used in Heredia et al. (2020); Zeng et al. (2022b) allows the analysis to follow similar lines to that of convex optimization literature. To the best of our knowledge, there is no existing literature that demonstrates how to extend convex optimization analysis, or an analogous approach, to the analysis of Q-learning under the tabular setup. This gap in the literature makes the analysis challenging and is the primary reason we rely on switched system analysis. Due to different settings, their sample complexity is not directly comparable with ours.

Wang et al. (2022) proposed a distributed Q-learning algorithm in the tabular setting, which is motivated from the adapt-then-combine algorithm, whereas our algorithm considers combine-and-adapt scheme (Chen and Sayed, 2012) in the distributed optimization literature. The work presents a sharper bound on the sample complexity $\frac{1}{(1-\gamma)^5 d_{\min} \epsilon^2}$ compared to ours $\frac{1}{(1-\gamma)^6 d_{\min}^4 \epsilon^2}$ but it only holds for restricted range of $\epsilon$, i.e., $\epsilon \in \left[0, \frac{1}{1-\gamma}\right)$ while our results do not have such restriction. More importantly, the algorithm proposed by Wang et al. (2022) requires two steps for a single update, whereas in our paper, we focus on a one-step algorithm that is algorithmically simpler and more efficient. Specifically, we analyze the traditional and widely adopted QD-learning algorithm proposed in Kar et al. (2013), for which a finite-time error analysis for the original form has been lacking in the literature. Additionally, we enhance the efficiency of QD-learning by employing a constant step-size, as opposed to the two-time-scale decaying step-size used in traditional QD-learning. This modification can significantly improve the convergence speed empirically.

## 6 CONCLUSION

In this paper, we have studied distributed version of Q-learning algorithm. We provided a sample complexity result of $\tilde{\mathcal{O}}\left(\max\left\{\frac{1}{\epsilon^2} \frac{1}{(1-\gamma)^6 d_{\min}^4}, \frac{1}{\epsilon} \frac{\sqrt{|\mathcal{S}||\mathcal{A}|}}{(1-\sigma_2(\boldsymbol{W}))(1-\gamma)^4 d_{\min}^3}\right\}\right)$, which appears to be the first non-asymptotic result for tabular Q-learning. Future work would include improving the dependency on $\frac{1}{1-\gamma}$ and $d_{\min}$ to match the known tightest sample complexity bound of single-agent Q-learning (Li et al., 2020). Furthermore, to resolve the scalability issue, two promising approaches would be adopting a mean-field approach or exploring convergence to sub-optimal point.

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

## A   APPENDIX : NOTATIONS

$\mathbb{R}^n$: set of real-valued $n$-dimensional vectors; $\mathbb{R}^{n \times m}$ : set of real-valued $n \times m$-dimensional matrices; $\Delta^n$ for $n \in \mathbb{N}$ : a probability simplex in $\mathbb{R}^n$; $[n]$ for $n \in \mathbb{N}$ : $\{1, 2, \ldots, n\}$; $\mathbf{1}_n$ : $n$-dimensional vector whose elements are all one; $\mathbf{0}$ : a vector whose elements are all zero with appropriate dimension; $[\boldsymbol{A}]_{ij}$ : $i$-th row and $j$-th column for any matrix $\boldsymbol{A}$; $\boldsymbol{e}_j$ : basis vector (with appropriate dimension) whose $j$-th element is one and others are all zero; $|\mathcal{S}|$ : cardinality of any finite set $\mathcal{S}$; $\otimes$ : Kronecker product between two matrices; $\boldsymbol{a} \geq \boldsymbol{b}$ for $\boldsymbol{a}, \boldsymbol{b} \in \mathbb{R}^n$ : $[\boldsymbol{a}]_i \geq [\boldsymbol{b}]_i$ for all $i \in [n]$.

## B   APPENDIX : CONSTRUCTING DOUBLY STOCHASTIC MATRIX

**Example B.1** (Lazy Metropolis matrix in Olshevsky (2014))**.** *To construct the doubly stochastic matrix $\boldsymbol{W}$ with only local information, we can set $[\boldsymbol{W}]_{ij} = \frac{1}{2 \max\{|\mathcal{N}_i|, |\mathcal{N}_j|\}}$ for $i \neq j$ and $i, j \in [N]$, letting $[\boldsymbol{W}]_{ii} = 1 - \sum_{j \in \mathcal{N}_i}[\boldsymbol{W}]_{ij}$. This uses only local information, and does not require any global information sharing.*

One can formulate a semi-definite program to construct a doubly stochastic matrix (Xiao and Boyd, 2004). It finds the doubly stochastic matrix with minimum possible $\sigma_2(W)$ but it requires a centralized controller to solve such system, and distributed the computed the result of each agents. Another choice is to use Sinkhorn-Knopp algorithm (Knight, 2008). However, it also requires a centralized computation scheme. Moreover, to our best knowledge, we are not aware of bound on the $\sigma_2(W)$ of the output of Sinkhorn-Knopp algorithm.

## C   APPENDIX : TECHNICAL DETAILS

**Lemma C.1.** *We have for $\boldsymbol{Q} \in \mathbb{R}^{|\mathcal{S}||\mathcal{A}|}$,*

$$\|\boldsymbol{A}_{\boldsymbol{Q}}\|_\infty \leq 1 - (1 - \gamma)d_{\min}\alpha.$$

*Proof.* For $i \in [|\mathcal{S}||\mathcal{A}|]$, we have

$$\sum_{j=1}^{|\mathcal{S}||\mathcal{A}|} |[\boldsymbol{A}_{\boldsymbol{Q}}]_{ij}| \leq 1 - [\boldsymbol{D}]_{ii}\alpha + \alpha[\boldsymbol{D}]_{ii}\gamma \sum_{j=1}^{|\mathcal{S}||\mathcal{A}|} [\boldsymbol{P}\boldsymbol{\Pi}^{\boldsymbol{Q}}]_{ij}$$
$$= 1 - [\boldsymbol{D}]_{ii}(1 - \gamma)\alpha.$$

The last equality follows from the fact that $\boldsymbol{P}\boldsymbol{\Pi}^{\boldsymbol{Q}}$ is a stochastic matrix, i.e., the row sum equals to one, and represents a probability distribution. Taking maximum over $i \in [|\mathcal{S}||\mathcal{A}|]$, we complete the proof. $\square$

**Lemma C.2.** *For $k \in \mathbb{N}$, we have*

$$\|\boldsymbol{\epsilon}_k^{\text{avg}}\|_\infty \leq \frac{4R_{\max}}{1 - \gamma}.$$

*Proof.* From the definition of $\boldsymbol{\epsilon}_k^{\text{avg}} = \frac{1}{N}\sum_{i=1}^N \boldsymbol{\delta}_k^i - \boldsymbol{\Delta}_k^i$ in (4), we have

$$\|\boldsymbol{\epsilon}_k^{\text{avg}}\|_\infty \leq 2\left(R_{\max} + \gamma\frac{R_{\max}}{1 - \gamma} + \frac{R_{\max}}{1 - \gamma}\right)$$
$$= \frac{4R_{\max}}{1 - \gamma},$$

where the first inequality comes from the bonundedness of $\bar{\boldsymbol{Q}}_k$ in Lemma 3.3. This completes the proof. $\square$

**Lemma C.3.** *For $a, b \in (0, 1)$, and for $k \in \mathbb{N}$, we have*

$$\sum_{i=0}^{k} a^{k-i} b^i \leq a^{\frac{k}{2}} \frac{1}{1-b} + b^{\frac{k}{2}} \frac{1}{1-a}.$$

*Furthermore, we have*

$$\sum_{i=\tau}^{k} a^{k-i} b^{i-\tau} \leq a^{\frac{k-\tau}{2}} \frac{1}{1-b} + b^{\frac{k-\tau}{2}} \frac{1}{1-a}.$$

*Proof.* We have

$$\sum_{i=0}^{k} a^{k-i} b^i \leq \sum_{i=0}^{\lceil \frac{k}{2} \rceil} a^{k-i} b^i + \sum_{i=\lfloor \frac{k}{2} \rfloor}^{k} a^{k-i} b^i$$

$$\leq a^{\frac{k}{2}} \frac{1}{1-b} + b^{\frac{k}{2}} \frac{1}{1-a}.$$

The last inequality follows from the summation of geometric series. As for the second item, we have

$$\sum_{i=\tau}^{k} a^{k-i} b^{i-\tau} \leq \sum_{i=\tau}^{\lceil \frac{k+\tau}{2} \rceil} a^{k-i} b^{i-\tau} + \sum_{i=\lfloor \frac{k+\tau}{2} \rfloor}^{k} a^{k-i} b^{i-\tau}$$

$$\leq a^{\frac{k-\tau}{2}} \frac{1}{1-b} + b^{\frac{k-\tau}{2}} \frac{1}{1-a}.$$

This completes the proof. $\square$

**Lemma C.4** (Azuma-Hoeffding Inequality, Theorem 2.19 in Chung and Lu (2006)). *Let $\{S_n\}_{n \in \mathbb{N}}$ be a Martingale sequence with $S_0 = 0$. Suppose $|S_k - S_{k-1}| \leq c_k$ for $k \in \mathbb{N}$. Then, for $\epsilon \geq 0$, we have*

$$\mathbb{P}\left[|S_k| \geq \epsilon\right] \leq 2 \exp\left(-\frac{\epsilon^2}{2 \sum_{j=1}^{k} c_j^2}\right).$$

**Lemma C.5.** *Suppose $X \geq 0$, $\mathbb{P}\left[X \geq \epsilon\right] \leq \min\left\{a \exp\left(-b\epsilon^2\right), 1\right\}$, and $a \geq 2$. Then, we have*

$$\mathbb{E}\left[X\right] \leq 2\sqrt{\frac{\ln a}{b}}.$$

*Proof.* We have

$$\mathbb{E}\left[X\right] = \int_0^\infty \mathbb{P}\left[X \geq s\right] ds$$

$$\leq \int_0^\infty \min\left\{a \exp\left(-bs^2\right), 1\right\} ds$$

$$\leq \int_0^{\sqrt{\frac{\ln a}{b}}} 1 ds + \int_{\sqrt{\frac{\ln a}{b}}}^\infty a \exp(-bs^2) ds$$

$$\leq \sqrt{\frac{\ln a}{b}} + \frac{1}{2\sqrt{b \ln a}}$$

$$\leq 2\sqrt{\frac{\ln a}{b}}.$$

The last inequality follows from the fact that $4 \ln a > 1 / \ln a$. The third inequality follows from the following relation:

$$
\begin{aligned}
\int_{\sqrt{\frac{\ln a}{b}}}^{\infty} a \exp(-bs^2) ds &= a \int_{\frac{\ln a}{b}}^{\infty} \frac{1}{2\sqrt{u}} \exp(-bu) du \\
&\leq \frac{a}{2} \sqrt{\frac{b}{\ln a}} \int_{\frac{\ln a}{b}}^{\infty} \exp(-bu) du \\
&= \frac{a}{2} \sqrt{\frac{b}{\ln a}} \frac{1}{b} \left[ -\exp(-bu) \right]_{\frac{\ln a}{b}}^{\infty} \\
&= \frac{1}{2\sqrt{b \ln a}}.
\end{aligned}
$$

where we used the change of variables $s^2 = u$ in the first equality. $\qquad\square$

**Definition C.6** (Martingale sequence, Section 4.2 in Durrett (2019)). *Consider a sequence of random variables $\{X_n\}_{n \in \mathbb{N}}$ and an increasing $\sigma$-field, $\mathcal{F}_n$, such that*

*1)* $\mathbb{E}[|X_n|] < \infty$;

*2)* $X_n$ *is* $\mathcal{F}_n$-*measurable*;

*3)* $\mathbb{E}[X_{n+1}|\mathcal{F}_n] = X_n, \quad \forall n \in \mathbb{N}$.

*Then, $X_n$ is said to be a Martingale sequence.*

**Lemma C.7** (Proposition 3.4 in Paulin (2015)). *For uniformly ergodic Markov chain in Section 4, we have, for $\epsilon > 0$,*

$$
\tau(\epsilon) \leq t_{\mathrm{mix}} \left( 1 + 2 \log\left(\frac{1}{\epsilon}\right) + \log\left(\frac{1}{d_{\min}}\right) \right),
$$

*where $\tau$ and $t_{\mathrm{mix}}$ are defined in (13).*

# D    Appendix : Omitted Proofs

## D.1    Proof of Lemma 3.2

*Proof.* From the definition of $\bar{\boldsymbol{W}}$ in (4), we have

$$
\begin{aligned}
(\bar{\boldsymbol{W}}^k \boldsymbol{\Theta})^{\top} \bar{\boldsymbol{W}}^k \boldsymbol{\Theta} &= \bar{\boldsymbol{W}}^{2k} - 2\bar{\boldsymbol{W}}^{k\top} \frac{1}{N} \left( (\mathbf{1}_N \mathbf{1}_N^{\top}) \otimes \boldsymbol{I}_{|\mathcal{S}||\mathcal{A}|} \right) + \frac{1}{N} (\mathbf{1}_N \mathbf{1}_N^{\top}) \otimes \boldsymbol{I}_{|\mathcal{S}||\mathcal{A}|} \\
&= \left( \boldsymbol{W}^{2k} - \frac{1}{N} \mathbf{1}_N \mathbf{1}_N^{\top} \right) \otimes \boldsymbol{I}_{|\mathcal{S}||\mathcal{A}|},
\end{aligned}
$$

where the second equality follows from the fact that $\bar{\boldsymbol{W}}(\mathbf{1}_N \mathbf{1}_N)^{\top} \otimes \boldsymbol{I}_{|\mathcal{S}||\mathcal{A}|} = (\mathbf{1}_N \mathbf{1}_N)^{\top} \otimes \boldsymbol{I}_{|\mathcal{S}||\mathcal{A}|}$. From the result, we can derive

$$
\left\| \bar{\boldsymbol{W}}^k \boldsymbol{\Theta} \right\|_2 = \sqrt{\lambda_{\max} \left( (\bar{\boldsymbol{W}}^k \boldsymbol{\Theta})^{\top} \bar{\boldsymbol{W}}^k \boldsymbol{\Theta} \right)} = \sqrt{\lambda_{\max} \left( \boldsymbol{W}^{2k} - \frac{1}{N} \mathbf{1}_N \mathbf{1}_N^{\top} \right)} = \sigma_2(\boldsymbol{W})^k < 1. \quad (17)
$$

To prove the inequality in (17), we first prove that 1 is the unique largest eigenvalue of $\boldsymbol{W}$. Noting that $\mathbf{1}_N$ is an eigenvector of $\boldsymbol{W}$ with eigenvalue of 1, and $\rho(\boldsymbol{W}) \leq \|\boldsymbol{W}\|_{\infty} = 1$ where $\rho(\cdot)$ is the spectral radius of a matrix, the largest eigenvalue of $\boldsymbol{W}$ should be one. This implies that $\sigma_2(\boldsymbol{W}) < 1$. The multiplicity of the eigenvalue 1 is one, which follows from the fact that $\boldsymbol{W}^k$ is a non-negative and irreducible matrix and that the largest eigenvalue of a non-negative and irreducible matrix is unique Pillai et al. (2005) from Perron-Frobenius theorem. Note that $\boldsymbol{W}^k$ is a non-negative and irreducible matrix due to the fact that the graph $\mathcal{G}$ is connected.

Next, we use the eigenvalue decomposition of a symmetric matrix to investigate the spectrum of $\boldsymbol{W}^{2k} - \frac{1}{N}\mathbf{1}_N\mathbf{1}_N^\top$. By eigendecomposition of a symmetric matrix, we have

$$\boldsymbol{W} = \lambda_1\boldsymbol{v}_1\boldsymbol{v}_1^\top + \sum_{j=2}^{N}\lambda_j\boldsymbol{v}_j\boldsymbol{v}_j^\top = \boldsymbol{T}\boldsymbol{\Lambda}\boldsymbol{T}^{-1},$$

where $\boldsymbol{v}_j$ and $\lambda_j$ are $j$-th eigenvector and eigenvalue of $\boldsymbol{W}$, $\lambda_1 = 1$, $\boldsymbol{v}_1 = \frac{1}{\sqrt{N}}\mathbf{1}_N$, $\boldsymbol{\Lambda}$ is a diagonal matrix whose diagonal elements are the eigenvalues of $\boldsymbol{W}$, and $\boldsymbol{T}$ and $\boldsymbol{T}^{-1}$ are formed from the eigenvectors of $\boldsymbol{W}$. From the uniqueness of the maximum eigenvalue of $\boldsymbol{W}$, we have $\lambda_1 = 1 > \lambda_j, j \in \{2, 3, \ldots, N\}$. Therefore, we have

$$\boldsymbol{W}^{2k} = \boldsymbol{T}\boldsymbol{\Lambda}^{2k}\boldsymbol{T}^{-1} = \left(\frac{1}{\sqrt{N}}\mathbf{1}_N\right)\left(\frac{1}{\sqrt{N}}\mathbf{1}_N^\top\right) + \sum_{j=2}^{N}\lambda_j^k\boldsymbol{v}_j\boldsymbol{v}_j^\top.$$

Therefore, we have $\lambda_{\max}\left(\boldsymbol{W}^{2k} - \frac{1}{N}\mathbf{1}_N\mathbf{1}_N^\top\right) = \sigma_2(\boldsymbol{W}^{2k})$. This completes the proof. $\qquad\square$

### D.2 PROOF OF LEMMA 3.3

*Proof.* Let us first assume that for some $k \in \mathbb{N}$, $\left\|\boldsymbol{Q}_k^i\right\|_\infty \leq \frac{R_{\max}}{1-\gamma}$ for all $i \in [N]$. Then, considering (2), for all $i \in [N]$, we have

$$|\boldsymbol{Q}_{k+1}^i(s_k, \boldsymbol{a}_k)| \leq ([\boldsymbol{W}]_{ii} - \alpha)\left\|\boldsymbol{Q}_k^i\right\|_\infty + \sum_{j \in [N]\setminus\{i\}}[\boldsymbol{W}]_{ij}\left\|\boldsymbol{Q}_k^j\right\|_\infty + \alpha\left(R_{\max} + \gamma\left\|\boldsymbol{Q}_k^i\right\|_\infty\right)$$

$$\leq (1-\alpha)\frac{R_{\max}}{1-\gamma} + \alpha\frac{R_{\max}}{1-\gamma}$$

$$= \frac{R_{\max}}{1-\gamma}.$$

The first inequality follows from the fact that $\alpha \leq \min_{i \in [N]}[\boldsymbol{W}]_{ii}$. The second inequality follows from the induction hypothesis. For, $s, \boldsymbol{a} \in \mathcal{S} \times \mathcal{A} \setminus \{s_k, \boldsymbol{a}_k\}$, we have

$$\left|\boldsymbol{Q}_{k+1}^i(s, \boldsymbol{a})\right| \leq \sum_{j \in \mathcal{N}_i}[\boldsymbol{W}]_{ij}\left|\boldsymbol{Q}_k^j(s, \boldsymbol{a})\right| \leq \frac{R_{\max}}{1-\gamma}.$$

The last line follows from the fact that $\boldsymbol{W}$ is a doubly stochastic matrix, and the induction hypothesis. The proof is completed by applying the induction argument.

$\qquad\square$

### D.3 PROOF OF THEOREM 3.4

*Proof.* Taking infinity norm on (7), we get

$$\left\|\boldsymbol{\Theta}\bar{\boldsymbol{Q}}_{k+1}\right\|_\infty \leq \left\|\bar{\boldsymbol{W}}^{k+1}\boldsymbol{\Theta}\bar{\boldsymbol{Q}}_0\right\|_2 + \alpha\sqrt{N|\mathcal{S}||\mathcal{A}|}\sum_{j=0}^{k}\left\|\bar{\boldsymbol{W}}^{k-j}\boldsymbol{\Theta}\right\|_2\left\|\left(\bar{\boldsymbol{D}}\left(\bar{\boldsymbol{R}} + \gamma\bar{\boldsymbol{P}}\bar{\boldsymbol{\Pi}}^{\bar{\boldsymbol{Q}}_j}\bar{\boldsymbol{Q}}_j - \bar{\boldsymbol{Q}}_j\right) + \bar{\boldsymbol{\epsilon}}_j\right)\right\|_\infty$$

$$\leq \left\|\bar{\boldsymbol{W}}^{k+1}\boldsymbol{\Theta}\bar{\boldsymbol{Q}}_0\right\|_2 + \alpha\sqrt{N|\mathcal{S}||\mathcal{A}|}\sum_{j=0}^{k}\left\|\bar{\boldsymbol{W}}^{k-j}\boldsymbol{\Theta}\right\|_2\frac{8R_{\max}}{1-\gamma}$$

$$\leq \sigma_2(\boldsymbol{W})^{k+1}\left\|\boldsymbol{\Theta}\bar{\boldsymbol{Q}}_0\right\|_2 + \alpha\sqrt{N|\mathcal{S}||\mathcal{A}|}\sum_{j=0}^{k}\sigma_2(\boldsymbol{W})^{k-j}\frac{8R_{\max}}{1-\gamma}$$

$$\leq \sigma_2(\boldsymbol{W})^{k+1}\left\|\boldsymbol{\Theta}\bar{\boldsymbol{Q}}_0\right\|_2 + \alpha\frac{8R_{\max}}{1-\gamma}\frac{\sqrt{N|\mathcal{S}||\mathcal{A}|}}{1-\sigma_2(\boldsymbol{W})}.$$

The first inequality follows from the inequality $\|\boldsymbol{A}\|_\infty \leq \sqrt{N|\mathcal{S}||\mathcal{A}|}\|\boldsymbol{A}\|_2$ for $\boldsymbol{A} \in \mathbb{R}^{N|\mathcal{S}||\mathcal{A}| \times N|\mathcal{S}||\mathcal{A}|}$. The second inequality follows from the bound on $\bar{\boldsymbol{Q}}_k$ in Lemma 3.3. The third inequality follows from Lemma 3.2. The last inequality follows from summation of geometric series. This completes the proof.

$\qquad\square$

### D.4 Proof of Lemma 3.5

*Proof.* From the definition of $\boldsymbol{E}_k$ in (9), we get

$$\|\boldsymbol{E}_k\|_\infty \leq \frac{\gamma}{N} \sum_{i=1}^N \left\| \boldsymbol{D} \boldsymbol{P} (\boldsymbol{\Pi}^{\boldsymbol{Q}_k^i} \boldsymbol{Q}_k^i - \boldsymbol{\Pi}^{\boldsymbol{Q}_k^{\mathrm{avg}}} \boldsymbol{Q}_k^{\mathrm{avg}}) \right\|_\infty$$

$$\leq \frac{\gamma d_{\max}}{N} \sum_{i=1}^N \left\| \begin{bmatrix} \max_{\boldsymbol{a} \in \mathcal{A}} \boldsymbol{Q}_k^i(1, \boldsymbol{a}) - \max_{\boldsymbol{a} \in \mathcal{A}} \boldsymbol{Q}_k^{\mathrm{avg}}(1, \boldsymbol{a}) \\ \max_{\boldsymbol{a} \in \mathcal{A}} \boldsymbol{Q}_k^i(2, \boldsymbol{a}) - \max_{\boldsymbol{a} \in \mathcal{A}} \boldsymbol{Q}_k^{\mathrm{avg}}(2, \boldsymbol{a}) \\ \vdots \\ \max_{\boldsymbol{a} \in \mathcal{A}} \boldsymbol{Q}_k^i(|\mathcal{S}|, \boldsymbol{a}) - \max_{\boldsymbol{a} \in \mathcal{A}} \boldsymbol{Q}_k^{\mathrm{avg}}(|\mathcal{S}|, \boldsymbol{a}) \end{bmatrix} \right\|_\infty$$

$$\leq \frac{\gamma d_{\max}}{N} \sum_{i=1}^N \left\| \boldsymbol{Q}_k^i - \boldsymbol{Q}_k^{\mathrm{avg}} \right\|_\infty$$

$$\leq \gamma d_{\max} \left\| \boldsymbol{\Theta} \bar{\boldsymbol{Q}}_k \right\|_\infty .$$

The third inequality follows from the fact that $|\max_{i \in [n]} [\boldsymbol{x}]_i - \max_i [\boldsymbol{y}]_i| \leq \max_{i \in [n]} |\boldsymbol{x}_i - \boldsymbol{y}_i|$ for $\boldsymbol{x}, \boldsymbol{y} \in \mathbb{R}^n$ and $n \in \mathbb{N}$. The last inequality follows from the fact that

$$\left\| \boldsymbol{Q}_k^i - \boldsymbol{Q}_k^{\mathrm{avg}} \right\|_\infty \leq \left\| \boldsymbol{\Theta} \bar{\boldsymbol{Q}}_k \right\|_\infty , \quad \forall i \in [N].$$

This completes the proof. $\qquad \square$

## E Appendix : Construction of upper and lower comparison system

### E.1 Construction of lower comparison system

**Lemma E.1.** *For $k \in \mathbb{N}$, if $\boldsymbol{Q}_0^{\mathrm{avg},l} \leq \boldsymbol{Q}_0^{\mathrm{avg}}$, we have*

$$\boldsymbol{Q}_k^{\mathrm{avg},l} \leq \boldsymbol{Q}_k^{\mathrm{avg}}.$$

*Proof.* The proof follows from the induction argument. Suppose the statement holds for some $k \in \mathbb{N}$. Then, we have

$$\boldsymbol{Q}_{k+1}^{\mathrm{avg},l} = \boldsymbol{Q}_k^{\mathrm{avg},l} + \alpha \boldsymbol{D} \left( \boldsymbol{R}^{\mathrm{avg}} + \gamma \boldsymbol{P} \boldsymbol{\Pi}^{\boldsymbol{Q}^*} \boldsymbol{Q}_k^{\mathrm{avg},l} - \boldsymbol{Q}_k^{\mathrm{avg},l} \right) + \alpha \boldsymbol{\epsilon}_k^{\mathrm{avg}} + \alpha \boldsymbol{E}_k$$

$$\leq \boldsymbol{Q}_k^{\mathrm{avg}} + \alpha \boldsymbol{D} \left( \boldsymbol{R}^{\mathrm{avg}} + \gamma \boldsymbol{P} \boldsymbol{\Pi}^{\boldsymbol{Q}_k^{\mathrm{avg}}} \boldsymbol{Q}_k^{\mathrm{avg}} - \boldsymbol{Q}_k^{\mathrm{avg}} \right) + \alpha \boldsymbol{\epsilon}_k^{\mathrm{avg}} + \alpha \boldsymbol{E}_k$$

$$= \boldsymbol{Q}_{k+1}^{\mathrm{avg}}.$$

The first inequality follows from the fact that $\boldsymbol{Q}_k^{\mathrm{avg},l} \leq \boldsymbol{Q}_k^{\mathrm{avg}}$ and $\boldsymbol{\Pi}^{\boldsymbol{Q}^*} \boldsymbol{Q}_k^{\mathrm{avg},l} \leq \boldsymbol{\Pi}^{\boldsymbol{Q}^*} \boldsymbol{Q}_k^{\mathrm{avg}} \leq \boldsymbol{\Pi}^{\boldsymbol{Q}_k^{\mathrm{avg}}} \boldsymbol{Q}_k^{\mathrm{avg}}$. The proof is completed by the induction argument. $\qquad \square$

### E.2 Construction of upper comparison system

**Lemma E.2.** *For $k \in \mathbb{N}$, if $\tilde{\boldsymbol{Q}}_0^{\mathrm{avg},u} \geq \tilde{\boldsymbol{Q}}_0^{\mathrm{avg}}$, we have*

$$\tilde{\boldsymbol{Q}}_k^{\mathrm{avg},u} \geq \tilde{\boldsymbol{Q}}_k^{\mathrm{avg}}.$$

*Proof.* As in the construction of the lower comparison system in Lemma E.1 in Appendix, the proof follows from an induction argument. Suppose that the statement holds for some $k \in \mathbb{N}$. Then, we have

$$\tilde{\boldsymbol{Q}}_{k+1}^{\mathrm{avg}} = \tilde{\boldsymbol{Q}}_k^{\mathrm{avg}} + \alpha \boldsymbol{D} \left( \gamma \boldsymbol{P} \boldsymbol{\Pi}^{\boldsymbol{Q}_k^{\mathrm{avg}}} \tilde{\boldsymbol{Q}}_k^{\mathrm{avg}} - \tilde{\boldsymbol{Q}}_k^{\mathrm{avg}} \right) + \alpha \gamma \boldsymbol{D} \boldsymbol{P} (\boldsymbol{\Pi}^{\boldsymbol{Q}_k^{\mathrm{avg}}} \boldsymbol{Q}^* - \boldsymbol{\Pi}^{\boldsymbol{Q}^*} \boldsymbol{Q}^*)$$

$$\quad + \alpha \boldsymbol{\epsilon}_k^{\mathrm{avg}} + \alpha \boldsymbol{D} \boldsymbol{E}_k$$

$$\leq (\boldsymbol{I} + \alpha \boldsymbol{D} (\gamma \boldsymbol{P} \boldsymbol{\Pi}^{\boldsymbol{Q}_k^{\mathrm{avg}}} - \boldsymbol{I}) \tilde{\boldsymbol{Q}}_k^{\mathrm{avg},u} + \alpha \boldsymbol{\epsilon}_k^{\mathrm{avg}} + \alpha \boldsymbol{D} \boldsymbol{E}_k$$

$$= \tilde{\boldsymbol{Q}}_{k+1}^{\mathrm{avg},u}.$$

The inequality follows from the fact that the elements of $\boldsymbol{I} + \alpha\boldsymbol{D}(\gamma\boldsymbol{P}\boldsymbol{\Pi}^{\boldsymbol{Q}_k^{\mathrm{avg}}} - \boldsymbol{I})$ are all non-negative, and $\boldsymbol{\Pi}^{\boldsymbol{Q}_k^{\mathrm{avg}}}\boldsymbol{Q}^* \leq \boldsymbol{\Pi}^{\boldsymbol{Q}^*}\boldsymbol{Q}^*$. The proof is completed by the induction argument. $\square$

## F  APPENDIX : I.I.D. OBSERVATION MODEL

**Proposition F.1.** *Assume i.i.d. observation model, and $\alpha \leq \min_{i\in[N]}[\boldsymbol{W}]_{ii}$. Then, we have, for $k \in \mathbb{N}$,*

$$\mathbb{E}\left[\left\|\tilde{\boldsymbol{Q}}_{k+1}^{\mathrm{avg},l}\right\|_\infty\right] = \tilde{\mathcal{O}}\left((1-(1-\gamma)d_{\min}\alpha)^{\frac{k}{2}} + \sigma_2(\boldsymbol{W})^{\frac{k}{2}}\right)$$

$$+ \tilde{\mathcal{O}}\left(\alpha^{\frac{1}{2}}\frac{R_{\max}}{(1-\gamma)^{\frac{3}{2}}d_{\min}^{\frac{1}{2}}} + \alpha d_{\max}\frac{R_{\max}\sqrt{N|\mathcal{S}||\mathcal{A}|}}{(1-\gamma)^2 d_{\min}(1-\sigma_2(\boldsymbol{W}))}\right).$$

Let us first introduce a key lemma to prove Proposition F.1:

**Lemma F.2.** *For $k \in \mathbb{N}$, we have*

$$\mathbb{E}\left[\left\|\sum_{i=0}^k \boldsymbol{A}_{\boldsymbol{Q}^*}^{k-i}\boldsymbol{\epsilon}_i^{\mathrm{avg}}\right\|_\infty\right] \leq \frac{8\sqrt{2}R_{\max}}{(1-\gamma)^{\frac{3}{2}}d_{\min}^{\frac{1}{2}}\alpha^{\frac{1}{2}}}\sqrt{\ln(2|\mathcal{S}||\mathcal{A}|)}.$$

*Proof.* For the proof, we will apply Azuma-Hoeffding inequality in Lemma C.4. For simplicity, let $\boldsymbol{S}_t = \sum_{i=0}^t \boldsymbol{A}_{\boldsymbol{Q}^*}^{k-i}\boldsymbol{\epsilon}_i^{\mathrm{avg}}$, for $0 \leq t \leq k$. Let $\mathcal{F}_t := \sigma(\{(s_i, \boldsymbol{a}_i, s_i')\}_{i=0}^t \cup \{\bar{\boldsymbol{Q}}_0\})$, which is the $\sigma$-algebra generated by $\{(s_i, \boldsymbol{a}_i, s_i')\}_{i=0}^t$ and $\bar{\boldsymbol{Q}}_0$. Letting $[\boldsymbol{S}_t]_{s,\boldsymbol{a}} = (\boldsymbol{e}_s \otimes \boldsymbol{e}_{\boldsymbol{a}})^\top \boldsymbol{S}_t$, for $s, \boldsymbol{a} \in \mathcal{S} \times \mathcal{A}$, let us check that $\{[\boldsymbol{S}_t]_{s,\boldsymbol{a}}\}_{t=0}^k$ is a Martingale sequence defined in Definition C.6. We can see that

$$\mathbb{E}[\boldsymbol{S}_t|\mathcal{F}_{t-1}] = \mathbb{E}\left[\boldsymbol{A}_{\boldsymbol{Q}^*}^{k-t}\boldsymbol{\epsilon}_t^{\mathrm{avg}} + \boldsymbol{S}_{t-1}\Big|\mathcal{F}_{t-1}\right]$$

$$= \boldsymbol{A}_{\boldsymbol{Q}^*}^{k-t}\mathbb{E}[\boldsymbol{\epsilon}_t^{\mathrm{avg}}|\mathcal{F}_{t-1}] + \boldsymbol{S}_{t-1}$$

$$= \boldsymbol{S}_{t-1},$$

where the second line is due to the fact that $\boldsymbol{S}_{t-1}$ is $\mathcal{F}_{t-1}$-measurable, and the last line follows from $\mathbb{E}[\boldsymbol{\epsilon}_t^{\mathrm{avg}}|\mathcal{F}_{t-1}] = \boldsymbol{0}$ thanks to the i.i.d. observation model. Therefore, we have $\mathbb{E}[[\boldsymbol{S}_t]_{s,\boldsymbol{a}}|\mathcal{F}_{t-1}] = [\boldsymbol{S}_{t-1}]_{s,\boldsymbol{a}}$.

Moreover, we have

$$\mathbb{E}[\boldsymbol{S}_0] = \mathbb{E}\left[\frac{1}{N}\sum_{i=1}^N (\boldsymbol{e}_{s_0} \otimes \boldsymbol{e}_{\boldsymbol{a}_0})(r_1^i + \boldsymbol{e}_{s_0'}^\top \gamma\boldsymbol{\Pi}^{\boldsymbol{Q}_0^i}\boldsymbol{Q}_0^i - (\boldsymbol{e}_{s_0} \otimes \boldsymbol{e}_{\boldsymbol{a}_0})^\top\boldsymbol{Q}_0^i)\right]$$

$$- \mathbb{E}\left[\frac{1}{N}\sum_{i=1}^N \boldsymbol{D}(\boldsymbol{R}^i + \gamma\boldsymbol{P}\boldsymbol{\Pi}^{\boldsymbol{Q}_0^i}\boldsymbol{Q}_k^i - \boldsymbol{Q}_0^i)\right]$$

$$= \boldsymbol{0}.$$

The last line follows from that $\mathbb{E}[\boldsymbol{e}_{s_0} \otimes \boldsymbol{e}_{\boldsymbol{a}_0}] = \boldsymbol{D}$ and $\mathbb{E}\left[(\boldsymbol{e}_{s_0} \otimes \boldsymbol{e}_{\boldsymbol{a}_0})\boldsymbol{e}_{s_0'}^\top\right] = \boldsymbol{D}\boldsymbol{P}$.

Therefore, $\{[\boldsymbol{S}_t]_{s,\boldsymbol{a}}\}_{t=0}^k$ is a Martingale sequence for any $s, \boldsymbol{a} \in \mathcal{S} \times \mathcal{A}$. Furthermore, we have

$$|[\boldsymbol{S}_t]_{s,\boldsymbol{a}} - [\boldsymbol{S}_{t-1}]_{s,\boldsymbol{a}}| \leq \|\boldsymbol{S}_t - \boldsymbol{S}_{t-1}\|_\infty = \left\|\boldsymbol{A}_{\boldsymbol{Q}^*}^{k-t}\boldsymbol{\epsilon}_t^{\mathrm{avg}}\right\|_\infty \leq (1-(1-\gamma)d_{\min}\alpha)^{k-t}\frac{4R_{\max}}{1-\gamma},$$

where the last inequality comes from Lemma C.1 and Lemma C.2. Furthermore, note that we have

$$\sum_{t=1}^k |[\boldsymbol{S}_t]_{s,\boldsymbol{a}} - [\boldsymbol{S}_{t-1}]_{s,\boldsymbol{a}}|^2 \leq \sum_{t=0}^k (1-(1-\gamma)d_{\min}\alpha)^{2k-2t}\frac{16R_{\max}^2}{(1-\gamma)^2}$$

$$\leq \frac{16R_{\max}^2}{(1-\gamma)^3 d_{\min}\alpha}.$$

Therefore, applying the Azuma-Hoeffding inequality in Lemma C.4 in the Appendix, we have

$$\mathbb{P}\left[|[\boldsymbol{S}_k]_{s,a}| \ge \epsilon\right] \le 2 \exp\left(-\frac{\epsilon^2 (1-\gamma)^3 d_{\min} \alpha}{32 R_{\max}^2}\right).$$

Noting that $\{\|\boldsymbol{S}_k\|_\infty \ge \epsilon\} \subseteq \cup_{s,\boldsymbol{a} \in \mathcal{S} \times \mathcal{A}}\{|[\boldsymbol{S}_k]_{s,\boldsymbol{a}}| \ge \epsilon\}$, using the union bound of the events, we get:

$$\mathbb{P}\left[\|\boldsymbol{S}_k\|_\infty \ge \epsilon\right] \le \sum_{s,\boldsymbol{a} \in \mathcal{S} \times \mathcal{A}} \mathbb{P}\left[|[\boldsymbol{S}_k]_{s,\boldsymbol{a}}| \ge \epsilon\right] \le 2|\mathcal{S}||\mathcal{A}| \exp\left(-\frac{\epsilon^2 (1-\gamma)^3 d_{\min} \alpha}{32 R_{\max}^2}\right).$$

Moreover, since a probability of an event is always smaller than one, we have

$$\mathbb{P}\left[\|\boldsymbol{S}_k\|_\infty \ge \epsilon\right] \le \min\left\{2|\mathcal{S}||\mathcal{A}| \exp\left(-\frac{\epsilon^2 (1-\gamma)^3 d_{\min} \alpha}{32 R_{\max}^2}\right), 1\right\}.$$

Now, we are ready to bound $\boldsymbol{S}_k$ from Lemma C.5 in the Appendix:

$$\mathbb{E}\left[\|\boldsymbol{S}_k\|_\infty\right] = \int_0^\infty \mathbb{P}\left[\|\boldsymbol{S}_k\|_\infty \ge x\right] dx \le \frac{8\sqrt{2} R_{\max}}{(1-\gamma)^{\frac{3}{2}} d_{\min}^{\frac{1}{2}} \alpha^{\frac{1}{2}}} \sqrt{\ln(2|\mathcal{S}||\mathcal{A}|)}.$$

This completes the proof.

$\square$

Now, we are ready prove Proposition F.1:

*Proof of Proposition F.1.* Recursively expanding the equation in (12), we get

$$\begin{aligned}
\tilde{\boldsymbol{Q}}_{k+1}^{\mathrm{avg},l} &= \boldsymbol{A}_{\boldsymbol{Q}^*} \tilde{\boldsymbol{Q}}_k^{\mathrm{avg},l} + \alpha \boldsymbol{\epsilon}_k^{\mathrm{avg}} + \alpha \boldsymbol{E}_k \\
&= \boldsymbol{A}_{\boldsymbol{Q}^*}^2 \tilde{\boldsymbol{Q}}_{k-1}^{\mathrm{avg},l} + \alpha \boldsymbol{A}_{\boldsymbol{Q}^*} \boldsymbol{\epsilon}_{k-1}^{\mathrm{avg}} + \alpha \boldsymbol{A}_{\boldsymbol{Q}^*} \boldsymbol{E}_{k-1} + \alpha \boldsymbol{\epsilon}_k^{\mathrm{avg}} + \alpha \boldsymbol{E}_k \\
&= \boldsymbol{A}_{\boldsymbol{Q}^*}^{k+1} \tilde{\boldsymbol{Q}}_0^{\mathrm{avg},l} + \alpha \sum_{i=0}^k \boldsymbol{A}_{\boldsymbol{Q}^*}^{k-i} \boldsymbol{\epsilon}_i^{\mathrm{avg}} + \alpha \sum_{i=0}^k \boldsymbol{A}_{\boldsymbol{Q}^*}^{k-i} \boldsymbol{E}_i.
\end{aligned}$$

Taking infinity norm and expectation on both sides of the above equation, we get

$$\begin{aligned}
&\mathbb{E}\left[\left\|\tilde{\boldsymbol{Q}}_{k+1}^{\mathrm{avg},l}\right\|_\infty\right] \\
&\le \mathbb{E}\left[\left\|\boldsymbol{A}_{\boldsymbol{Q}^*}^{k+1}\right\|_\infty \left\|\tilde{\boldsymbol{Q}}_0^{\mathrm{avg},l}\right\|_\infty + \alpha \left\|\sum_{i=0}^k \boldsymbol{A}_{\boldsymbol{Q}^*}^{k-i} \boldsymbol{\epsilon}_i^{\mathrm{avg}}\right\|_\infty + \alpha \sum_{i=0}^k \left\|\boldsymbol{A}_{\boldsymbol{Q}^*}^{k-i}\right\|_\infty \|\boldsymbol{E}_i\|_\infty\right] \\
&\le (1-(1-\gamma)d_{\min}\alpha)^{k+1} \left\|\tilde{\boldsymbol{Q}}_0^{\mathrm{avg},l}\right\|_\infty + \alpha \mathbb{E}\left[\left\|\sum_{i=0}^k \boldsymbol{A}_{\boldsymbol{Q}^*}^{k-i} \boldsymbol{\epsilon}_i^{\mathrm{avg}}\right\|_\infty\right] \\
&\quad + \alpha \mathbb{E}\left[\sum_{i=0}^k \left\|\boldsymbol{A}_{\boldsymbol{Q}^*}^{k-i}\right\|_\infty \|\boldsymbol{E}_i\|_\infty\right] \\
&\le (1-(1-\gamma)d_{\min}\alpha)^{k+1} \left\|\tilde{\boldsymbol{Q}}_0^{\mathrm{avg},l}\right\|_\infty + \alpha^{\frac{1}{2}} \frac{8\sqrt{2} R_{\max}}{(1-\gamma)^{\frac{3}{2}} d_{\min}^{\frac{1}{2}}} \sqrt{\ln(2|\mathcal{S}||\mathcal{A}|)} \\
&\quad + \alpha \mathbb{E}\left[\sum_{i=0}^k \left\|\boldsymbol{A}_{\boldsymbol{Q}^*}^{k-i}\right\|_\infty \|\boldsymbol{E}_i\|_\infty\right] \\
&\le (1-(1-\gamma)d_{\min}\alpha)^{k+1} \left\|\tilde{\boldsymbol{Q}}_0^{\mathrm{avg},l}\right\|_\infty + \alpha^{\frac{1}{2}} \frac{8\sqrt{2} R_{\max}}{(1-\gamma)^{\frac{3}{2}} d_{\min}^{\frac{1}{2}}} \sqrt{\ln(2|\mathcal{S}||\mathcal{A}|)} \\
&\quad + \gamma d_{\max} \left\|\boldsymbol{\Theta}\bar{\boldsymbol{Q}}_0\right\|_2 \left((1-(1-\gamma)d_{\min}\alpha)^{\frac{k}{2}} \frac{\alpha}{1-\sigma_2(\boldsymbol{W})} + \sigma_2(\boldsymbol{W})^{\frac{k}{2}} \frac{1}{(1-\gamma)d_{\min}}\right) \\
&\quad + \alpha \gamma d_{\max} \frac{8 R_{\max} \sqrt{N|\mathcal{S}||\mathcal{A}|}}{(1-\gamma)^2 d_{\min}(1-\sigma_2(\boldsymbol{W}))}.
\end{aligned}$$

The second inequality follows from Lemma C.1. The third inequality follows from Lemma F.2. The last line follows from bounding $\sum_{i=0}^{k} \left\| \boldsymbol{A}_{\boldsymbol{Q}^*}^{k-i} \right\|_\infty \|\boldsymbol{E}_i\|_\infty$ as follows:

$$s \sum_{i=0}^{k} \left\| \boldsymbol{A}_{\boldsymbol{Q}^*}^{k-i} \right\|_\infty \|\boldsymbol{E}_i\|_\infty$$

$$\leq \gamma d_{\max} \sum_{i=0}^{k} (1-(1-\gamma)d_{\min}\alpha)^{k-i} \left( \sigma_2(\boldsymbol{W})^i \left\| \boldsymbol{\Theta}\bar{\boldsymbol{Q}}_0 \right\|_2 + \alpha \frac{8R_{\max}}{1-\gamma} \frac{\sqrt{N|\mathcal{S}||\mathcal{A}|}}{1-\sigma_2(\boldsymbol{W})} \right)$$

$$\leq \gamma d_{\max} \left\| \boldsymbol{\Theta}\bar{\boldsymbol{Q}}_0 \right\|_2 \left( (1-(1-\gamma)d_{\min}\alpha)^{\frac{k}{2}} \frac{1}{1-\sigma_2(\boldsymbol{W})} + \sigma_2(\boldsymbol{W})^{\frac{k}{2}} \frac{1}{(1-\gamma)d_{\min}\alpha} \right)$$

$$+ \gamma d_{\max} \frac{8R_{\max}\sqrt{N|\mathcal{S}||\mathcal{A}|}}{(1-\gamma)^2 d_{\min}(1-\sigma_2(\boldsymbol{W}))}.$$

The first inequality follows from Lemma 3.5 and Theorem 3.4. The second inequality follows from Lemma C.3 in the Appendix. This completes the proof. $\qquad\square$

Now, we bound $\tilde{\boldsymbol{Q}}_k^{\mathrm{avg},u}$ in (12). It is difficult to directly prove the convergence of upper comparison system. Therefore, we bound the difference of upper and lower comparison system, $\boldsymbol{Q}_k^{\mathrm{avg},u} - \boldsymbol{Q}_k^{\mathrm{avg},l}$. The good news is that since $\boldsymbol{Q}_k^{\mathrm{avg},u}$ and $\boldsymbol{Q}_k^{\mathrm{avg},l}$ shares the same error term $\boldsymbol{\epsilon}_k^{\mathrm{avg}}$ and $\boldsymbol{E}_k$, such terms will be removed if we subtract each others.

**Proposition F.3.** *For $k \in \mathbb{N}$, and $\alpha \leq \min_{i\in[N]}[\boldsymbol{W}]_{ii}$, we have*

$$\mathbb{E}\left[ \left\| \boldsymbol{Q}_{k+1}^{\mathrm{avg},u} - \boldsymbol{Q}_{k+1}^{\mathrm{avg},l} \right\|_\infty \right] = \tilde{\mathcal{O}}\left( (1-\alpha(1-\gamma)d_{\min})^{\frac{k}{2}} + \sigma_2(\boldsymbol{W})^{\frac{k}{4}} \right)$$

$$+ \tilde{\mathcal{O}}\left( \alpha^{\frac{1}{2}} \frac{d_{\max}R_{\max}}{(1-\gamma)^{\frac{5}{2}}d_{\min}^{\frac{3}{2}}} + \alpha \frac{d_{\max}^2 \sqrt{N|\mathcal{S}||\mathcal{A}|}R_{\max}}{(1-\gamma)^3 d_{\min}^2 (1-\sigma_2(\boldsymbol{W}))} \right).$$

*Proof.* Subtracting $\boldsymbol{Q}_{k+1}^{\mathrm{avg},l}$ from $\boldsymbol{Q}_{k+1}^{\mathrm{avg},u}$ in (12), we have

$$\boldsymbol{Q}_{k+1}^{\mathrm{avg},u} - \boldsymbol{Q}_{k+1}^{\mathrm{avg},l} = \boldsymbol{A}_{\boldsymbol{Q}_k^{\mathrm{avg}}}\tilde{\boldsymbol{Q}}_k^{\mathrm{avg},u} - \boldsymbol{A}_{\boldsymbol{Q}^*}\tilde{\boldsymbol{Q}}_k^{\mathrm{avg},l}$$

$$= \boldsymbol{A}_{\boldsymbol{Q}_k^{\mathrm{avg}}}(\boldsymbol{Q}_k^{\mathrm{avg},u} - \boldsymbol{Q}_k^{\mathrm{avg},l}) + (\boldsymbol{A}_{\boldsymbol{Q}_k^{\mathrm{avg}}} - \boldsymbol{A}_{\boldsymbol{Q}^*})\tilde{\boldsymbol{Q}}_k^{\mathrm{avg},l}$$

$$= \boldsymbol{A}_{\boldsymbol{Q}_k^{\mathrm{avg}}}(\boldsymbol{Q}_k^{\mathrm{avg},u} - \boldsymbol{Q}_k^{\mathrm{avg},l}) + \alpha\gamma \boldsymbol{D}\boldsymbol{P}(\boldsymbol{\Pi}^{\boldsymbol{Q}_k^{\mathrm{avg}}} - \boldsymbol{\Pi}^{\boldsymbol{Q}^*})\tilde{\boldsymbol{Q}}_k^{\mathrm{avg},l}. \qquad (18)$$

The last equality follows from the definition of $\boldsymbol{A}_{\boldsymbol{Q}_k^{\mathrm{avg}}}$ and $\boldsymbol{A}_{\boldsymbol{Q}^*}$ in (10).

Recursively expanding the terms, we get

$$\boldsymbol{Q}_{k+1}^{\mathrm{avg},u} - \boldsymbol{Q}_{k+1}^{\mathrm{avg},l} = \prod_{i=0}^{k} \boldsymbol{A}_{\boldsymbol{Q}_i^{\mathrm{avg}}}(\boldsymbol{Q}_0^{\mathrm{avg},u} - \boldsymbol{Q}_0^{\mathrm{avg},l})$$

$$+ \alpha\gamma \sum_{i=0}^{k-1}\prod_{j=i}^{k-1} \boldsymbol{A}_{\boldsymbol{Q}_{j+1}^{\mathrm{avg}}}\boldsymbol{D}\boldsymbol{P}(\boldsymbol{\Pi}^{\boldsymbol{Q}_i^{\mathrm{avg}}} - \boldsymbol{\Pi}^{\boldsymbol{Q}^*})\tilde{\boldsymbol{Q}}_i^{\mathrm{avg},l} + \alpha\gamma \boldsymbol{D}\boldsymbol{P}(\boldsymbol{\Pi}^{\boldsymbol{Q}_k^{\mathrm{avg}}} - \boldsymbol{\Pi}^{\boldsymbol{Q}^*})\tilde{\boldsymbol{Q}}_k^{\mathrm{avg},l}.$$

Taking infinity norm on both sides of the above equation, and using triangle inequality yields

$$\mathbb{E}\left[ \left\| \boldsymbol{Q}_{k+1}^{\mathrm{avg},u} - \boldsymbol{Q}_{k+1}^{\mathrm{avg},l} \right\|_\infty \right] \leq (1-\alpha(1-\gamma)d_{\min})^{k+1} \left\| \boldsymbol{Q}_0^{\mathrm{avg},u} - \boldsymbol{Q}_0^{\mathrm{avg},l} \right\|_\infty$$

$$+ 2\alpha\gamma d_{\max} \underbrace{\sum_{i=0}^{k} (1-\alpha(1-\gamma)d_{\min})^{k-i} \mathbb{E}\left[ \left\| \tilde{\boldsymbol{Q}}_i^{\mathrm{avg},l} \right\|_\infty \right]}_{(\star)}. \qquad (19)$$

The first inequality follows from Lemma C.1.

Now, we will use Proposition F.1 to bound $(\star)$ in the above inequality. We have

$$\sum_{i=0}^{k}(1-\alpha(1-\gamma)d_{\min})^{k-i}\mathbb{E}\left[\left\|\tilde{\boldsymbol{Q}}_i^{\mathrm{avg},l}\right\|_{\infty}\right] =\tilde{\mathcal{O}}\left(\sum_{j=0}^{k}(1-\alpha(1-\gamma)d_{\min})^{k-\frac{j}{2}}+(1-\alpha(1-\gamma)d_{\min})^{k-i}\sigma_2(\boldsymbol{W})^{\frac{i}{2}}\right)$$

$$+\tilde{\mathcal{O}}\left(\frac{R_{\max}}{\alpha^{\frac{1}{2}}(1-\gamma)^{\frac{5}{2}}d_{\min}^{\frac{3}{2}}}+\frac{d_{\max}\sqrt{N|\mathcal{S}||\mathcal{A}|}2R_{\max}}{(1-\gamma)^3 d_{\min}^2(1-\sigma_2(\boldsymbol{W}))}\right)$$

$$=\tilde{\mathcal{O}}\left((1-\alpha(1-\gamma)d_{\min})^{\frac{k}{2}}+\sigma_2(\boldsymbol{W})^{\frac{k}{4}}\right)$$

$$+\tilde{\mathcal{O}}\left(\frac{R_{\max}}{\alpha^{\frac{1}{2}}(1-\gamma)^{\frac{5}{2}}d_{\min}^{\frac{3}{2}}}+\frac{d_{\max}\sqrt{N|\mathcal{S}||\mathcal{A}|}R_{\max}}{(1-\gamma)^3 d_{\min}^2(1-\sigma_2(\boldsymbol{W}))}\right).$$

The last inequality follows from Lemma C.3. Applying this result to (19), we get

$$\mathbb{E}\left[\left\|\boldsymbol{Q}_{k+1}^{\mathrm{avg},u}-\boldsymbol{Q}_{k+1}^{\mathrm{avg},l}\right\|_{\infty}\right] =\tilde{\mathcal{O}}\left((1-\alpha(1-\gamma)d_{\min})^{\frac{k}{2}}+\sigma_2(\boldsymbol{W})^{\frac{k}{4}}\right)$$

$$+\tilde{\mathcal{O}}\left(\alpha^{\frac{1}{2}}d_{\max}\frac{R_{\max}}{(1-\gamma)^{\frac{5}{2}}d_{\min}^{\frac{3}{2}}}+\alpha\frac{d_{\max}^2\sqrt{N|\mathcal{S}||\mathcal{A}|}R_{\max}}{(1-\gamma)^3 d_{\min}^2(1-\sigma_2(\boldsymbol{W}))}\right).$$

This completes the proof. $\square$

### F.1 Proof of Theorem 3.6

*Proof.* $\left\|\tilde{\boldsymbol{Q}}_k^{\mathrm{avg}}\right\|_{\infty}$ can be bounded using the fact that $\tilde{\boldsymbol{Q}}_k^{\mathrm{avg},l}\leq\tilde{\boldsymbol{Q}}_k^{\mathrm{avg}}\leq\tilde{\boldsymbol{Q}}_k^{\mathrm{avg},u}$ :

$$\left\|\tilde{\boldsymbol{Q}}_k^{\mathrm{avg}}\right\|_{\infty} \leq\max\left\{\left\|\tilde{\boldsymbol{Q}}_k^{\mathrm{avg},l}\right\|_{\infty},\left\|\tilde{\boldsymbol{Q}}_k^{\mathrm{avg},u}\right\|_{\infty}\right\}$$

$$\leq\max\left\{\left\|\tilde{\boldsymbol{Q}}_k^{\mathrm{avg},l}\right\|_{\infty},\left\|\tilde{\boldsymbol{Q}}_k^{\mathrm{avg},l}\right\|_{\infty}+\left\|\tilde{\boldsymbol{Q}}_k^{\mathrm{avg},u}-\tilde{\boldsymbol{Q}}_k^{\mathrm{avg},l}\right\|_{\infty}\right\}$$

$$\leq\left\|\tilde{\boldsymbol{Q}}_k^{\mathrm{avg},l}\right\|_{\infty}+\left\|\tilde{\boldsymbol{Q}}_k^{\mathrm{avg},u}-\tilde{\boldsymbol{Q}}_k^{\mathrm{avg},l}\right\|_{\infty}$$

$$=\left\|\tilde{\boldsymbol{Q}}_k^{\mathrm{avg},l}\right\|_{\infty}+\left\|\boldsymbol{Q}_k^{\mathrm{avg},u}-\boldsymbol{Q}_k^{\mathrm{avg},l}\right\|_{\infty}.$$

The second inequality follows from triangle inequality. Taking expectation, from Proposition F.1 and Proposition F.3, we have the desired result. $\square$

### F.2 Proof of Theorem 3.7

*Proof.* Using triangle inequality, we have

$$\mathbb{E}\left[\left\|\bar{\boldsymbol{Q}}_k-\boldsymbol{1}_N\otimes\boldsymbol{Q}^*\right\|_{\infty}\right] \leq\mathbb{E}\left[\left\|\bar{\boldsymbol{Q}}_k-\boldsymbol{1}_N\otimes\boldsymbol{Q}_k^{\mathrm{avg}}\right\|_{\infty}\right]+\mathbb{E}\left[\left\|\boldsymbol{1}_N\otimes\boldsymbol{Q}_k^{\mathrm{avg}}-\boldsymbol{1}_N\otimes\boldsymbol{Q}^*\right\|_{\infty}\right]$$

$$=\mathbb{E}\left[\left\|\bar{\boldsymbol{Q}}_k-\boldsymbol{1}_N\otimes\boldsymbol{Q}_k^{\mathrm{avg}}\right\|_{\infty}\right]+\mathbb{E}\left[\left\|\boldsymbol{Q}_k^{\mathrm{avg}}-\boldsymbol{Q}^*\right\|_{\infty}\right]$$

$$=\tilde{\mathcal{O}}\left(\sigma_2(\boldsymbol{W})^k+\alpha\frac{\sqrt{N|\mathcal{S}||\mathcal{A}|}R_{\max}}{(1-\gamma)(1-\sigma_2(\boldsymbol{W}))}\right)$$

$$+\tilde{\mathcal{O}}\left((1-\alpha(1-\gamma)d_{\min})^{\frac{k}{2}}+\sigma_2(\boldsymbol{W})^{\frac{k}{4}}\right)$$

$$+\tilde{\mathcal{O}}\left(\alpha^{\frac{1}{2}}\frac{d_{\max}R_{\max}}{(1-\gamma)^{\frac{5}{2}}d_{\min}^{\frac{3}{2}}}+\alpha\frac{d_{\max}^2\sqrt{N|\mathcal{S}||\mathcal{A}|}R_{\max}}{(1-\gamma)^3 d_{\min}^2(1-\sigma_2(\boldsymbol{W}))}\right)$$

$$=\tilde{\mathcal{O}}\left((1-\alpha(1-\gamma)d_{\min})^{\frac{k}{2}}+\sigma_2(\boldsymbol{W})^{\frac{k}{4}}\right)$$

$$+\tilde{\mathcal{O}}\left(\alpha^{\frac{1}{2}}d_{\max}\frac{R_{\max}}{(1-\gamma)^{\frac{5}{2}}d_{\min}^{\frac{3}{2}}}+\alpha\frac{d_{\max}^2\sqrt{N|\mathcal{S}||\mathcal{A}|}R_{\max}}{(1-\gamma)^3 d_{\min}^2(1-\sigma_2(\boldsymbol{W}))}\right).$$

The first inequality comes from (6). The second inequality comes from Theorem 3.4 and 3.6. This completes the proof.

$\square$

### F.3 Proof of Corollary 3.8

*Proof.* Let us first bound the terms $\alpha^{\frac{1}{2}} d_{\max} \frac{R_{\max}}{(1-\gamma)^{\frac{5}{2}} d_{\min}^{\frac{3}{2}}} + \alpha \frac{d_{\max}^2 \sqrt{|\mathcal{S}||\mathcal{A}|} R_{\max}}{(1-\gamma)^3 d_{\min}^2 (1-\sigma_2(\boldsymbol{W}))}$ in Theorem 3.7 with $\epsilon$. We require

$$\alpha = \tilde{\mathcal{O}}\left(\min\left\{\frac{(1-\gamma)^5 d_{\min}^3}{R_{\max}^2 d_{\max}^2}\epsilon^2, \frac{(1-\gamma)^3 d_{\min}^2 (1-\sigma_2(\boldsymbol{W}))}{R_{\max} d_{\max}^2 \sqrt{|\mathcal{S}||\mathcal{A}|}}\epsilon\right\}\right).$$

Next, we bound the terms $(1 - \alpha(1-\gamma)d_{\min})^{\frac{k}{2}} + \sigma_2(\boldsymbol{W})^{\frac{k}{4}}$. Noting that

$$(1 - \alpha(1-\gamma)d_{\min})^{\frac{k}{2}} \le \exp\left(-\alpha(1-\gamma)d_{\min}\frac{k}{2}\right),$$

we require

$$k = \tilde{\mathcal{O}}\left(\frac{1}{(1-\gamma)d_{\min}\alpha}\ln\left(\frac{1}{\epsilon}\right) + \ln\left(\frac{1}{\epsilon}\right) / \ln\left(\frac{1}{\sigma_2(\boldsymbol{W})}\right)\right)$$

$$= \tilde{\mathcal{O}}\left(\ln\left(\frac{1}{\epsilon}\right)\max\left\{\frac{R_{\max}^2 d_{\max}^2}{\epsilon^2(1-\gamma)^6 d_{\min}^4}, \frac{R_{\max} d_{\max}^2 \sqrt{|\mathcal{S}||\mathcal{A}|}}{\epsilon(1-\gamma)^4 d_{\min}^3 (1-\sigma_2(\boldsymbol{W}))}\right\}\right).$$

This completes the proof. □

## G Appendix : Markovian observation model

In this section, we provide the analysis tools for the Markovian observation model in Section 4.

Considering a sequence of state-action trajectory $\{(s_k, \boldsymbol{a}_k)\}_{k\in\mathbb{N}}$ induced by the behavior policy $\beta$, the update of Q-function at time $k$ becomes

$$\boldsymbol{Q}_{k+1}^i(s_k, \boldsymbol{a}_k) = \sum_{j\in\mathcal{N}_i} [\boldsymbol{W}]_{ij}\boldsymbol{Q}_k^j(s_k, \boldsymbol{a}_k) + \alpha\left(r_{k+1}^i + \gamma\max_{\boldsymbol{a}\in\mathcal{A}}\boldsymbol{Q}_k^i(s_{k+1}, \boldsymbol{a}) - \boldsymbol{Q}_k^i(s_k, \boldsymbol{a}_k)\right)$$

$$\boldsymbol{Q}_{k+1}^i(s, \boldsymbol{a}) = \sum_{j\in\mathcal{N}_i} [\boldsymbol{W}]_{ij}\boldsymbol{Q}_k^j(s, \boldsymbol{a}), \quad s, \boldsymbol{a} \in \mathcal{S} \times \mathcal{A} \setminus \{(s_k, \boldsymbol{a}_k)\},$$

(20)

where we have replaced $s_k'$ in (2) with $s_{k+1}$. The overall algorithm is given in Algorithm 2 in the Appendix Section J.

We follow the same definitions in Section 3 by letting $\boldsymbol{D}$ to be $\boldsymbol{D}_\infty$. That is, we have

$$\boldsymbol{A_Q} = \boldsymbol{I} + \alpha\boldsymbol{D}_\infty(\gamma\boldsymbol{P}\boldsymbol{\Pi}^{\boldsymbol{Q}} - \boldsymbol{I}), \quad \boldsymbol{b_Q} = \gamma\boldsymbol{D}_\infty\boldsymbol{P}(\boldsymbol{\Pi}^{\boldsymbol{Q}} - \boldsymbol{\Pi}^{\boldsymbol{Q}^*})\boldsymbol{Q}^*,$$

which are defined in (10).

Furthermore, let us define for $\boldsymbol{Q} \in \mathbb{R}^{|\mathcal{S}||\mathcal{A}|}$, $\bar{\boldsymbol{Q}} \in \mathbb{R}^{N|\mathcal{S}||\mathcal{A}|}$, and $\bar{\boldsymbol{Q}}^i \in \mathbb{R}^{|\mathcal{S}||\mathcal{A}|}$ such that $[\boldsymbol{Q}^i]_j = [\bar{\boldsymbol{Q}}]_{|\mathcal{S}||\mathcal{A}|(i-1)+j}$ for $j \in [|\mathcal{S}||\mathcal{A}|]$:

$$\boldsymbol{\Delta}^{\mathrm{avg}}(\bar{\boldsymbol{Q}}) = \boldsymbol{D}_\infty \frac{1}{N}\sum_{i=1}^{N}\left(\boldsymbol{R}^i + \gamma\boldsymbol{P}\boldsymbol{\Pi}^{\boldsymbol{Q}^i}\boldsymbol{Q}^i - \boldsymbol{Q}^i\right),$$

$$\boldsymbol{\Delta}_{k-\tau,\tau}^{\mathrm{avg}}(\bar{\boldsymbol{Q}}) := \boldsymbol{D}_\tau^{s_{k-\tau}, \boldsymbol{a}_{k-\tau}} \frac{1}{N}\sum_{i=1}^{N}\left(\boldsymbol{R}^i + \gamma\boldsymbol{P}\boldsymbol{\Pi}^{\boldsymbol{Q}^i}\boldsymbol{Q}^i - \boldsymbol{Q}^i\right),$$

where $\boldsymbol{D}_\tau^{s_{k-\tau}, \boldsymbol{a}_{k-\tau}}$ is defined in (14).

Note that we did not use any property of the i.i.d. distribution in proving the consensus error. Therefore, we can directly use the result in Theorem 3.4 for the consensus error for Markovian observation model. Hence, in this section, we focus on bounding the optimality error, $\boldsymbol{Q}_k^{\mathrm{avg}} - \boldsymbol{Q}^*$. As in the case of i.i.d. observation model in Section 3, we will analyze the error bound of lower and upper comparison system in the subsequent sections.

### G.1 ANALYSIS OF OPTIMALITY ERROR UNDER MARKOVIAN OBSERVATION MODEL

As in Section 3.3, we will analyze the error bound for $\tilde{\boldsymbol{Q}}_k^{\mathrm{avg},u}$ and $\tilde{\boldsymbol{Q}}_k^{\mathrm{avg},l}$ to bound the optimality error, $\tilde{\boldsymbol{Q}}_k^{\mathrm{avg}}$. We will present an error bound on the lower comparison system, $\tilde{\boldsymbol{Q}}_k^{\mathrm{avg},l}$, in Proposition G.5, and the error bound on $\tilde{\boldsymbol{Q}}_k^{\mathrm{avg},u} - \tilde{\boldsymbol{Q}}_k^{\mathrm{avg},l}$ in Proposition G.6. Collecting the results, the result on the optimality error, $\tilde{\boldsymbol{Q}}_k^{\mathrm{avg}}$, will be presented in Theorem G.7.

Let us first investigate the lower comparison system. $\tilde{\boldsymbol{Q}}_k^{\mathrm{avg},l}$ evolves via (12) where we replace $\boldsymbol{\epsilon}_k^{\mathrm{avg}}$ with $\boldsymbol{\epsilon}^{\mathrm{avg}}(o_k, \bar{\boldsymbol{Q}}_k)$ where $o_k = (s_k, \boldsymbol{a}_k, s_{k+1})$. To analyze the error under Markovian observation model, we decompose the terms, for $k \geq \tau$ as follows:

$$
\begin{aligned}
\tilde{\boldsymbol{Q}}_{k+1}^{\mathrm{avg},l} =& \boldsymbol{A}_{\boldsymbol{Q}^*} \tilde{\boldsymbol{Q}}_k^{\mathrm{avg},l} + \alpha \boldsymbol{\epsilon}^{\mathrm{avg}}(o_k, \bar{\boldsymbol{Q}}_k) + \alpha \boldsymbol{E}_k \\
=& \boldsymbol{A}_{\boldsymbol{Q}^*} \tilde{\boldsymbol{Q}}_k^{\mathrm{avg},l} + \alpha \boldsymbol{\epsilon}^{\mathrm{avg}}(o_k, \bar{\boldsymbol{Q}}_{k-\tau}) + \alpha(\boldsymbol{\epsilon}^{\mathrm{avg}}(o_k, \bar{\boldsymbol{Q}}_k) - \boldsymbol{\epsilon}^{\mathrm{avg}}(o_k, \bar{\boldsymbol{Q}}_{k-\tau})) + \alpha \boldsymbol{E}_k \\
=& \boldsymbol{A}_{\boldsymbol{Q}^*} \tilde{\boldsymbol{Q}}_k^{\mathrm{avg},l} + \alpha \underbrace{(\boldsymbol{\delta}^{\mathrm{avg}}(o_k, \bar{\boldsymbol{Q}}_{k-\tau}) - \boldsymbol{\Delta}_{k-\tau,\tau}^{\mathrm{avg}}(\bar{\boldsymbol{Q}}_{k-\tau}))}_{:=\boldsymbol{w}_{k,1}} + \alpha \underbrace{(\boldsymbol{\Delta}_{k-\tau,\tau}^{\mathrm{avg}}(\bar{\boldsymbol{Q}}_{k-\tau}) - \boldsymbol{\Delta}^{\mathrm{avg}}(\bar{\boldsymbol{Q}}_{k-\tau}))}_{:=\boldsymbol{w}_{k,2}} \\
& + \alpha \underbrace{(\boldsymbol{\epsilon}^{\mathrm{avg}}(o_k, \bar{\boldsymbol{Q}}_k) - \boldsymbol{\epsilon}^{\mathrm{avg}}(o_k, \bar{\boldsymbol{Q}}_{k-\tau}))}_{:=\boldsymbol{w}_{k,3}} + \alpha \boldsymbol{E}_k.
\end{aligned}
\tag{21}
$$

The decomposition is motivated to invoke Azuma-Hoeffding inequality as explained in Section 4. Recursively expanding the terms in (21), we get

$$
\tilde{\boldsymbol{Q}}_{k+1}^{\mathrm{avg},l} = \boldsymbol{A}_{\boldsymbol{Q}^*}^{k-\tau+1} \tilde{\boldsymbol{Q}}_\tau^{\mathrm{avg},l} + \alpha \sum_{j=\tau}^k \boldsymbol{A}_{\boldsymbol{Q}^*}^{k-j} \boldsymbol{w}_{j,1} + \alpha \sum_{j=\tau}^k \boldsymbol{A}_{\boldsymbol{Q}^*}^{k-j} \boldsymbol{w}_{j,2} + \alpha \sum_{j=\tau}^k \boldsymbol{A}_{\boldsymbol{Q}^*}^{k-j} \boldsymbol{w}_{j,3} + \alpha \sum_{j=\tau}^k \boldsymbol{A}_{\boldsymbol{Q}^*}^{k-j} \boldsymbol{E}_j.
\tag{22}
$$

Now, let us provide an analysis on the lower comparison system.

We will provide the bounds of $\sum_{j=\tau}^k \boldsymbol{A}_{\boldsymbol{Q}^*}^{k-j} \boldsymbol{w}_{j,1}$, $\sum_{j=\tau}^k \boldsymbol{A}_{\boldsymbol{Q}^*}^{k-j} \boldsymbol{w}_{j,2}$, and $\sum_{j=\tau}^k \boldsymbol{A}_{\boldsymbol{Q}^*}^{k-j} \boldsymbol{w}_{j,3}$ in Lemma G.2, Lemma G.3, and Lemma G.4, respectively. We first provide an important property to bound $\sum_{j=\tau}^k \boldsymbol{A}_{\boldsymbol{Q}^*}^{k-j} \boldsymbol{w}_{j,1}$.

**Lemma G.1.** *For* $t \geq \tau$, *let* $\mathcal{F}_t := \sigma(\{\bar{\boldsymbol{Q}}_0, s_0, \boldsymbol{a}_0, s_1, \boldsymbol{a}_1, \ldots, s_t, \boldsymbol{a}_t\})$. *Then,*

$$
\mathbb{E}\left[\boldsymbol{w}_{t,1} | \mathcal{F}_{t-\tau}\right] = \boldsymbol{0}.
$$

*Proof.* We have

$$
\begin{aligned}
\mathbb{E}\left[\boldsymbol{w}_{t,1} | \mathcal{F}_{t-\tau}\right] =& \mathbb{E}\left[\boldsymbol{\delta}^{\mathrm{avg}}(o_k, \bar{\boldsymbol{Q}}_{k-\tau}) - \boldsymbol{\Delta}_{k-\tau,\tau}^{\mathrm{avg}}(\bar{\boldsymbol{Q}}_{k-\tau}) \Big| \mathcal{F}_{t-\tau}\right] \\
=& \frac{1}{N} \sum_{i=1}^N \mathbb{E}\left[(\boldsymbol{e}_{s_t} \otimes \boldsymbol{e}_{\boldsymbol{a}_t})(r_{t+1} + \boldsymbol{e}_{s_{t+1}}^\top \gamma \boldsymbol{\Pi}^{\boldsymbol{Q}_{t-\tau}^i} \boldsymbol{Q}_{t-\tau}^i - (\boldsymbol{e}_{s_t} \otimes \boldsymbol{e}_{\boldsymbol{a}_t})^\top \boldsymbol{Q}_{t-\tau}^i) \Big| \mathcal{F}_{t-\tau}\right] \\
& - \frac{1}{N} \boldsymbol{D}_\tau^{s_{t-\tau}, \boldsymbol{a}_{t-\tau}} \sum_{i=1}^N \left(\boldsymbol{R}^i + \gamma \boldsymbol{P} \boldsymbol{\Pi}^{\boldsymbol{Q}_{t-\tau}^i} - \boldsymbol{Q}_{t-\tau}^i\right) \\
=& \boldsymbol{0}.
\end{aligned}
$$

The second equality follows from the fact that $\boldsymbol{Q}_{t-\tau}^i$ is $\mathcal{F}_{t-\tau}$-measurable. This completes the proof. $\square$

**Lemma G.2.** *For* $k \in \mathbb{N}$, *and* $\alpha \leq \min\left\{\min_{i\in[N]}[\boldsymbol{W}]_{ii}, \frac{1}{2\tau}\right\}$, *we have*

$$
\mathbb{E}\left[\left\|\sum_{j=\tau}^k \boldsymbol{A}_{\boldsymbol{Q}^*}^{k-j} \boldsymbol{w}_{j,1}\right\|_\infty\right] \leq 2\sqrt{\ln(2\tau|\mathcal{S}||\mathcal{A}|)} \frac{15\sqrt{\tau} R_{\max}}{(1-\gamma)^{\frac{3}{2}} d_{\min}^{\frac{1}{2}} \alpha^{\frac{1}{2}}}.
$$

*Proof.* For $0 \leq q \leq \tau - 1$, let for $t \in \mathbb{N}$ such that $q \leq \tau t + q \leq k$:

$$\mathcal{F}_{k,t}^q := \mathcal{F}_{\tau t + q}.$$

Then, let us consider the sequence $\{\boldsymbol{S}_{k,t}^q\}_{t \in \{t \in \mathbb{N}: q \leq \tau t + q \leq k\}}$ as follows:

$$\boldsymbol{S}_{k,t}^q := \sum_{j=1}^{t} \boldsymbol{A}_{\boldsymbol{Q}^*}^{k - \tau j - q} \boldsymbol{w}_{\tau j + q, 1}.$$

Next, we will apply Azuma-Hoeffding inequality in Lemma C.4. Let us first check that $\{\boldsymbol{S}_{k,t}^q\}_{t \in \{t \in \mathbb{N}: \tau t + q \leq k\}}$ is a Martingale sequence. We can see that

$$\mathbb{E}\left[\boldsymbol{S}_{k,t}^q \middle| \mathcal{F}_{k,t-1}^q\right] = \mathbb{E}\left[\boldsymbol{A}_{\boldsymbol{Q}^*}^{k - \tau t - q} \boldsymbol{w}_{\tau t + q, 1} \middle| \mathcal{F}_{k,t-1}^q\right] + \mathbb{E}\left[\sum_{j=1}^{t-1} \boldsymbol{A}_{\boldsymbol{Q}^*}^{k - \tau j - q} \boldsymbol{w}_{\tau j + q, 1} \middle| \mathcal{F}_{k,t-1}^q\right]$$

$$= \boldsymbol{S}_{k,t-1}^q.$$

The second equality follows from Lemma G.1, and the fact that $\boldsymbol{S}_{k,t-1}^q$ is $\mathcal{F}_{k,t-1}^q$-measurable. Moreover, we have $\mathbb{E}\left[\boldsymbol{S}_{k,1}^q \middle| \mathcal{F}_q\right] = \boldsymbol{0}$, and

$$\left\|\boldsymbol{S}_{k,t}^q - \boldsymbol{S}_{k,t-1}^q\right\|_\infty = \left\|\boldsymbol{A}_{\boldsymbol{Q}^*}^{k - \tau t - q} \boldsymbol{w}_{\tau t + q, 1}\right\|_\infty \leq (1 - (1 - \gamma)d_{\min}\alpha)^{k - \tau t - q} \frac{4R_{\max}}{1 - \gamma},$$

where the last inequality follows from Lemma C.1. Now, we have, for $s, \boldsymbol{a} \in \mathcal{S} \times \mathcal{A}$,

$$\sum_{j \in \{t \in \mathbb{N}: q < \tau t + q \leq k\}} \left|[\boldsymbol{S}_{k,j}^q]_{s,\boldsymbol{a}} - [\boldsymbol{S}_{k,j-1}^q]_{s,\boldsymbol{a}}\right| \leq \sum_{j \in \{t \in \mathbb{N}: \tau t + q \leq k\}} (1 - (1 - \gamma)d_{\min}\alpha)^{2k - 2\tau j - 2q} \frac{16R_{\max}^2}{(1 - \gamma)^2}$$

$$\leq \frac{1}{(1 - (1 - (1 - \gamma)d_{\min}\alpha)^{2\tau})} \frac{16R_{\max}^2}{(1 - \gamma)^2}.$$

Therefore, we can now apply Azuman-Hoeffding inequality in Lemma C.4, which yields

$$\mathbb{P}\left[\left\|\boldsymbol{S}_{k,t^*(q)}^q\right\|_\infty \geq \epsilon\right] \leq 2|\mathcal{S}||\mathcal{A}| \exp\left(-\frac{\epsilon^2(1 - (1 - (1 - \gamma)d_{\min}\alpha)^{2\tau})}{2} \frac{(1 - \gamma)^2}{16R_{\max}^2}\right),$$

where $t^*(q) = \max\{t \in \mathbb{N}: \tau t + q \leq k\}$. Considering that

$$\cap_{q=0}^{\tau-1} \left\{\left\|\boldsymbol{S}_{k,t^*(q)}^q\right\|_\infty < \epsilon/\tau\right\} \subset \left\{\|\boldsymbol{S}_k\|_\infty < \epsilon\right\},$$

taking the union bound of the events,

$$\mathbb{P}\left[\|\boldsymbol{S}_k\|_\infty \geq \epsilon\right] \leq \min\left\{\sum_{0 \leq q \leq \tau - 1} \mathbb{P}\left[\left\|\boldsymbol{S}_{k,t^*(q)}^q\right\|_\infty \geq \epsilon/\tau\right], 1\right\}$$

$$\leq \min\left\{2\tau|\mathcal{S}||\mathcal{A}| \exp\left(-\frac{\epsilon^2(1 - (1 - (1 - \gamma)d_{\min}\alpha)^{2\tau})}{2\tau^2} \frac{(1 - \gamma)^2}{16R_{\max}^2}\right), 1\right\}.$$

Therefore, from Lemma C.5, we have

$$\mathbb{E}\left[\|\boldsymbol{S}_k\|_\infty\right] \leq 2\sqrt{\ln(2\tau|\mathcal{S}||\mathcal{A}|)} \frac{6\tau R_{\max}}{(1 - \gamma)\sqrt{(1 - (1 - (1 - \gamma)d_{\min}\alpha)^{2\tau})}}$$

$$\leq 2\sqrt{\ln(2\tau|\mathcal{S}||\mathcal{A}|)} \frac{6\tau R_{\max}}{(1 - \gamma)^{\frac{3}{2}} d_{\min}^{\frac{1}{2}} \alpha^{\frac{1}{2}} \sqrt{(\sum_{j=0}^{2\tau-1}(1 - (1 - \gamma)d_{\min}\alpha)^j}}$$

$$\leq 2\sqrt{\ln(2\tau|\mathcal{S}||\mathcal{A}|)} \frac{6\tau R_{\max}}{(1 - \gamma)^{\frac{3}{2}} d_{\min}^{\frac{1}{2}} \alpha^{\frac{1}{2}} \sqrt{2\tau(1 - (1 - \gamma)d_{\min}\alpha)^{2\tau-1}}}$$

$$\leq 2\sqrt{\ln(2\tau|\mathcal{S}||\mathcal{A}|)} \frac{5\sqrt{\tau} R_{\max}}{(1 - \gamma)^{\frac{3}{2}} d_{\min}^{\frac{1}{2}} \alpha^{\frac{1}{2}}} \exp((1 - \gamma)d_{\min}\alpha(2\tau - 1))$$

$$\leq 2\sqrt{\ln(2\tau|\mathcal{S}||\mathcal{A}|)} \frac{15\sqrt{\tau} R_{\max}}{(1 - \gamma)^{\frac{3}{2}} d_{\min}^{\frac{1}{2}} \alpha^{\frac{1}{2}}}.$$

The second inequality follows from $1 - x^{2\tau} = (1-x)(1 + x + x^2 + \cdots + x^{2\tau-1})$ for $x \in \mathbb{R}$. The third inequality follows from the fact that $\sum_{j=0}^{2\tau-1}(1-(1-\gamma)d_{\min}\alpha)^j \geq \sum_{j=0}^{2\tau-1}(1-(1-\gamma)d_{\min}\alpha)^{2\tau-1}$.

The second last inequality follows from the relation such that $\exp(-2x) \leq 1-x$ for $x \in [0, 0.75]$. The condition $\alpha \leq \frac{1}{2\tau}$ leads to $\exp((1-\gamma)d_{\min}\alpha(2\tau-1)) \leq 3$, yielding the last line. This completes the proof. $\square$

Now, we bound $\left\| \sum_{j=\tau}^k \boldsymbol{A}_{\boldsymbol{Q}^*}^{k-j} \boldsymbol{w}_{j,2} \right\|_\infty$.

**Lemma G.3.** *For $k \geq \tau$, we have*

$$\mathbb{E}\left[ \left\| \sum_{j=\tau}^k \boldsymbol{A}_{\boldsymbol{Q}^*}^{k-j} \boldsymbol{w}_{j,2} \right\|_\infty \right] \leq \frac{8R_{\max}}{(1-\gamma)^2 d_{\min}}.$$

*Proof.* Recalling the definition of $\boldsymbol{D}_\infty$ and $\boldsymbol{D}_\tau^{s_{j-\tau}, \boldsymbol{a}_{j-\tau}}$ in (14), we have

$$\begin{aligned}
\left\| \boldsymbol{D}_\infty - \boldsymbol{D}_\tau^{s_{j-\tau}, \boldsymbol{a}_{j-\tau}} \right\|_\infty &= \max_{s, \boldsymbol{a} \in \mathcal{S} \times \mathcal{A}} \left| [((\boldsymbol{e}_{s_{j-\tau}} \otimes \boldsymbol{e}_{\boldsymbol{a}_{j-\tau}})^\top \boldsymbol{P}^\tau)^\top]_{s,\boldsymbol{a}} - [\boldsymbol{\mu}_\infty]_{s,\boldsymbol{a}} \right| \\
&\leq 2 d_{\mathrm{TV}}(((\boldsymbol{e}_{s_{j-\tau}} \otimes \boldsymbol{e}_{\boldsymbol{a}_{j-\tau}})^\top \boldsymbol{P}^\tau)^\top, \boldsymbol{\mu}_\infty) \\
&\leq 2m\rho^\tau \\
&\leq 2\alpha.
\end{aligned}$$

The first inequality follows from the definition of the total variation distance, and the second and third inequalities follow from the definition of the mixing time in (13).

Now, we can see that

$$\begin{aligned}
\|\boldsymbol{w}_{j,2}\|_\infty &= \left\| (\boldsymbol{D} - \boldsymbol{D}_\tau^{s_{j-\tau}, \boldsymbol{a}_{j-\tau}}) \frac{1}{N} \sum_{i=1}^N \left( \boldsymbol{R}^i + \gamma \boldsymbol{P}\boldsymbol{\Pi}^{\boldsymbol{Q}_j^i} \boldsymbol{Q}_j^i - \boldsymbol{Q}_j^i \right) \right\|_\infty \\
&\leq \frac{1}{N} \left\| \boldsymbol{D} - \boldsymbol{D}_\tau^{s_{j-\tau}, \boldsymbol{a}_{j-\tau}} \right\|_\infty \left\| \sum_{i=1}^N \boldsymbol{R}^i + \gamma \boldsymbol{P}\boldsymbol{\Pi}^{\boldsymbol{Q}_j^i} \boldsymbol{Q}_j^i - \boldsymbol{Q}_j^i \right\|_\infty \\
&\leq \alpha \frac{8R_{\max}}{1-\gamma},
\end{aligned}$$

where the last inequality follows from Lemma 3.3.

Therefore, we have

$$\left\| \sum_{j=\tau}^k \boldsymbol{A}_{\boldsymbol{Q}^*}^{k-j} \boldsymbol{w}_{j,2} \right\|_\infty \leq \alpha \frac{8R_{\max}}{1-\gamma} \sum_{j=\tau}^k (1-\alpha(1-\gamma)d_{\min})^{k-j} \leq \frac{8R_{\max}}{(1-\gamma)^2 d_{\min}},$$

where the first inequality follows from Lemma C.1. This completes the proof. $\square$

**Lemma G.4.** *For $k \geq \tau$, we have*

$$\begin{aligned}
\left\| \sum_{j=\tau}^k \boldsymbol{A}_{\boldsymbol{Q}^*}^{k-j} \boldsymbol{w}_{j,3} \right\|_\infty &\leq 8 \left\| \bar{\boldsymbol{Q}}_0 \right\|_2 \left( \sigma_2(\boldsymbol{W})^{\frac{k-\tau}{2}} \frac{1}{(1-\gamma)d_{\min}\alpha} + (1-(1-\gamma)d_{\min}\alpha)^{\frac{k-\tau}{2}} \frac{1}{1-\sigma_2(\boldsymbol{W})} \right) \\
&\quad + \frac{64R_{\max}\sqrt{N|\mathcal{S}||\mathcal{A}|}}{(1-\gamma)^2 d_{\min}(1-\sigma_2(\boldsymbol{W}))} + 4\tau \frac{2R_{\max}}{(1-\gamma)^2 d_{\min}}.
\end{aligned}$$

*Proof.* Recalling the definition of $\boldsymbol{w}_{j,3}$ in (21), we get

$$
\begin{aligned}
\boldsymbol{w}_{j,3} =& \boldsymbol{\delta}^{\mathrm{avg}}(o_j, \bar{\boldsymbol{Q}}_j) - \boldsymbol{\delta}^{\mathrm{avg}}(o_j, \bar{\boldsymbol{Q}}_{j-\tau}) - \boldsymbol{\Delta}^{\mathrm{avg}}(\bar{\boldsymbol{Q}}_j) + \boldsymbol{\Delta}^{\mathrm{avg}}(\bar{\boldsymbol{Q}}_{j-\tau}) \\
=& \frac{1}{N} \sum_{i=1}^{N} \left( (\boldsymbol{e}_{s_j} \otimes \boldsymbol{e}_{\boldsymbol{a}_j}) \boldsymbol{e}_{s_{j+1}}^{\top} \gamma \left( \boldsymbol{\Pi}^{\boldsymbol{Q}_j^i} \boldsymbol{Q}_j^i - \boldsymbol{\Pi}^{\boldsymbol{Q}_{j-\tau}^i} \boldsymbol{Q}_{j-\tau}^i \right) - (\boldsymbol{e}_{s_j} \otimes \boldsymbol{e}_{\boldsymbol{a}_j})(\boldsymbol{e}_{s_j} \otimes \boldsymbol{e}_{\boldsymbol{a}_j})^{\top} (\boldsymbol{Q}_j^i - \boldsymbol{Q}_{j-\tau}^i) \right) \\
&+ \boldsymbol{D}_{\infty} \frac{1}{N} \sum_{i=1}^{N} \left( \gamma \boldsymbol{P} \boldsymbol{\Pi}^{\boldsymbol{Q}_j^i} \boldsymbol{Q}_j^i - \gamma \boldsymbol{P} \boldsymbol{\Pi}^{\boldsymbol{Q}_{j-\tau}^i} \boldsymbol{Q}_{j-\tau}^i + \boldsymbol{Q}_j^i - \boldsymbol{Q}_{j-\tau}^i \right).
\end{aligned}
$$

Taking infinity norm, we get

$$
\begin{aligned}
\|\boldsymbol{w}_{j,3}\|_{\infty} \leq& \frac{1}{N} \sum_{i=1}^{N} 2 \left\| \boldsymbol{Q}_j^i - \boldsymbol{Q}_{j-\tau}^i \right\|_{\infty} + \frac{d_{\max}}{N} \sum_{i=1}^{N} 2 \left\| \boldsymbol{Q}_j^i - \boldsymbol{Q}_{j-\tau}^i \right\|_{\infty} \\
\leq& \frac{4}{N} \sum_{i=1}^{N} \left( \left\| \boldsymbol{Q}_j^i - \boldsymbol{Q}_j^{\mathrm{avg}} \right\|_{\infty} + \left\| \boldsymbol{Q}_j^{\mathrm{avg}} - \boldsymbol{Q}_{j-\tau}^{\mathrm{avg}} \right\|_{\infty} + \left\| \boldsymbol{Q}_{j-\tau}^{\mathrm{avg}} - \boldsymbol{Q}_{j-\tau}^i \right\|_{\infty} \right) \\
\leq& 4 \left\| \boldsymbol{\Theta} \bar{\boldsymbol{Q}}_j \right\|_{\infty} + 4 \left\| \boldsymbol{\Theta} \bar{\boldsymbol{Q}}_{j-\tau} \right\|_{\infty} + 4 \left\| \boldsymbol{Q}_j^{\mathrm{avg}} - \boldsymbol{Q}_{j-\tau}^{\mathrm{avg}} \right\|_{\infty}.
\end{aligned}
\tag{23}
$$

The first inequality follows from the non-expansive property of max-operator. The second inequality follows from the triangle inequality. The term $\left\| \boldsymbol{Q}_j^{\mathrm{avg}} - \boldsymbol{Q}_{j-\tau}^{\mathrm{avg}} \right\|_{\infty}$ can be bounded as follows:

$$
\begin{aligned}
\left\| \boldsymbol{Q}_j^{\mathrm{avg}} - \boldsymbol{Q}_{j-\tau}^{\mathrm{avg}} \right\|_{\infty} \leq& \sum_{t=j-\tau}^{j-1} \left\| \boldsymbol{Q}_{t+1}^{\mathrm{avg}} - \boldsymbol{Q}_t^{\mathrm{avg}} \right\|_{\infty} \\
\leq& \alpha \sum_{t=j-\tau}^{j-1} \frac{1}{N} \sum_{i=1}^{N} \left\| \boldsymbol{e}_{s_t, \boldsymbol{a}_t} \left( r_t^i + \gamma \max_{\boldsymbol{a} \in \mathcal{A}} \boldsymbol{Q}_t^i(s_{t+1}, \boldsymbol{a}) - \boldsymbol{Q}_t^i(s_t, \boldsymbol{a}_t) \right) \right\|_{\infty} \\
\leq& \alpha \tau \frac{2 R_{\max}}{1 - \gamma}.
\end{aligned}
\tag{24}
$$

The second inequality follows from (2). The last inequality follows from Lemma 3.3.

Applying the result in Theorem 3.4 together with (24) to (23), we get

$$
\|\boldsymbol{w}_{j,3}\|_{\infty} \leq 8 \sigma_2(\boldsymbol{W})^{j-\tau} \left\| \bar{\boldsymbol{Q}}_0 \right\|_2 + 8\alpha \frac{8 R_{\max}}{1 - \gamma} \frac{\sqrt{N |\mathcal{S}| |\mathcal{A}|}}{1 - \sigma_2(\boldsymbol{W})} + 4 \alpha \tau \frac{2 R_{\max}}{1 - \gamma}.
\tag{25}
$$

Now, we are ready to derive our desired statement:

$$
\begin{aligned}
& \left\| \sum_{j=\tau}^{k} \boldsymbol{A}_{\boldsymbol{Q}^*}^{k-j} \boldsymbol{w}_{j,3} \right\|_{\infty} \\
\leq& \sum_{j=\tau}^{k} (1 - (1-\gamma) d_{\min} \alpha)^{k-j} \left( 8 \sigma_2(\boldsymbol{W})^{j-\tau} \left\| \bar{\boldsymbol{Q}}_0 \right\|_2 + 8\alpha \frac{8 R_{\max}}{1 - \gamma} \frac{\sqrt{N |\mathcal{S}| |\mathcal{A}|}}{1 - \sigma_2(\boldsymbol{W})} + 4 \alpha \tau \frac{2 R_{\max}}{1 - \gamma} \right) \\
\leq& 8 \left\| \bar{\boldsymbol{Q}}_0 \right\|_2 \left( \sigma_2(\boldsymbol{W})^{\frac{k-\tau}{2}} \frac{1}{(1-\gamma) d_{\min} \alpha} + (1 - (1-\gamma) d_{\min} \alpha)^{\frac{k-\tau}{2}} \frac{1}{1 - \sigma_2(\boldsymbol{W})} \right) \\
& + \frac{64 R_{\max} \sqrt{N |\mathcal{S}| |\mathcal{A}|}}{(1-\gamma)^2 d_{\min} (1 - \sigma_2(\boldsymbol{W}))} + 4\tau \frac{2 R_{\max}}{(1-\gamma)^2 d_{\min}}.
\end{aligned}
$$

The first inequality follows from Lemma C.1 and (25). The last inequality follows from Lemma C.3. This completes the proof. $\qquad \square$

Now, collecting the results we have the following bound for the lower comparison system:

**Proposition G.5.** *For $k \in \mathbb{N}$, and $\alpha \leq \min\left\{\min_{i \in [N]}[\boldsymbol{W}]_{ii}, \frac{1}{2\tau}\right\}$, we have*

$$\mathbb{E}\left[\left\|\tilde{\boldsymbol{Q}}_{k+1}^{\mathrm{avg},l}\right\|_\infty\right] = \tilde{\mathcal{O}}\left((1-(1-\gamma)d_{\min}\alpha)^{\frac{k-\tau}{2}} + \sigma_2(\boldsymbol{W})^{\frac{k-\tau}{2}}\right)$$
$$+ \tilde{\mathcal{O}}\left(\alpha^{\frac{1}{2}}\frac{\sqrt{\tau}R_{\max}}{(1-\gamma)^{\frac{3}{2}}d_{\min}^{\frac{1}{2}}} + \alpha\frac{R_{\max}\sqrt{N|\mathcal{S}||\mathcal{A}|}}{(1-\gamma)^2 d_{\min}(1-\sigma_2(\boldsymbol{W}))}\right).$$

*Proof.* Collecting the results in Lemma G.2, Lemma G.3, Lemma G.4, and Lemma 3.5, we can bound (22) as follows:

$$\mathbb{E}\left[\left\|\tilde{\boldsymbol{Q}}_{k+1}^{\mathrm{avg},l}\right\|_\infty\right]$$
$$\leq (1-(1-\gamma)d_{\min}\alpha)^{k-\tau+1}\mathbb{E}\left[\left\|\tilde{\boldsymbol{Q}}_\tau^{\mathrm{avg},l}\right\|_\infty\right]$$
$$+ 2\alpha^{\frac{1}{2}}\sqrt{\ln(2\tau|\mathcal{S}||\mathcal{A}|)}\frac{15\sqrt{\tau}R_{\max}}{(1-\gamma)^{\frac{3}{2}}d_{\min}^{\frac{1}{2}}}$$
$$+ \alpha\frac{8R_{\max}}{(1-\gamma)^2 d_{\min}}$$
$$+ 8\left\|\bar{\boldsymbol{Q}}_0\right\|_2\left(\sigma_2(\boldsymbol{W})^{\frac{k-\tau}{2}}\frac{1}{(1-\gamma)d_{\min}} + (1-(1-\gamma)d_{\min}\alpha)^{\frac{k-\tau}{2}}\frac{\alpha}{1-\sigma_2(\boldsymbol{W})}\right)$$
$$+ \alpha\frac{64R_{\max}\sqrt{N|\mathcal{S}||\mathcal{A}|}}{(1-\gamma)^2 d_{\min}(1-\sigma_2(\boldsymbol{W}))} + 4\alpha\tau\frac{2R_{\max}}{(1-\gamma)^2 d_{\min}}$$
$$+ \gamma d_{\max}\left\|\boldsymbol{\Theta}\bar{\boldsymbol{Q}}_0\right\|_2\left((1-(1-\gamma)d_{\min}\alpha)^{\frac{k-\tau}{2}}\frac{\alpha}{1-\sigma_2(\boldsymbol{W})} + \sigma_2(\boldsymbol{W})^{\frac{k-\tau}{2}}\frac{1}{(1-\gamma)d_{\min}}\right)$$
$$+ \alpha\gamma d_{\max}\frac{8R_{\max}\sqrt{N|\mathcal{S}||\mathcal{A}|}}{(1-\gamma)^2 d_{\min}(1-\sigma_2(\boldsymbol{W}))}.$$

That is,

$$\mathbb{E}\left[\left\|\tilde{\boldsymbol{Q}}_{k+1}^{\mathrm{avg},l}\right\|_\infty\right] = \tilde{\mathcal{O}}\left((1-(1-\gamma)d_{\min}\alpha)^{\frac{k-\tau}{2}} + \sigma_2(\boldsymbol{W})^{\frac{k-\tau}{2}}\right)$$
$$+ \tilde{\mathcal{O}}\left(\alpha^{\frac{1}{2}}\frac{\sqrt{\tau}R_{\max}}{(1-\gamma)^{\frac{3}{2}}d_{\min}^{\frac{1}{2}}} + \alpha\frac{R_{\max}\sqrt{N|\mathcal{S}||\mathcal{A}|}}{(1-\gamma)^2 d_{\min}(1-\sigma_2(\boldsymbol{W}))}\right).$$

This completes the proof. $\qquad\square$

The rest of the proof follows the same logic in Section 3. We consider the upper comparison system, and derive the convergence rate of $\boldsymbol{Q}_k^{\mathrm{avg},u} - \boldsymbol{Q}_k^{\mathrm{avg},l}$. As can be seen in (18), if we subtract $\boldsymbol{Q}_{k+1}^{\mathrm{avg},l}$ from $\boldsymbol{Q}_{k+1}^{\mathrm{avg},u}$, $\boldsymbol{\epsilon}_k^{\mathrm{avg}}$ and $\boldsymbol{E}_k$ are eliminated. Therefore, we can follow the same lines of the proof in Proposition F.3:

**Proposition G.6.** *For $k \in \mathbb{N}$, and $\alpha \leq \min\left\{\min_{i \in [N]}[\boldsymbol{W}]_{ii}, \frac{1}{2\tau}\right\}$, we have*

$$\mathbb{E}\left[\left\|\boldsymbol{Q}_{k+1}^{\mathrm{avg},u} - \boldsymbol{Q}_{k+1}^{\mathrm{avg},l}\right\|_\infty\right] = \tilde{\mathcal{O}}\left((1-\alpha(1-\gamma)d_{\min})^{\frac{k-\tau}{2}} + \sigma_2(\boldsymbol{W})^{\frac{k-\tau}{4}}\right)$$
$$+ \tilde{\mathcal{O}}\left(\alpha^{\frac{1}{2}}d_{\max}\frac{\sqrt{\tau}R_{\max}}{(1-\gamma)^{\frac{5}{2}}d_{\min}^{\frac{3}{2}}} + \alpha\frac{d_{\max}R_{\max}\sqrt{N|\mathcal{S}||\mathcal{A}|}}{(1-\gamma)^3 d_{\min}^2(1-\sigma_2(\boldsymbol{W}))}\right).$$

*Proof.* As from the proof of Proposition F.3, we have

$$\mathbb{E}\left[\left\|\boldsymbol{Q}_{k+1}^{\mathrm{avg},u} - \boldsymbol{Q}_{k+1}^{\mathrm{avg},l}\right\|_\infty\right] \leq (1-\alpha(1-\gamma)d_{\min})^{k-\tau+1}\mathbb{E}\left[\left\|\boldsymbol{Q}_\tau^{\mathrm{avg},u} - \boldsymbol{Q}_\tau^{\mathrm{avg},l}\right\|_\infty\right]$$
$$+ 2\alpha\gamma d_{\max}\underbrace{\sum_{i=\tau}^k (1-\alpha(1-\gamma)d_{\min})^{k-i}\mathbb{E}\left[\left\|\tilde{\boldsymbol{Q}}_i^{\mathrm{avg},l}\right\|_\infty\right]}_{(\star)}. \qquad (26)$$

We will use Proposition G.5 to bound $(\star)$ in the above inequality. We have

$$\sum_{i=\tau}^{k}(1-\alpha(1-\gamma)d_{\min})^{k-i}\mathbb{E}\left[\left\|\tilde{\boldsymbol{Q}}_i^{\mathrm{avg},l}\right\|_\infty\right]$$

$$=\tilde{\mathcal{O}}\left(\sum_{i=\tau}^{k}(1-\alpha(1-\gamma)d_{\min})^{k-\frac{i+\tau}{2}}+(1-\alpha(1-\gamma)d_{\min})^{k-i}\sigma_2(\boldsymbol{W})^{\frac{k-\tau}{2}}\right)$$

$$+\tilde{\mathcal{O}}\left(\alpha^{-\frac{1}{2}}\frac{\sqrt{\tau}R_{\max}}{(1-\gamma)^{\frac{5}{2}}d_{\min}^{\frac{3}{2}}}+\frac{R_{\max}\sqrt{N|\mathcal{S}||\mathcal{A}|}}{(1-\gamma)^3 d_{\min}^2(1-\sigma_2(\boldsymbol{W}))}\right)$$

$$=\tilde{\mathcal{O}}\left((1-\alpha(1-\gamma)d_{\min})^{\frac{k-\tau}{2}}+\sigma_2(\boldsymbol{W})^{\frac{k-\tau}{4}}\right)$$

$$+\tilde{\mathcal{O}}\left(\alpha^{-\frac{1}{2}}\frac{\sqrt{\tau}R_{\max}}{(1-\gamma)^{\frac{5}{2}}d_{\min}^{\frac{3}{2}}}+\frac{R_{\max}\sqrt{N|\mathcal{S}||\mathcal{A}|}}{(1-\gamma)^3 d_{\min}^2(1-\sigma_2(\boldsymbol{W}))}\right).$$

The last inequality follows from Lemma C.3. Applying this result to (26), we get

$$\mathbb{E}\left[\left\|\boldsymbol{Q}_{k+1}^{\mathrm{avg},u}-\boldsymbol{Q}_{k+1}^{\mathrm{avg},l}\right\|_\infty\right]=\tilde{\mathcal{O}}\left((1-\alpha(1-\gamma)d_{\min})^{\frac{k-\tau}{2}}+\sigma_2(\boldsymbol{W})^{\frac{k-\tau}{4}}\right)$$

$$+\tilde{\mathcal{O}}\left(\alpha^{\frac{1}{2}}d_{\max}\frac{\sqrt{\tau}R_{\max}}{(1-\gamma)^{\frac{5}{2}}d_{\min}^{\frac{3}{2}}}+\alpha\frac{d_{\max}R_{\max}\sqrt{N|\mathcal{S}||\mathcal{A}|}}{(1-\gamma)^3 d_{\min}^2(1-\sigma_2(\boldsymbol{W}))}\right).$$

This completes the proof. $\qquad\square$

Now, we are ready to provide the optimality error under Markovian observation model:

**Theorem G.7.** *For $k\geq\tau$, and $\alpha\leq\min\left\{\min_{i\in[N]}[\boldsymbol{W}]_{ii},\frac{1}{2\tau}\right\}$, we have*

$$\mathbb{E}\left[\|\boldsymbol{Q}_k^{\mathrm{avg}}-\boldsymbol{Q}^*\|_\infty\right]=\tilde{\mathcal{O}}\left((1-\alpha(1-\gamma)d_{\min})^{\frac{k-\tau}{2}}+\sigma_2(\boldsymbol{W})^{\frac{k-\tau}{4}}\right)$$

$$+\tilde{\mathcal{O}}\left(\alpha^{\frac{1}{2}}d_{\max}\frac{\sqrt{\tau}R_{\max}}{(1-\gamma)^{\frac{5}{2}}d_{\min}^{\frac{3}{2}}}+\alpha\frac{R_{\max}d_{\max}\sqrt{N|\mathcal{S}||\mathcal{A}|}}{(1-\gamma)^3 d_{\min}^2(1-\sigma_2(\boldsymbol{W}))}\right).$$

*Proof.* The proof follows the same logic as in Theorem 3.6 using the fact that $\tilde{\boldsymbol{Q}}_k^{\mathrm{avg},l}\leq\tilde{\boldsymbol{Q}}_k^{\mathrm{avg}}\leq\tilde{\boldsymbol{Q}}_k^{\mathrm{avg},u}$. Therefore, we omit the proof. $\qquad\square$

### G.2 PROOF OF THEOREM 4.1

*Proof.* The proof follows the same line as in Theorem 3.7. From Theorem 3.4 and Theorem G.7, we get

$$\mathbb{E}\left[\left\|\bar{\boldsymbol{Q}}_k-\mathbf{1}_N\otimes\boldsymbol{Q}^*\right\|_\infty\right]\leq\mathbb{E}\left[\left\|\bar{\boldsymbol{Q}}_k-\mathbf{1}_N\otimes\boldsymbol{Q}_k^{\mathrm{avg}}\right\|_\infty\right]+\mathbb{E}\left[\|\boldsymbol{Q}_k^{\mathrm{avg}}-\boldsymbol{Q}^*\|_\infty\right]$$

$$=\tilde{\mathcal{O}}\left(\sigma_2(\boldsymbol{W})^k+\alpha\frac{R_{\max}\sqrt{N|\mathcal{S}||\mathcal{A}|}}{(1-\gamma)(1-\sigma_2(\boldsymbol{W}))}\right)$$

$$+\tilde{\mathcal{O}}\left((1-\alpha(1-\gamma)d_{\min})^{\frac{k-\tau}{2}}+\sigma_2(\boldsymbol{W})^{\frac{k-\tau}{4}}\right)$$

$$+\tilde{\mathcal{O}}\left(\alpha^{\frac{1}{2}}d_{\max}\frac{\sqrt{\tau}R_{\max}}{(1-\gamma)^{\frac{5}{2}}d_{\min}^{\frac{3}{2}}}+\alpha\frac{R_{\max}d_{\max}\sqrt{N|\mathcal{S}||\mathcal{A}|}}{(1-\gamma)^3 d_{\min}^2(1-\sigma_2(\boldsymbol{W}))}\right)$$

$$=\tilde{\mathcal{O}}\left((1-\alpha(1-\gamma)d_{\min})^{\frac{k-\tau}{2}}+\sigma_2(\boldsymbol{W})^{\frac{k-\tau}{4}}\right)$$

$$+\tilde{\mathcal{O}}\left(\alpha^{\frac{1}{2}}\frac{d_{\max}\sqrt{\tau}R_{\max}}{(1-\gamma)^{\frac{5}{2}}d_{\min}^{\frac{3}{2}}}+\alpha\frac{R_{\max}d_{\max}\sqrt{|\mathcal{S}||\mathcal{A}|}}{(1-\gamma)^3 d_{\min}^2(1-\sigma_2(\boldsymbol{W}))}\right).$$

This completes the proof. $\qquad\square$

### G.3 Proof of Corollary 4.2

*Proof.* For $\mathbb{E}\left[\left\|\bar{\boldsymbol{Q}}_k - \mathbf{1}_N \otimes \boldsymbol{Q}^*\right\|_\infty\right] \leq \epsilon$, we bound the each terms in Theorem 4.1 with $\frac{\epsilon}{4}$. We require

$$\alpha^{\frac{1}{2}} d_{\max} \frac{\sqrt{\tau} R_{\max}}{(1-\gamma)^{\frac{5}{2}} d_{\min}^{\frac{3}{2}}} \leq \epsilon/4,$$

which is satisfied if

$$\alpha = \tilde{\mathcal{O}}\left(\frac{\epsilon^2}{\ln\left(\frac{1}{\epsilon^2}\right)} \frac{(1-\gamma)^5 d_{\min}^3}{t_{\mathrm{mix}} d_{\max}^2}\right),$$

where $\tau$ is bounded by $t_{\mathrm{mix}}$ by Lemma C.7 in the Appendix. Likewise, bounding $\alpha \frac{R_{\max} d_{\max} \sqrt{|\mathcal{S}||\mathcal{A}|}}{(1-\gamma)^3 d_{\min}^2 (1-\sigma_2(\boldsymbol{W}))} \leq \epsilon/4$, together with the above condition, we require

$$\alpha = \tilde{\mathcal{O}}\left(\min\left\{\frac{\epsilon^2}{\ln\left(\frac{1}{\epsilon^2}\right)} \frac{(1-\gamma)^5 d_{\min}^3}{d_{\max}^2 t_{\mathrm{mix}}}, \frac{\epsilon(1-\gamma)^3 d_{\min}^2 (1-\sigma_2(\boldsymbol{W}))}{d_{\max}\sqrt{|\mathcal{S}||\mathcal{A}|}}\right\}\right).$$

Furthermore bounding the terms $(1 - \alpha(1-\gamma)d_{\min})^{\frac{k-\tau}{2}} + \sigma_2(\boldsymbol{W})^{\frac{k-\tau}{4}}$ in Theorem 4.1 with $\frac{\epsilon}{4}$, respectively, we require,

$$k \geq \tilde{\mathcal{O}}\left(\min\left\{\frac{\ln^2\left(\frac{1}{\epsilon^2}\right)}{\epsilon^2} \frac{t_{\mathrm{mix}} d_{\max}^2}{(1-\gamma)^6 d_{\min}^4}, \frac{\ln\left(\frac{1}{\epsilon}\right)}{\epsilon} \frac{d_{\max}\sqrt{|\mathcal{S}||\mathcal{A}|}}{(1-\gamma)^4 d_{\min}^3 (1-\sigma_2(\boldsymbol{W}))}\right\} + \ln\left(\frac{1}{\epsilon}\right) \Big/ \ln\left(\frac{1}{\sigma_2(\boldsymbol{W})}\right)\right).$$

This completes the proof. $\qquad\square$

## H Appendix : Examples mentioned in Section 5

Let us provide an example where the condition (15) used in Heredia et al. (2020) is not met in tabular MDP. Since the condition only depends on the state-action distribution, consider an MDP that consists of two states and single action, where $\mathcal{S} := \{1, 2\}$ and $\mathcal{A} := \{1\}$ with $d(1,1) = 0.1$, $d(2,1) = 0.9$, and $\gamma = 0.5$ Then, $d_{\min} = 0.1$ and $d_{\max} = 0.9$, then $d_{\min} < \gamma^2 d_{\max}$ which contradicts the condition in (15).

Next, we provide an MDP where the condition (16) required in Zeng et al. (2022b) is not met:

$$\boldsymbol{P} = \begin{bmatrix} 1 & 0 \\ 0 & 1 \\ 1 & 0 \\ 0 & 1 \end{bmatrix}, \quad \boldsymbol{R} = \begin{bmatrix} 0 \\ 0.1 \\ 0 \\ 0.1 \end{bmatrix}, \quad [\boldsymbol{D}]_{s,a} = \frac{1}{4}, \; \forall s, a \in \mathcal{S} \times \mathcal{A}.$$

Letting $\gamma = 0.99$, we can check that $\boldsymbol{Q}^* = \begin{bmatrix} 9.9 \\ 10 \\ 9.9 \\ 10 \end{bmatrix}$ and $\boldsymbol{\Pi}^{\boldsymbol{Q}^*} = \begin{bmatrix} 0 & 1 & 0 & 0 \\ 0 & 0 & 0 & 1 \end{bmatrix}$. Consider

$\boldsymbol{Q} = \begin{bmatrix} 12 \\ 10 \\ 11 \\ 10 \end{bmatrix}$. Then, we have

$$(\gamma \boldsymbol{D}\boldsymbol{P}(\boldsymbol{\Pi}^{\boldsymbol{Q}}\boldsymbol{Q} - \boldsymbol{\Pi}^{\boldsymbol{Q}^*}\boldsymbol{Q}^*) - \boldsymbol{D}(\boldsymbol{Q} - \boldsymbol{Q}^*))^\top (\boldsymbol{Q} - \boldsymbol{Q}^*) = 0.179,$$

which is contradiction to the condition in (16).

## I Experiments

The experiment used the MDP where and $|\mathcal{A}_i| = 2$ for each agent $i \in [N]$ where $N$ denotes the number of agents. For each run, we have randomly generated the transition and reward

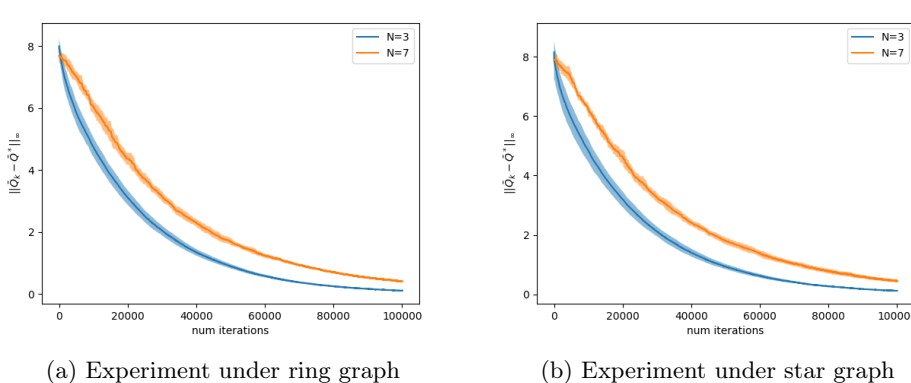

(a) Experiment under ring graph      (b) Experiment under star graph

Figure 1: $\alpha = 0.1$. The result was averaged over five runs.

matrix. Each elements were chosen uniformly random between zero and one, and for the transition matrix, each row is normalized to be a probability distribution. We can see that the distributed Q-learning algorithm converges to close to $Q^*$, where the constant bias is induced by using the constant step-size. As number of agents increase, the convergence rate becomes slower.

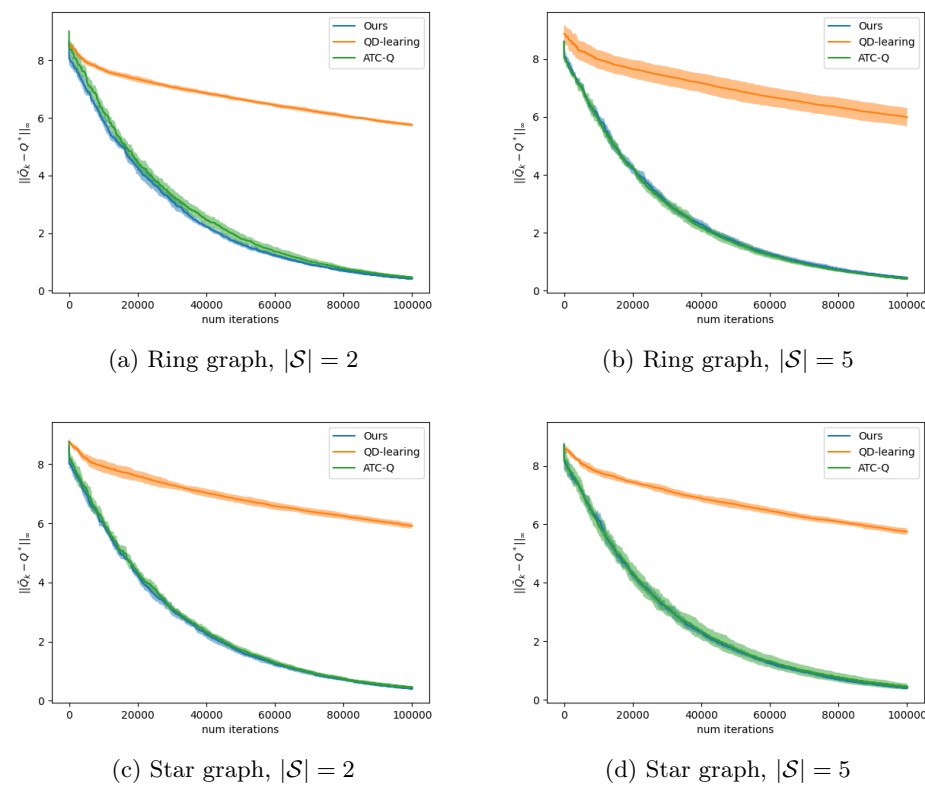

(a) Ring graph, $|\mathcal{S}| = 2$      (b) Ring graph, $|\mathcal{S}| = 5$

(c) Star graph, $|\mathcal{S}| = 2$      (d) Star graph, $|\mathcal{S}| = 5$

Figure 2: $\alpha = 0.1$. The result was averaged over five runs and $N = 7$

The Figure 2 shows comparison with QD-learning developed in Kar et al. (2013). QD-learning uses a two-time scale approach, and therefore we have set the two-step-sizes as 0.1 and 0.0.1, where the faster time-scale matches the single-step-size of distributed Q-learning. As in

the figure, distributed Q-learning shows faster convergence rate compared to QD-learning. ATC-Q refers to the adapt-then-combine scheme in Wang et al. (2022).

## J Appendix : Pseudo code

---

**Algorithm 1** Distributed Q-learning : i.i.d. observation model

---

**Require:** Initialize $\boldsymbol{Q}_0^i \in \mathbb{R}^{|\mathcal{S}||\mathcal{A}|}$ such that $||\boldsymbol{Q}_0^i|| \leq \frac{R_{\max}}{1-\gamma}$ for all $i \in [N]$, and $0 \leq \alpha \leq$
$\min_{i \in [N]}[\boldsymbol{W}]_{ii}$.
   **for** $k = 0, 1, \ldots,$ **do**
      Observe $s_k, \boldsymbol{a}_k \sim d(\cdot, \cdot), s_k' \sim \mathcal{P}(s_k, \boldsymbol{a}_k, \cdot)$.
      **for** $i = 1, 2, \ldots, N$ **do**
         Update as follows:

$$\boldsymbol{Q}_{k+1}^i(s_k, \boldsymbol{a}_k) = \sum_{j \in \mathcal{N}_i} [\boldsymbol{W}]_{ij} \boldsymbol{Q}_k^j(s_k, \boldsymbol{a}_k) + \alpha \left( r_{k+1}^i + \gamma \max_{\boldsymbol{a} \in \mathcal{A}} \boldsymbol{Q}_k^i(s_k', \boldsymbol{a}) - \boldsymbol{Q}_k^i(s_k, \boldsymbol{a}_k) \right).$$

      **end for**
   **end for**

---

---

**Algorithm 2** Distributed Q-learning : Markovian observation model

---

**Require:** Initialize $\boldsymbol{Q}_0^i \in \mathbb{R}^{|\mathcal{S}||\mathcal{A}|}$ such that $||\boldsymbol{Q}_0^i|| \leq \frac{R_{\max}}{1-\gamma}$ for all $i \in [N]$, and $0 \leq \alpha \leq$
$\min \left\{ \min_{i \in [N]}[\boldsymbol{W}]_{ii}, \frac{1}{2\tau} \right\}$.
   Observe $s_0, \boldsymbol{a}_0 \sim \boldsymbol{\mu}_0$.
   **for** $k = 0, 1, \ldots,$ **do**
      Observe $s_{k+1} \sim \mathcal{P}(s_k, \boldsymbol{a}_k, \cdot)$ and $\boldsymbol{a}_{k+1} \sim \beta(\cdot \mid s_k)$.
      **for** $i = 1, 2, \ldots, N$ **do**
         Update as follows:

$$\boldsymbol{Q}_{k+1}^i(s_k, \boldsymbol{a}_k) = \sum_{j \in \mathcal{N}_i} [\boldsymbol{W}]_{ij} \boldsymbol{Q}_k^j(s_k, \boldsymbol{a}_k) + \alpha \left( r_{k+1}^i + \gamma \max_{\boldsymbol{a} \in \mathcal{A}} \boldsymbol{Q}_k^i(s_{k+1}, \boldsymbol{a}) - \boldsymbol{Q}_k^i(s_k, \boldsymbol{a}_k) \right).$$

      **end for**
   **end for**

---

