# OpenReview forum: "A Finite-Time Analysis of Distributed Q-Learning"
_ICLR.cc/2025/Conference — Submitted to ICLR 2025_

### Official Review · Reviewer_Gtxi · 2024-10-15

**Soundness:** 3
**Presentation:** 2
**Contribution:** 2
**Rating:** 5
**Confidence:** 4

**Summary:**

This paper studies distributed Q learning, where the agent communicate through a graph. The finite-time convergence under both iid and Markovian samples are studied.

**Strengths:**

The problem of distributed Q learning is important and interesting. The convergence results also are SOTA.

**Weaknesses:**

1. Although the notations are defined mathematically, it can be more clear with additional explaniation. For example, in lines 225-227, it would be godd to explan what $\delta, \Delta$ are.
2. The introduction of the switched system looks odd to me. If it is only for proof technique, might should introduced in the proof part.
3. The proof technique is standard by following single agent Q-learning, and analysis for multi agent system is also studied. For example, a quick search results in a study [1], where robust multi-agent Q learning is studing. The non-robust is a special case with $R=0$. Hence it make me feel the novelty here is incremental.
4. Analysis of Markovian vs iid is also not new.



[1] Wang, Yudan, et al. "Data-driven robust multi-agent reinforcement learning." 2022 IEEE 32nd International Workshop on Machine Learning for Signal Processing (MLSP). IEEE, 2022.

**Questions:**

What is the novelty compared to existing works?

Can you address the weakness part?

---

> ### Author Response · Authors · 2024-11-18
> **Response (1/2)**
>
> **Q1.** *Although the notations are defined mathematically, it can be more clear with additional explanation. For example, in lines 225-227, it would be godd to explan what are $\delta,\Delta$.*
>
> **A1.** We thank the reviewer for the constructive suggestions. We have updated the manuscript with additional explanations.
>
>
> **Q2.** *The introduction of the switched system looks odd to me. If it is only for proof technique, might should introduced in the proof part.*
>
> **A2.**  We would like to thank the reviewer for the thoughtful feedback. Although we agree with the reviewer, we would like to note that the adoption of switched system analysis of single-agent case [2] to the multi-agent case is one of our key contribution. The key challenge in extending from single-agent to multi-agent cases can be best explained in this context. For this reason, we have chosen to formally present it in an independent section. In the following, we will elaborate on the key challenge and discuss how it can be overcome within this framework. The key difference from the single-agent case is that we take average of the maximum of Q-function of each agent, i.e., the term $\frac{1}{N}\sum_{i=1}^N\Pi^{Q^i\_k}Q^i\_k$ in equation (9) in the manuscript, rather than the maximum of average of Q-function of each agents, i.e., $\Pi^{Q^{avg}\_k}Q^{avg}\_k$. This poses difficulty in the analysis since $\frac{1}{N}\sum_{i=1}^N\Pi^{Q^i\_k}Q^i\_k$ cannot be represented in terms of $Q^{avg}\_k$. Consequently, it makes difficult to interpret it as switched affine system whose state-variable is $Q^{avg}\_k$. We introduce an additional error term $\frac{1}{N}\sum_{i=1}^N\ \Pi^{Q^i\_k}Q^i\_k-\Pi^{Q^{avg}\_k}Q^{avg}\_k$, and this allows us to follow the switched system analysis. The number of agents are chosen as seven. Following the reviewer's comment, the related discussions have been highlighted in the revision in order to further clarify this contribution.
>
> **Q3.** *The proof technique is standard by following single agent Q-learning, and analysis for multi agent system is also studied. For example, a quick search results in a study [1], where robust multi-agent Q learning is studying. The non-robust is a special case with . Hence it make me feel the novelty here is incremental.*
>
>
> **A3.** We thank the reviewer for highlighting a closely related work. The study in [1] examines a similar algorithm, though with some differences. Our algorithm is inspired by the combine-and-adapt scheme, whereas the authors in [1] focus on a adapt-and-combine scheme [3]. The combine-and-adapt scheme first aggregates information from the neighbours and then executes update while the adapt-and-combine scheme [3], the update occurs before aggregation. Most importantly, the results in [1] are valid only for bounded $\epsilon$, i.e., $\epsilon\in \left[ 0,\frac{1}{1-\gamma} \right)$, while our results do not have such restriction. Following the reviewer's comments, we have incorporated this in the revised version of the manuscript.
>
>
> **Q4.** *Analysis of Markovian vs iid is also not new.*
>
>
> **A4.** Thank you for the valuable comment. We would like to first note that our focus is not on comparing Markovian and i.i.d observation models. Instead, we would like to mention that the i.i.d case is provided only for conceptual ideas for the proof. According to the reviewer's comment, we have clarified this in the revised manuscript.

---

> > ### Author Response · Authors · 2024-11-18
> > **Response (2/2)**
> >
> > **Q5.** What is the novelty compared to existing works?
> >
> >
> > **A5.** We thank the reviewer for the valuable comments. The novelty of our work lies in the following:
> >
> > 1) In our paper, we derive, for the first time, the finite-time analysis of QD-learning in its original form, which is one of the most fundamental and widely used distributed Q-learning methods. While several works have addressed other types of distributed Q-learning, the analysis of QD-learning has remained unexplored until now. We believe that analyzing the original QD-learning is a valuable topic in its own right. Moreover, we established a new theoretical result for distributed Q-learning, addressing the problem under more relaxed assumptions compared to previous approaches [Heredia et al., 2020; Zeng et al., 2022b]. a) Unlike the previous works, we adopted the switched system analysis of Q-learning, which provides additional insights and tools. b) The previous works relied on stronger assumptions that allows their analysis to follow similar lines to that of convex optimization literature. To the best of our knowledge, there is no existing literature that demonstrates how to extend convex optimization analysis, or an analogous approach, to the analysis of Q-learning under the tabular setup. As a result, neither the findings of existing works can be directly applied to our context, nor can they be generalized. c) Furthermore, compared to the work of Wang et al., our bound holds for any $\epsilon >0$ whereas the bound in Wang et al. holds for restricted range of $\epsilon$, i.e., $\epsilon\in \left[0,\frac{1}{1-\gamma} \right)$, while in our approach, such a restriction is not required. More importantly, the algorithm proposed by Wang et al. requires two steps for a single update, whereas in our paper, we focus on a one-step algorithm that is algorithmically simpler and more efficient. Specifically, we analyze the traditional and widely adopted QD-learning algorithm proposed in [4], for which a finite-time error analysis has been lacking in the literature. Additionally, we enhance the efficiency of QD-learning by employing a constant step-size, as opposed to the two-time-scale decaying step-size used in traditional QD-learning. This modification can significantly improve the convergence speed empirically.
> >
> >
> > 2) We developed a new approach for the analysis of distributed Q-learning. We agree that switched system analysis of Q-learning has been studied in the literature before for single-agent scenario. However, we would like to note that the extension from the single-agent to multi-agent is non-trivial, as it required overcoming significant challenges, which in turn led to the development of new concepts, tools, and insights. In particular, the approach developed in the current manuscript significantly differs from those in the recent existing papers. This is because our contribution lies in using the decomposition of dynamics of optimality error in equation (9) in the manuscript, which allows us to apply the techniques from the switched system analysis of Q-learning. In particular, the key difference from the single-agent case is that we take average of the maximum of Q-function of each agent, i.e., the term $\frac{1}{N}\sum_{i=1}^N\Pi^{Q^i\_k}Q^i\_k$ in equation (9) in the manuscript, rather than the maximum of average of Q-function of each agents, .i.e., $\Pi^{Q^{avg}\_k}Q^{avg}\_k$. This poses difficulty in the analysis since $\frac{1}{N}\sum_{i=1}^N\Pi^{Q^i\_k}Q^i\_k$ cannot be represented in terms of $Q^{avg}\_k$. Consequently, it makes difficult to interpret it as switched affine system whose state-variable is $Q^{avg}\_k$. We introduce an additional error term $\frac{1}{N}\sum_{i=1}^N \Pi^{Q^i\_k}Q^i\_k-\Pi^{Q^{avg}\_k}Q^{avg}\_k$, and this allows us to follow the switched system analysis.
> >
> > Following the reviewer's comment, we will incorporate the discussion in the revised manuscript. We again thank the reviewer for the constructive comments and engagement in the discussion.
> >
> >
> > **References**
> >
> > [1] Wang, Yudan, et al. "Data-driven robust multi-agent reinforcement learning." 2022 IEEE 32nd International Workshop on Machine Learning for Signal Processing (MLSP). IEEE, 2022.
> >
> > [2] D. Lee, J. Hu, and N. He. A discrete-time switching system analysis of q-learning. SIAM Journal
> > on Control and Optimization, 61(3):1861–1880, 2023.
> >
> > [3] Chen, Jianshu, and Ali H. Sayed. "Diffusion adaptation strategies for distributed optimization and learning over networks." IEEE Transactions on Signal Processing 60.8 (2012): 4289-4305.

---

> > > ### Comment · Reviewer_Gtxi · 2024-11-20
> > >
> > > Thank you for your response. I now see the difference between this work and previous one, and one of the major difference is illustrated in eq 9. I will keep this in mind in the discussion phase.

---

> ### Author Response · Authors · 2024-11-22
>
> We sincerely thank the reviewer for the time and effort in evaluating our manuscript. If there are any further concerns, please feel free to inform us as the discussion period is nearing its end.

---

### Official Review · Reviewer_zSXF · 2024-10-30

**Soundness:** 3
**Presentation:** 2
**Contribution:** 3
**Rating:** 5
**Confidence:** 3

**Summary:**

This paper provides the first sample complexity bounds for distributed Q-learning scenario in tabular setting. It is a topic of increasing interest and importance given the empirical success achieved in applications of single-agent reinforcement learning.

**Strengths:**

It is the first paper that provides the sample complexity bounds for tabular distributed Q-learning without relying on any strong assumptions.

**Weaknesses:**

1.	In Corollary 3.8 and Corollary 4.2, the restrictions regarding the step size alpha are missing. These restrictions are used in the proof but are not mentioned in these two corollaries. Additionally, the restriction on $\alpha$ in line 1194 of the proof for Corollary 3.8 should be proportional to $\epsilon$ rather than its reciprocal.
2.	Instead of comparing the dependencies on graph structure, I recommend a comparison of the entire sample complexity bounds with the other two baselines.
3.	The experiment design is not very clear. For example, how are the transition functions and rewards generated? What is the number of agents? Why do you choose to compare your method with QD-learning? Moreover, $S = 2$ seems small and it is better to try some larger-scale experiments.
4.	There are some undefined notations and typos in the paper:
- The agent collection $V$ and the discount factor $\gamma$ are not defined at the beginning of the preliminary section.
- In the Bellman equation, $s$ in the maximum term should be $s’$.
- In line 220, the dimensions of the two matrices are incorrect.
- In line 270, $R_{max}$ is not defined.
- In line 436, it should be noted that $\rho < 1$.
- In line 1006, the content exceeds the page limitation.

**Questions:**

Your multi-agent MDP setting differs from the conventional MAMDP model, as you assume the agents share a common trajectory. Is this setting applicable in real-world scenarios, and are there other papers employing a similar setting?

---

> ### Author Response · Authors · 2024-11-18
> **Response**
>
> **Q1.** *In Corollary 3.8 and Corollary 4.2, the restrictions regarding the step size alpha are missing. These restrictions are used in the proof but are not mentioned in these two corollaries. Additionally, the restriction on in line 1194 of the proof for Corollary 3.8 should be proportional to rather than its reciprocal.*
>
> **A1.** Thank you for carefully investigating our manuscript. Following the reviewer's comment, we have corrected the issues in the revised manuscript.
>
> **Q2.** *Instead of comparing the dependencies on graph structure, I recommend a comparison of the entire sample complexity bounds with the other two baselines.*
>
> **A2.** Thank you for the valuable comments. We fully agree with the reviewer. We would like to note that due the different assumptions than the other two baselines, the sample complexity results are indeed not directly comparable. Nonetheless, we also agree with the reviewer's comment that the sample complexity comparison will give additional insights. Therefore, we have revised the comparison in Table 1 as the reviewer suggested.
>
> **Q3.** *The experiment design is not very clear. For example, how are the transition functions and rewards generated? What is the number of agents? Why do you choose to compare your method with QD-learning? Moreover, seems small and it is better to try some larger-scale experiments.*
>
> **A3.** We appreciate the reviewer's valuable comments and agree with the reviewer. We chose comparison among distributed Q-learning algorithms which are guaranteed to converge in the tabular setup, and, in our opinion, only a fair and reasonable comparison baseline is QD-learning. Moreover, the transition function was chosen uniformly random between zero and one and then normalized to be a probability distribution. The reward function was also generated uniformly random between zero and one. Following the reviewer's comment, we have clarified this in the revised manuscript.
>
> **Q4.** *There are some undefined notations and typos in the paper:*
>
> **A4.** We thank the reviewer for carefully investigating our manuscript. We have corrected the errors in the revised manuscript.
>
> **Q5.** *Your multi-agent MDP setting differs from the conventional MAMDP model, as you assume the agents share a common trajectory. Is this setting applicable in real-world scenarios, and are there other papers employing a similar setting?*
>
> **A5.** We thank the reviewer for the valuable comments. A large body of literature in theoretical development of distributed learning scheme in reinforcement learning relied on the same setting as ours, e.g., [1,2,3], to name a few. The topic has long been an active area of theoretical research, as evidenced by many papers, though practical and industrial applications remain limited in our opinion. However, one can easily envision intriguing applications. For example, consider a scenario where multiple RL agents collaborate to learn a policy while avoiding the sharing of their samples and/or rewards due to privacy or security concerns. Following the reviewer’s comment, we will incorporate these discussions in the revised version.
>
> **References**
>
> [1] Doan, S. Maguluri, and J. Romberg. Finite-time analysis of distributed td (0) with linear function approximation on multi-agent reinforcement learning. In International Conference on Machine Learning, pages 1626–1635. PMLR, 2019
>
> [2] Sun, G. Wang, G. B. Giannakis, Q. Yang, and Z. Yang. Finite-time analysis of decentralized temporal-difference learning with linear function approximation. In International Conference on Artificial Intelligence and Statistics, pages 4485–4495. PMLR, 2020.
>
> [3] K. Zhang, Z. Yang, H. Liu, T. Zhang, and T. Basar. Fully decentralized multi-agent reinforcement learning with networked agents. In International Conference on Machine Learning, pages 5872–5881. PMLR, 2018b.
>
> [4] Lowe, Ryan, et al. "Multi-agent actor-critic for mixed cooperative-competitive environments." Advances in neural information processing systems 30 (2017).

---

> ### Author Response · Authors · 2024-11-22
>
> We sincerely thank the reviewer for the time and effort in evaluating our manuscript. If there are any further concerns, please feel free to inform us as the discussion period is nearing its end.

---

> > ### Author Response · Authors · 2024-11-27
> >
> > We sincerely appreciate for the valuable feedback and insightful suggestions, which have helped us to improve our manuscript. We have carefully addressed all the raised concerns in our response.
> >
> > We would be grateful if Reviewer zSXF could provide additional feedback, as the reviewer's feedback is essential to enhancing the quality and clarity of our work. Alternatively, if our responses have satisfactorily addressed the concerns, we kindly request reconsideration of the score based on the clarifications and improvements made.

---

> > > ### Comment · Reviewer_zSXF · 2024-11-27
> > > **Additional questions**
> > >
> > > Thanks to the authors for their reply. The issues mentioned in my review are fixed. I have another question regarding the paper (Wang et al) mentioned in the general response. According the authors' description,  it seems that Wang et al reaches a sharper rate under the same setting. However, the differences and advantages provided are not convincing enough. For G1, it is very natural to assume such a mildly restricted range. For G2, the claimed simplicity and efficiency are not quantified. Moreover, as shown in the updated figures of your revised manuscript, the numerical performances between these two works are similar. It would be better to provide more direct evidences on the differences: e.g. difference in assumptions (settings), difference in computational complexity or communication cost.

---

> > > > ### Comment · Reviewer_Gtxi · 2024-11-27
> > > >
> > > > I agree with this. The region of epsilon is very natural, as the value functions are upper bounded by such a value. And the technique novelty and differences between the two scheme are unclear.

---

> > > > > ### Author Response · Authors · 2024-11-28
> > > > >
> > > > > We appreciate the reviewer’s constructive and valuable feedback. To begin with, our algorithm is based on the combine-then-adapt (CTA) scheme, which is also known as decentralized gradient descent in the distributed optimization literature. On the contrary, Wang et al. adopts the so-called adapt-then-combine (ATC) scheme. The simplicity of the CTA and its advantage over ATC is described in the following:
> > > > >
> > > > > The update of CTA scheme is as follows: $\bar{Q}\_{k+1}\leftarrow \bar{W}\bar{Q}_k+\alpha\cdot \rm{local\\;TD\\;error}$. In CTA scheme, the combining step ($\bar{W}\bar{Q}\_k$) and the computation of TD error can be done simultaneously. Meanwhile, ATC scheme first updates its local parameter $\psi_k\leftarrow \bar{Q}\_k+\alpha \cdot \rm{local\\;TD\\;error}$ and then combines the information $\bar{Q}\_{k+1}\leftarrow \bar{W}\psi\_k$. In the ATC scheme, these steps need to be done in separate steps, i.e., in a sequential manner. Therefore in an idealistic computing setting, the computation time of CTA will be faster than ATC if parallel computing is allowed.
> > > > >
> > > > > The CTA scheme has served as a foundational algorithm for advancing distributed optimization methods and widely used in distributed RL literature including QD-learning or distributed TD-learning algorithms. Consequently, we believe that providing a finite-time analysis based on the CTA scheme is both important and significant.

---

### Official Review · Reviewer_4VWU · 2024-11-01

**Soundness:** 3
**Presentation:** 2
**Contribution:** 2
**Rating:** 5
**Confidence:** 2

**Summary:**

In this paper, the authors present a finite-time analysis of distributed Q-Learning, where the agents need to cooperatively maximize an average of local rewards. Specifically, the agents do not have access to any global information and can only communicate with neighboring agents during the learning process. They show that a distributed Q-learning algorithm has a $\tilde{\mathcal{O}}\left(\max \left\\{\frac{1}{\epsilon^2} \frac{t_{\text{mix}}}{(1-\gamma)^6 d_{\min }^4}, \frac{1}{\epsilon} \frac{\sqrt{|\mathcal{S}||\mathcal{A}|}}{(1-\sigma_2(\mathbf{W}))(1-\gamma)^4 d_{\text {min }}^3}\right\\}\right)$ sample complexity bound.

**Strengths:**

* The paper is the first to give sample complexity bounds for the distributed Q-learning algorithm without restrictive assumptions.

* The results are technically sound.

**Weaknesses:**

* The technical contribution of this paper is unclear. It appears that the results are obtained by accommodating the analysis technique in [1] to the distributed setting. I suggest the authors elaborate on their technical contribution in the intro.

* The sample complexity bounds are of the order $\tilde{\mathcal{O}}\left(\max \left\\{\frac{t_{\text {mix }}}{\epsilon^2} \frac{|\mathcal{S}|^2 |\mathcal{A}|^{2}}{(1-\gamma)^6}, \frac{1}{\epsilon} \frac{|\mathcal{S}|^{\frac{5}{2}}  |\mathcal{A}|^{\frac{5}{2}}}{(1-\gamma)^4\left(1-\sigma_2(\mathbf{W})\right)}\right\\}\right)$. This bound is significantly larger compared to existing results in single-agent and federated RL. The authors should discuss whether this large bound arises from a loose analysis or the inherent nature of the problem setting. It may also help to compare the results with results on federated RL.

* The main body of the paper is written in an immense technical style. I suggest the authors try to make the proof outlines more accessible and discuss the implications of their theoretical results.


Since I am not very familiar with the field of MARL, I choose to assign a low confidence score to my evaluation.

[1] D. Lee and N. He. A unified switching system perspective and convergence analysis of q-learning algorithms. Advances in Neural Information Processing Systems, 33:15556–15567, 2020.

**Questions:**

* What would happen if we choose matrix $W$ to let all agents be able to communicate with each other? Would the problem setting degenerate into a centralized case? Is a better theoretical guarantee possible?

* Except for the case in Appendix B, are there any other choices of $W$ in common examples of practical applications? What is the scale of $\sigma_2(W)$, respectively?

---

> ### Author Response · Authors · 2024-11-18
> **Response (1/2)**
>
> **Q1.** *The technical contribution of this paper is unclear. It appears that the results are obtained by accommodating the analysis technique in [1] to the distributed setting. I suggest the authors elaborate on their technical contribution in the intro.*
>
>
> **A1.**  Thank you for pointing out important insights on the paper. In our paper, we derive, for the first time, the finite-time analysis of QD-learning in its original form, which is one of the most fundamental and widely used distributed Q-learning methods. While several works have addressed other types of distributed Q-learning, the analysis of QD-learning has remained unexplored until now.
>
> Moreover, we would like to emphasize that extending from the single-agent to the multi-agent setting is non-trivial, as it required overcoming significant challenges. This process led to the development of new concepts, tools, and insights. In particular, the key difference from the single-agent case is that we take average of the maximum of Q-function of each agent, i.e., the term $\frac{1}{N}\sum_{i=1}^N\Pi^{Q^i\_k}Q^i\_k$ in equation~(9) in the manuscript, rather than the maximum of average of Q-function of each agents, .i.e., $\Pi^{Q^{avg}\_k}Q^{avg}\_k$. This poses difficulty in the analysis since $\frac{1}{N}\sum_{i=1}^N\Pi^{Q^i\_k}Q^i\_k$ cannot be represented in terms of $Q^{avg}\_k$. Consequently, it makes difficult to interpret it as switched affine system whose state-variable is $Q^{avg}\_k$. We introduce an additional error term $\frac{1}{N}\sum_{i=1}^N\Pi^{Q^i\_k}Q^i\_k-\Pi^{Q^{avg}\_k}Q^{avg}\_k$, and this allows us to follow the switched system analysis. Following the reviewer's comment, we have highlighted the contribution part in the revised manuscript. Moreover, the related discussions has been newly added in the revision in order to further clarify this contribution.
>
>
> **Q2.** *The sample complexity bounds are of the order .... This bound is significantly larger compared to existing results in single-agent and federated RL. The authors should discuss whether this large bound arises from a loose analysis or the inherent nature of the problem setting. It may also help to compare the results with results on federated RL.*
>
> **A2.** We thank the reviewer for the insightful comments. We agree with the reviewer and would like to emphasize that significantly different analytical tools are required for the multi-agent case. Furthermore, it is natural that the bound for the multi-agent case is larger than that for the single-agent case. We note that the tight lower bound for the sample complexity has not yet been proven for the distributed Q-learning scenario. Identifying the tightness of the bound would require significantly more analysis and falls outside the scope of this paper. However, these are intriguing directions for future research, and we intend to explore them in subsequent studies. We appreciate the reviewer for pointing out these interesting topics. In response to the reviewer’s comments, we have revised the manuscript to emphasize this discussion.
>
> Despite these limitations, we believe our work represents a significant advancement in the field, as discussed in our previous responses regarding the contribution of this paper.
>
> Meanwhile, federated RL differs from the distributed learning scenario in two key aspects: it employs a centralized controller, and all agents share a common reward function. The distributed learning scenario does not employ a centralized controller and all the agents have their own local reward, which makes the problem more difficult. Therefore, the sample complexity result is not directly comparable, and should be worse than the that of federated RL. We have highlighted the discussion in the revised manuscript.
>
>
> **Q3.** *The main body of the paper is written in an immense technical style. I suggest the authors try to make the proof outlines more accessible and discuss the implications of their theoretical results.*
>
> **A3.** We fully agree and thank the reviewer for constructive comments. Following the reviewer's comment, the sketch of proof is provided in Section 3(i.i.d. observation part) and the implications are mostly summarized in Section 4(Markovian observation part). We have clarified this in the revised the manuscript as the reviewer suggested.
>
> **Q4.** *What would happen if we choose matrix to let all agents be able to communicate with each other? Would the problem setting degenerate into a centralized case? Is a better theoretical guarantee possible?*
>
> **A4.** We thank the reviewer for providing insightful comments. As the reviewer mentioned, if all the agents are connected, then every agents are being synchornozied, which is ``slightly different from the setting of centrzlied training'' where one centralized coordinator exists. Then, $\sigma_2(W)$ will be zero which makes the overall sample complexity smaller. Following the reviewer's comment, the related discussions will be newly added in the revision.

---

> > ### Author Response · Authors · 2024-11-18
> > **Response (2/2)**
> >
> > **Q5.** *Except for the case in Appendix B, are there any other choices of in common examples of practical applications? What is the scale of $\sigma_2(W)$ respectively?*
> >
> > **A5.** Thank you for pointing out insightful comments for our paper. One can formulate a semi-definite program to construct a doubly stochastic matrix [1].  It finds the doubly stochastic matrix with minimum possible $\sigma_2(W)$ but it requires a centralized controller to solve such system, and distributed the computed the result of each agents. Another choice is to use Sinkhorn-Knopp algorithm [2]. However, it also requires a centralized computation scheme. Moreover, to our best knowledge, we are not aware of bound on the $\sigma_2(W)$ of the output of Sinkhorn-Knopp algorithm. ccording to the reviewer's comment, we have included the discussion in the revised manuscript.
> >
> > **References**
> >
> > [1] Xiao, Lin, and Stephen Boyd. "Fast linear iterations for distributed averaging." Systems \& Control Letters 53.1 (2004): 65-78.
> >
> > [2] Knight, Philip A. "The Sinkhorn–Knopp algorithm: convergence and applications." SIAM Journal on Matrix Analysis and Applications 30.1 (2008): 261-275.

---

> ### Author Response · Authors · 2024-11-22
>
> We sincerely thank the reviewer for the time and effort in evaluating our manuscript. If there are any further concerns, please feel free to inform us as the discussion period is nearing its end.

---

> > ### Comment · Reviewer_4VWU · 2024-11-22
> >
> > I want to thank the authors for their responses. I will finalize my rating after the discussion with other reviewers.

---

### Official Review · Reviewer_3Rf8 · 2024-11-04

**Soundness:** 3
**Presentation:** 2
**Contribution:** 2
**Rating:** 5
**Confidence:** 3

**Summary:**

This work explores multi-agent reinforcement learning, where multiple agents collaboratively learn their individual policies in order to maximize the total cumulative rewards of all agents. They introduce a distributed Q-learning algorithm in which each agent updates its Q-estimates by aggregating the Q-values of its neighbors, weighted by a specified weight matrix. Furthermore, they provide a finite-time convergence analysis of the proposed algorithm in a tabular setting under mild assumptions.

**Strengths:**

1. The paper presents a finite-time convergence analysis under relaxed conditions.

**Weaknesses:**

1. The sample complexity presented is significantly distant from the tightest bound for the single-agent case in \cite{1}. In particular, considering that \( d_{\text{min}} \le \frac{1}{SA} \), the dependence on \( \frac{1}{(d_{\text{min}})^3} \) and \( \frac{1}{(d_{\text{min}})^4} \) seems critical. It remains uncertain whether the reported sample complexity adequately reflects the impact of key parameters on the learning process.
2. While the analysis is presented under milder conditions compared to the baselines, it is conducted in a tabular setting, which is simpler than those of the baselines. Therefore, it remains unclear whether the relaxation of assumptions in the simpler setting shows sufficient technical novelty.
3. It lacks a comparison with other decentralized RL algorithms, such as [2], which could provide better insights into the unique dependencies observed in decentralized scenarios and the unique challenges present in multi-agent settings. While this paper addresses a more general scenario where agents' actions can influence the rewards of others, it appears that this setting can encompass decentralized reinforcement learning with environmental heterogeneity, where agents aim to maximize average rewards, which seems more relevant to this work than the single-agent case.

[1] Li, C. Cai, Y. Chen, Y. Wei, and Y. Chi. Is q-learning minimax optimal? a tight sample complexity analysis. Operations Research, 72(1):222–236, 2024. \
[2] Tong Yang, Shicong Cen, Yuting Wei, Yuxin Chen, and Yuejie Chi. Federated natural policy gradient methods for multi-task reinforcement learning. arXiv preprint arXiv:2311.00201, 2023. \

**Questions:**

1. I wonder if  \frac{1}{(d_{\text{min}})^4} also appeared in the sample complexity bound in previous works [3,4].
2. Could you provide intuitions or simple examples that explain what enables the relaxation of the assumptions? Additionally, I am curious whether this relaxation is specific to the tabular setting.

[3] P. Heredia, H. Ghadialy, and S. Mou. Finite-sample analysis of distributed q-learning for multi-agent networks. In 2020 American Control Conference (ACC), pages 3511–3516. IEEE, 2020. \
[4] Finite-time convergence rates of decentralized stochastic approximation with applications in multi-agent and multi-task learning.

---

> ### Author Response · Authors · 2024-11-18
> **Response (1/2)**
>
> **Q1.**  *The sample complexity presented is significantly distant from the tightest bound for the single-agent case in [1]. In particular, considering that $d_{\min} \le \frac{1}{SA} $, the dependence on $ \frac{1}{(d_{\min})^3}  and ( \frac{1}{(d_{\min})^4} $ seems critical. It remains uncertain whether the reported sample complexity adequately reflects the impact of key parameters on the learning process.*
>
> **A1.** We agree with the reviewer and thank the reviewer for the valuable comments.
> We would like to emphasize that significantly different analytical tools are required for the multi-agent case. Furthermore, it is natural that the bound for the multi-agent case is larger than that for the single-agent case. While the proposed bound is not smoothly connected to the single-agent case when $N=1$, we believe this does not necessarily indicate that the bound is loose for the multi-agent case. However, we acknowledge the reviewer's observation that the bound could indeed be loose, and its optimality, as well as its relationship to the single-agent case, remains unclear at this stage. These are intriguing directions for future research, which we intend to explore in subsequent studies.
>
> Despite these limitations, we believe our work represents a significant advancement in the field. In response to the reviewer's comments, we have revised the manuscript to further elaborate on and emphasize this discussion.
>
> **Q2.** *While the analysis is presented under milder conditions compared to the baselines, it is conducted in a tabular setting, which is simpler than those of the baselines. Therefore, it remains unclear whether the relaxation of assumptions in the simpler setting shows sufficient technical novelty.*
>
> **A2.**  Thank you for providing insightful comments. As mentioned by the reviewer, we adopt the tabular representation whereas the baselines [3,4] use linear function approximation. However, the key difference is that they assume strong assumptions which do not hold even in the tabular case. Such examples are provided in Appendix Section H in the manuscript. The imposed assumptions play key role in their analysis. On the other hand, our analysis does not require such assumptions.
>
> In details, depending on the assumption, the overall analysis differs substantially. The assumptions used in [3,4] allows the analysis to follow similar lines to that of convex optimization literature. To the best of our knowledge, there is no existing literature that demonstrates how to extend convex optimization analysis, or an analogous approach, to the analysis of Q-learning under the tabular setup. This gap in the literature makes the analysis of Q-learning challenging and is the primary reason we rely on switched system analysis. Following the reviewer's comment, we have incorporated this discussion in the revised manuscript to clarify our approach and its relation to existing work.
>
> **Q3.** *It lacks a comparison with other decentralized RL algorithms, such as [2], which could provide better insights into the unique dependencies observed in decentralized scenarios and the unique challenges present in multi-agent settings. While this paper addresses a more general scenario where agents' actions can influence the rewards of others, it appears that this setting can encompass decentralized reinforcement learning with environmental heterogeneity, where agents aim to maximize average rewards, which seems more relevant to this work than the single-agent case.*
>
> **A3.** We thank the reviewer for sharing relevant works related to ours. The setting of [2] is the same as ours, where the agents share the same state and transition, while the reward function is kept private. Nonetheless, [2] considers a policy gradient approach whereas we consider Q-learning, which are substantially different algorithms. Following the reviewer's comments, we have included the comparison with [2] in the revised version.
>
> **Q4.** *I wonder if $\frac{1}{d_{\min}^4}$ also appeared in the sample complexity bound in previous works [3,4].*
>
> **A4.** We thank the reviewer for the valuable comments. We would like to first note that due to different settings, our sample complexity is not directly comparable with [3] or [4]. [3] considers a continuous-state space, and [4] uses a monotone condition assumes a monotonicity condition, $(\gamma DP(\Pi^Q Q-  \Pi^{Q^*}Q^*))^{\top}(Q-Q^*) \leq -\alpha || Q-Q^*||^2$, which makes the dependency on $d_{\min}$ implicit in their final bound of $\frac{1}{\alpha^2}$. Following the reviewer's comment, the related discussions on the comparison will be added in the revised manuscript.

---

> > ### Author Response · Authors · 2024-11-18
> > **Response (2/2)**
> >
> > **Q5.** *Could you provide intuitions or simple examples that explain what enables the relaxation of the assumptions? Additionally, I am curious whether this relaxation is specific to the tabular setting.*
> >
> > **A5.** Thank you for the constructive comments. In the tabular setting, the Bellman operator is a contraction. However, when we use linear function approximation, the composition of projection map and Bellman operator is no more a contractive operator. As a result, the works [3,4], which uses linear function approximation, impose a monotonicity condition or somewhat similar condition to overcome this huddle. However, this assumption does not hold even in the tabular setting. Consequently, this assumption makes the problem easier and allows to follow the spirit of convex optimization literature. In contrast, our approach extends the switched system analysis of single-agent Q-learning to the multi-agent case, and do not require such a restrictive assumption. We also acknowledge that our analysis cannot be extended to the case of linear function approximation, as Q-learning is known to diverge in certain counterexamples without restrictive assumptions. However, with modifications to Q-learning, such as regularized Q-learning, it is possible to develop convergent multi-agent Q-learning algorithms. Exploring these modifications presents an interesting avenue for future research. Following the reviewer's comment, we have included the discussions in the revised version.
> >
> > **References**
> >
> > [1] Li, C. Cai, Y. Chen, Y. Wei, and Y. Chi. Is q-learning minimax optimal? a tight sample complexity analysis. Operations Research, 72(1):222–236, 2024.
> >
> > [2] Tong Yang, Shicong Cen, Yuting Wei, Yuxin Chen, and Yuejie Chi. Federated natural policy gradient methods for multi-task reinforcement learning. arXiv preprint arXiv:2311.00201, 2023.
> >
> > [3] P. Heredia, H. Ghadialy, and S. Mou. Finite-sample analysis of distributed q-learning for multi-agent networks. In 2020 American Control Conference (ACC), pages 3511–3516. IEEE, 2020.
> >
> > [4] S. Zeng, T. T. Doan, and J. Romberg. Finite-time convergence rates of decentralized
> > stochastic approximation with applications in multi-agent and multi-task learning. IEEE
> > Transactions on Automatic Control, 2022b.
> >
> > [5] Even-Dar, Eyal, Yishay Mansour, and Peter Bartlett. "Learning Rates for Q-learning." Journal of machine learning Research 5.1 (2003).

---

> ### Author Response · Authors · 2024-11-22
>
> We sincerely thank the reviewer for the time and effort in evaluating our manuscript. If there are any further concerns, please feel free to inform us as the discussion period is nearing its end.

---

> > ### Comment · Reviewer_3Rf8 · 2024-11-26
> >
> > I sincerely appreciate the authors' responses. I am still uncertain about the significance of the paper's contribution in relation to the existing literature, and I would like to maintain my score.

---

> ### Author Response · Authors · 2024-11-26
>
> We appreciate the reviewer's thoughtful engagement in the discussion.
>
> Our research significantly diverges from the works in references [3, 4], which rely on linear function approximation. While the analysis in [3, 4] leverages convex optimization, which is feasible due to **their restrictive assumptions**, **we cannot follow such approach** since we do not impose their assumptions. In essence, **linear function approximation (with assumptions in [3,4]) and tabular Q-learning are fundamentally distinct problems**. The extensive literature in the field substantiates this distinction. A substantial body of finite-time Q-learning analyses has predominantly focused on the tabular setting not linear function approximation. This is because the assumptions in [3, 4] effectively reduce the analysis to a convex optimization approach, which oversimplifies the inherent difficulty of Q-learning and deviates from its core challenge.
>
>
> Finally, we acknowledge that our bound is suboptimal compared to the single-agent case, as explicitly stated in the manuscript. Nevertheless, our work provides a first finite-time bound for the  QD-learning algorithm [Kar et al.], a pioneering algorithm in the distributed RL literature. Given the substantial time—over 20 years—required to achieve an optimal bound in the single-agent setting, we believe our contribution both significant and original.

---

### Author Response · Authors · 2024-11-18
**General response (1/2)**

We would like to thank all the reviewers for their valuable feedback, as well as for the time and effort they dedicated to reviewing our paper. In response to their comments, we have made several revisions to the manuscript, with the changes highlighted in $\textcolor{red}{red}$. Below, we provide a summary of the modifications in the revised version:


**G1. Comparison with Wang et al.:**

Reviewer Gtxi pointed out a closely related work by Wang et al., which we have missed in the initial submission. The work proposed a distributed Q-learning algorithm in the tabular setting, which is motivated from the adapt-then-combine algorithm [2], whereas our algorithm considers combine-and-adapt scheme in the distributed optimization literature. The combine-and-adapt scheme first aggregates information from the neighbours and then executes update while in the adapt-and-combine scheme, the update occurs before aggregation. The work presents a sharper bound on the sample complexity $\frac{1}{(1-\gamma)^5d_{\min}\epsilon^2}$ compared to ours, $\frac{1}{(1-\gamma)^6d_{\min}^4\epsilon^2}$. However, we would like to note several differences and advantages of the proposed analysis as follows:**1)** In Wang et al., the finite-time bound only holds for restricted range of $\epsilon$, i.e., $\epsilon\in \left[0,\frac{1}{1-\gamma} \right)$, while our result does not have such a restriction.**2)** More importantly, the algorithm proposed by Wang et al. requires two steps for a single update, whereas in our paper, we focus on a one-step algorithm that is algorithmically simpler and more efficient. Specifically, we analyze the traditional and widely adopted QD-learning algorithm proposed in [4], for which a finite-time error analysis for the original form has been lacking in the literature. Additionally, we enhance the efficiency of QD-learning by employing a constant step-size, as opposed to the two-time-scale decaying step-size used in traditional QD-learning. This modification can significantly improve the convergence speed empirically.

**G2. Sub-optimal sample complexity:**

As we have mentioned as the limitation of our work, the dependency on $d_{\min}$ or $\frac{1}{1-\gamma}$ is sub-optimal compared to the single-agent case. While we acknowledge there is a room for improvement in our finite-time bound, even in the case of single-agent Q-learning, it took over 20 years of research—from [2] to [1]—to refine the sample complexity. Therefore, we believe our work represents a significant advancement in the field.

---

> ### Author Response · Authors · 2024-11-18
> **General response (2/2)**
>
> **G3. Novelty and contribution of the paper:**
>
> In the following, we summarize the novelty and contribution of our paper:
>
> 1) We derived a new theoretical result for distributed Q-learning, addressing the problem under more relaxed assumptions compared to previous approaches [Heredia et al., 2020; Zeng et al., 2022b]. In our paper, we derive, for the first time, the finite-time analysis of QD-learning in its original form, which is one of the most fundamental and widely used distributed Q-learning methods. While several works have addressed other types of distributed Q-learning, the analysis of QD-learning has remained unexplored until now. We believe that analyzing the original QD-learning is a valuable topic in its own right. Moreover, unlike the previous works, we adopted the switched system analysis of Q-learning. The previous works relied on stronger assumptions that
> allows their analysis to follow similar lines to that of convex optimization literature. To the best of our knowledge, there is no existing literature that demonstrates how to extend convex optimization analysis, or an analogous approach, to the analysis of Q-learning under the tabular setup.  As a result, neither the findings of existing works can be directly applied to our context, nor can they be generalized. Furthermore, compared to the work of Wang et al., our bound holds for any $\epsilon >0$ whereas the bound in Wang et al. holds for restricted range of $\epsilon$, i.e., $\epsilon\in \left[0,\frac{1}{1-\gamma} \right)$, while our result does not have such a restriction. The related discussions have been or will be added in the revised version.
>
>
> 2) We developed a new approach for the analysis of distributed Q-learning. Our contribution lies in extending the switched system analysis of single-agent Q-learning [3] to the multi-agent case. We would like to emphasize that this extension is non-trivial, as it required overcoming significant challenges, which in turn led to the development of new concepts, tools, and insights. In details, our contribution lies in using the decomposition of dynamics of optimality error in equation (9) in the manuscript, which allows us to apply the techniques from the switched system analysis of Q-learning. In particular, the key difference from the single-agent case is that we take average of the maximum of Q-function of each agent, i.e., the term $\frac{1}{N}\sum_{i=1}^N\Pi^{Q^i_k}Q^i_k$ in equation (9) in the manuscript, rather than the maximum of average of Q-function of each agents, .i.e., $\Pi^{Q^{avg}\_k}Q^{avg}\_k$. This poses difficulty in the analysis since $\frac{1}{N}\sum_{i=1}^N\Pi^{Q^i_k}Q^i_k$ cannot be represented in terms of $Q^{avg}\_k$. Consequently, it makes difficult to interpret it as switched affine system whose state-variable is $Q^{avg}\_k$. We introduce an additional error term $\frac{1}{N}\sum_{i=1}^N\ \Pi^{Q^i_k}Q^i_k-\Pi^{Q^{avg}_k}Q^{avg}_k$, and this allows us to follow the switched system analysis. The related discussions have been or will be added in the revised version.
>
> **References**
>
> [1] Wang, Yudan, et al. "Data-driven robust multi-agent reinforcement learning." 2022 IEEE 32nd International Workshop on Machine Learning for Signal Processing (MLSP). IEEE, 2022.
>
> [2] Chen, Jianshu, and Ali H. Sayed. "Diffusion adaptation strategies for distributed optimization and learning over networks." IEEE Transactions on Signal Processing 60.8 (2012): 4289-4305.
>
> [3] D. Lee, J. Hu, and N. He. A discrete-time switching system analysis of q-learning. SIAM Journal
> on Control and Optimization, 61(3):1861–1880, 2023.
>
> [4] Kar, Soummya, José MF Moura, and H. Vincent Poor. "${{\cal Q}{\cal D}} $-Learning: A Collaborative Distributed Strategy for Multi-Agent Reinforcement Learning Through ${\rm Consensus}+{\rm Innovations} $." IEEE Transactions on Signal Processing 61.7 (2013): 1848-1862.

---

### Meta-Review · Area_Chair_a1Td · 2024-12-18

**Metareview:**

This paper studies the finite-sample analysis of a type of distributed Q-learning algorithm, QD-learning, in multi-agent distributed reinforcement learning. The analyses extended the switched-system-based perspective and technique for analyzing (single-agent) Q-learning, to this multi-agent distributed case. The paper is overall well-written, with solid technical results. However, there were some concerns regarding the technical novelty, the significance and suboptimality of the bounds, and the lack of comparison with some closely related works. The clarity of some technical results/notation and the experimental results could also be improved to strengthen the paper further. I suggest the authors incorporate the feedback from this round, and prepare for other upcoming ML venues.

**Additional Comments On Reviewer Discussion:**

There were some shared concerns regarding the technical novelty, the suboptimality of the bounds, and the lack of discussions of closely related literature. The authors acknowledged the feedback, and revised the paper accordingly. In particular, it included a detailed discussion of a closely related work [Wang et. al., 2024]. However, there were remaining concerns regarding the significance of the few advantages of the results in the present paper compared to those in [Wang et. al., 2024], at the cost of worse sample complexity.

---

### Decision · Program_Chairs · 2025-01-22

Reject